# Prefrontal mechanisms combining rewards and beliefs in human decision-making

Marion Rouault [1,2,3], Jan Drugowitsch [2,4] & Etienne Koechlin[1,2,5]

In uncertain and changing environments, optimal decision-making requires integrating reward expectations with probabilistic beliefs about reward contingencies. Little is known, however, about how the prefrontal cortex (PFC), which subserves decision-making, combines these quantities. Here, using computational modelling and neuroimaging, we show that the ventromedial PFC encodes both reward expectations and proper beliefs about reward contingencies, while the dorsomedial PFC combines these quantities and guides choices that are at variance with those predicted by optimal decision theory: instead of integrating reward expectations with beliefs, the dorsomedial PFC built context-dependent reward expectations commensurable to beliefs and used these quantities as two concurrent appetitive components, driving choices. This neural mechanism accounts for well-known risk aversion effects in human decision-making. The results reveal that the irrationality of human choices commonly theorized as deriving from optimal computations over false beliefs, actually stems from suboptimal neural heuristics over rational beliefs about reward contingencies.

---

[1] Institut National de la Santé et de la Recherche Médicale, Paris 75013, France. [2] Department of Cognitive Studies, ENS, PSL Research University, 29, rue d'Ulm, 75005 Paris, France. [3] Wellcome Trust Centre for Neuroimaging, University College London, 12 Queen Square, London, WC1N 3AR, UK. [4] Department of Neurobiology, Harvard Medical School, 220 Longwood Avenue. Goldenson Building, Boston, MA 02115, USA. [5] Université Pierre et Marie Curie, Paris 75005, France. Correspondence and requests for materials should be addressed to E.K. (email: etienne.koechlin@upmc.fr)

Everyday life features uncertain and changing situations associated with distinct reward contingencies. In such environments, optimal adaptive behaviour requires detecting changes in external situations, which relies on making probabilistic inferences about the latent causes or hidden states generating the external contingencies of the agent experiences. Previous studies show that humans make such inferences, i.e., they develop state beliefs to guide their behaviour in uncertain and changing environments[1–4]. More specifically, the prefrontal cortex (PFC) that subserves reward-based decision-making is involved in inferring state beliefs about how reward contingencies map onto choice options[5–8]. Optimal decision-making for driving behaviour then requires integrating these state beliefs and reward expectations through probabilistic marginalisation processes[9]. This integration is required to derive reward probabilities associated with choice options and to choose the option maximising the expected utility[10]. Consistently, PFC regions involved in inferring these state beliefs also exhibit activations associated with reward expectations[11–16].

However, human choices often differ from optimal choices systematically[17], raising the open issue of how the PFC combines these state beliefs and reward expectations to drive behaviour. A common hypothesis is that these quantities are integrated as posited in the expected utility theory, but choice computations derive from distorted representations of reward probabilities, usually named subjective probabilities[17–21]. Yet the origin of subjective probability remains unclear. As marginalisation processes are complex cross-product processes[9], the notion of subjective probability might then reflect that state beliefs and reward expectations are actually combined in a suboptimal way at variance with the expected utility theory[22]. Thus, an alternative plausible hypothesis is that state beliefs about reward contingencies are processed as an additional value component that contributes to choices independently of reward expectations rather than through marginalisation processes, i.e., state beliefs about reward contingencies act in decision-making as affective values that combine linearly with the appetitive value of reward expectations.

Here, we address this open issue using computational modelling and functional magnetic resonance imaging (fMRI). We confirm here that participants make decisions as if they marginalise reward expectations over state beliefs and compute choices based on distorted subjective probabilities. Using a model falsification approach[23], however, we show that participants' performance varies with these subjective probabilities in a way contradicting this theoretical construct. We then provide evidence that participants' choices actually derive from the independent contribution of state beliefs regarding the most frequently rewarded option and reward expectations based on an efficient coding mechanism of context-dependent value normalisation[24–26]. We identify the PFC regions involved in this decision process combining linearly these state beliefs and reward expectations, which at variance with the standard expected utility theory, results in (1) the mutual dependence of option utilities and (2) the processing of state beliefs as affective values rather than probability measures in decision-making.

## Results

**Behavioural protocol.** Twenty-two participants were asked to make successive choices between two visually presented one-armed bandits (square vs. diamond bandit, Fig. 1a) (Methods). In every trial, each bandit proposed a potential monetary reward varying pseudo-randomly from 2 to 10 €. One bandit led to rewards more frequently (frequencies: $q_M = 80\%$ vs. $q_m = 20\%$). Following participants' choices, the chosen-bandit outcome was

visually revealed, with zero indicating no rewards. Otherwise, participants received the proposed reward approximately (±1 €). Reward frequencies episodically reversed between the two bandits (unpredictably every 16–28 trials), so that bandits' reward frequencies remained uncertain to participants. This uncertainty induces the formation of probabilistic state beliefs about the identity of the 80% and 20% rewarded bandit. In the neutral condition, proposed rewards were independent of bandits, so that beliefs could be inferred only from previous choice outcomes (Fig. 1b). To properly dissociate belief probabilistic inferences from reinforcement learning processes (RL), the protocol included two additional conditions (administered in separate days): in the congruent condition, proposed rewards were biased towards higher values for the more frequently rewarded bandit (and vice versa), whereas in the incongruent condition, proposed rewards were biased in the exact opposite direction. In both these conditions, thus, proposed rewards identically convey some additional information about bandits' reward frequencies dissociable from reward values: beliefs could be inferred in every trial from both previous choice outcomes and proposed rewards. Thus, the protocol properly dissociated belief inferences from RL processes over trials. Note also that due to the reversal/symmetrical structure of the protocol, the task required no exploration for maximising rewards: participants got the same information about bandits' reward frequencies, whatever the option they choose in every trial.

In every trial, the optimal performance model (named model OPT) forms probabilistic beliefs from previous trials about how reward frequencies map onto bandits, updates these beliefs according to proposed rewards and finally, chooses the bandit maximising the (objective) expected utility by marginalising reward expectations over beliefs[27] (Methods). After reversals, model OPT gradually acquires almost perfect beliefs and regardless of conditions, starts selecting the true best bandit (i.e., maximising reward frequencies x proposed rewards) almost systematically (Fig. 1c). This optimal performance is reached similarly in the congruent and incongruent conditions, but is slower in the neutral condition. As expected, participants performed suboptimally: after reversals, their performance gradually reached a plateau, selecting the true best bandits with a maximal frequency close to ~80% (corresponding to probability matching) which in contrast to optimal performance, further decreased monotonically from the congruent to neutral and incongruent condition (mean over trials from trial#10, paired T-tests: both $Ts(21) > 2.97$, $ps < 0.01$) (Fig. 1c).

**Distortion models of human decision-making.** To account for human suboptimal performances, we first considered the standard distortion hypothesis named model DIST[17–19,21]. This model is identical to model OPT, except that choices maximise the subjective expected utility of choice options involving: (1) subjective probabilities distorting objective probabilities that derive from marginalising reward expectations over state beliefs; (2) subjective appetitive values distorting monetary values of proposed rewards. We used the standard distortion function encompassing all previously documented distortions (convex, concave, S-shape and inverted S-shape) on reward probabilities derived from state beliefs as well as on proposed rewards modelling subjective appetitive values[21] (Methods). Model DIST thus includes four additional free parameters characterising such distortions with model OPT as a special case.

Suboptimal performances might also stem from RL processes whereby the reward history of past choices biases current choices. To assess this effect, we considered the extension of model DIST (referred to as model DIST+RL) comprising an additional RL

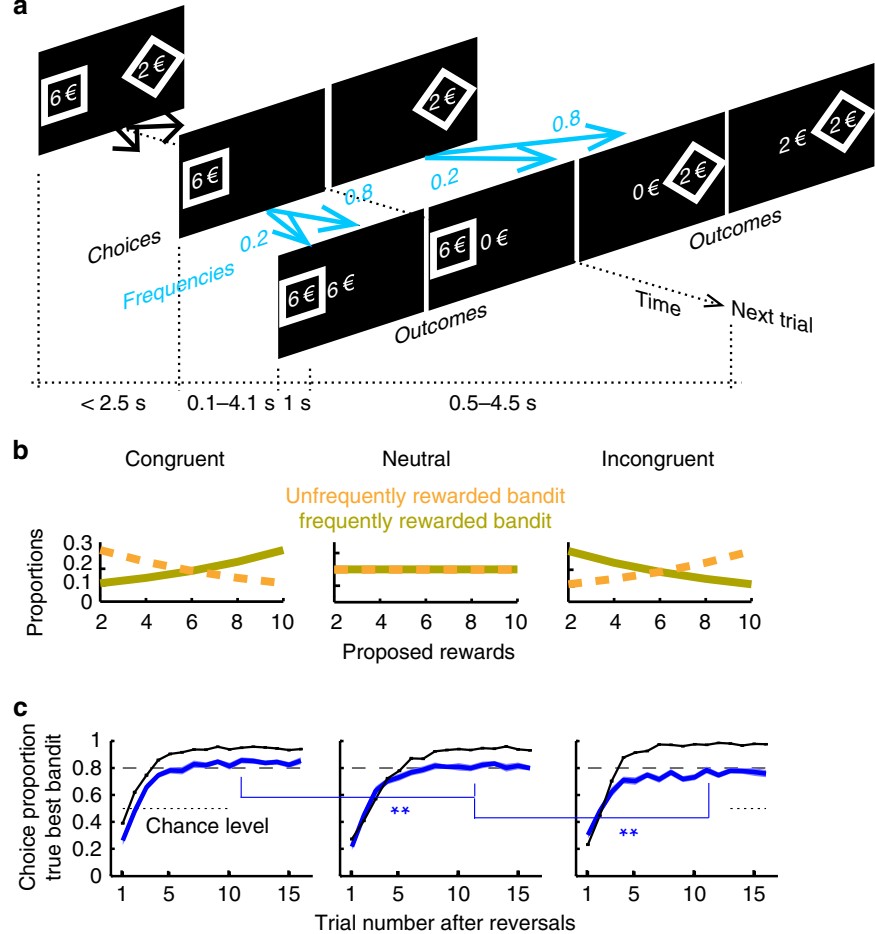

**Fig. 1** Behavioural protocol. **a** trial structure. A square and diamond one-armed bandit with the offered rewards (euros) were presented on the screen (maximal duration: 2.5 s) until participants chose one bandit by pressing one response button. The chosen bandit remained on display. A feedback centred on the screen was then presented to reveal the bandit outcome (duration 1 s). One bandit led to proposed rewards (±1 €) more frequently (blue arrows 80% vs. 20%) but this advantage reversed episodically. The next trial started with the presentation of both bandits again. Response-feedback onset asynchronies and intertrial intervals were uniformly and independently jittered (ranges: 0.1–4.1 s and 0.5–4.5 s, resp.). **b** Proposed rewards were biased in opposite directions in the congruent and incongruent condition (exponential biases: slope=±0.13). **c** Proportions of choosing true best bandits (maximising reward frequencies × proposed rewards) following reversals. Mean proportions over participants (blue, ±s.e.m, $N = 22$) and for optimal model OPT (black line) are shown in the congruent, neutral and incongruent condition. Dashed line corresponds to the 80% reward frequency. **p < 0.01 (T-tests)

component based on the Rescorla & Wagner's rule[28] that alter the subjective expected utilities of bandits (Methods).

**Mixture models of human decision-making**. As outlined in the Introduction, the distortion hypothesis might reflect that state beliefs and reward expectations are actually not integrated as posited in the expected utility theory (i.e., as in models OPT and DIST)[22]. First, the distortion over monetary values of proposed rewards might actually reflect the recently proposed idea that option utilities are encoded in the brain according to a context-dependent divisive normalisation mechanism[25,26], so that utilities of choices' options are mutually dependent. Second, the notion of subjective probabilities distorting objective probabilities might actually reflect that no marginalisation processes occur over abstract state beliefs such as bandit $i$ is rewarded with frequency $q_i$ (as in models OPT and DIST). Instead, beliefs might bear upon affective states such as bandit $i$ is the most frequently rewarded bandit, so that utilities and beliefs act as two normalised appetitive value components contributing to choices independently, i.e., additively. These two hypotheses, utility-normalisation and affect-additivity, are tightly linked because (1) affect-additivity requires making utilities commensurable to probabilistic beliefs, thereby

leading to utility-normalisation; (2) model OPT with normalised utilities and affective state beliefs is equivalent to maximising a weighted sum of normalised utilities and affective state beliefs with specific weights denoted $1-\omega_0$ and $\omega_0$, respectively (Methods).

Accordingly, we proposed the more general decision-making model (named model MIX) assuming that choices maximise a linear combination of normalised utilities and affective state beliefs with weighting parameters $1-\omega$ and $\omega$, respectively (Methods). If $\omega$ significantly differs from critical value $\omega_0$, choices are unlikely to derive from computing bandits' expected normalised utilities based on marginalisation processes over affective state beliefs. Moreover, $\omega > \omega_0$ vs. $\omega < \omega_0$ implies more belief-based (risk-adverse) vs. more utility-based (risk-prone) choices, respectively. Model MIX includes no distortion functions but, according the context-dependent divisive normalisation[25,26], utilities $v_1$, $v_2$ of choice options correspond to proposed rewards possibly altered by the reward history and normalised across the two choice options:

$$v_i = \frac{\varphi V_i^{\text{proposed}} + (1-\varphi)V_i^{\text{RL}}}{\sum_{i=1,2}\left(\varphi V_i^{\text{proposed}} + (1-\varphi)V_i^{\text{RL}}\right)}$$

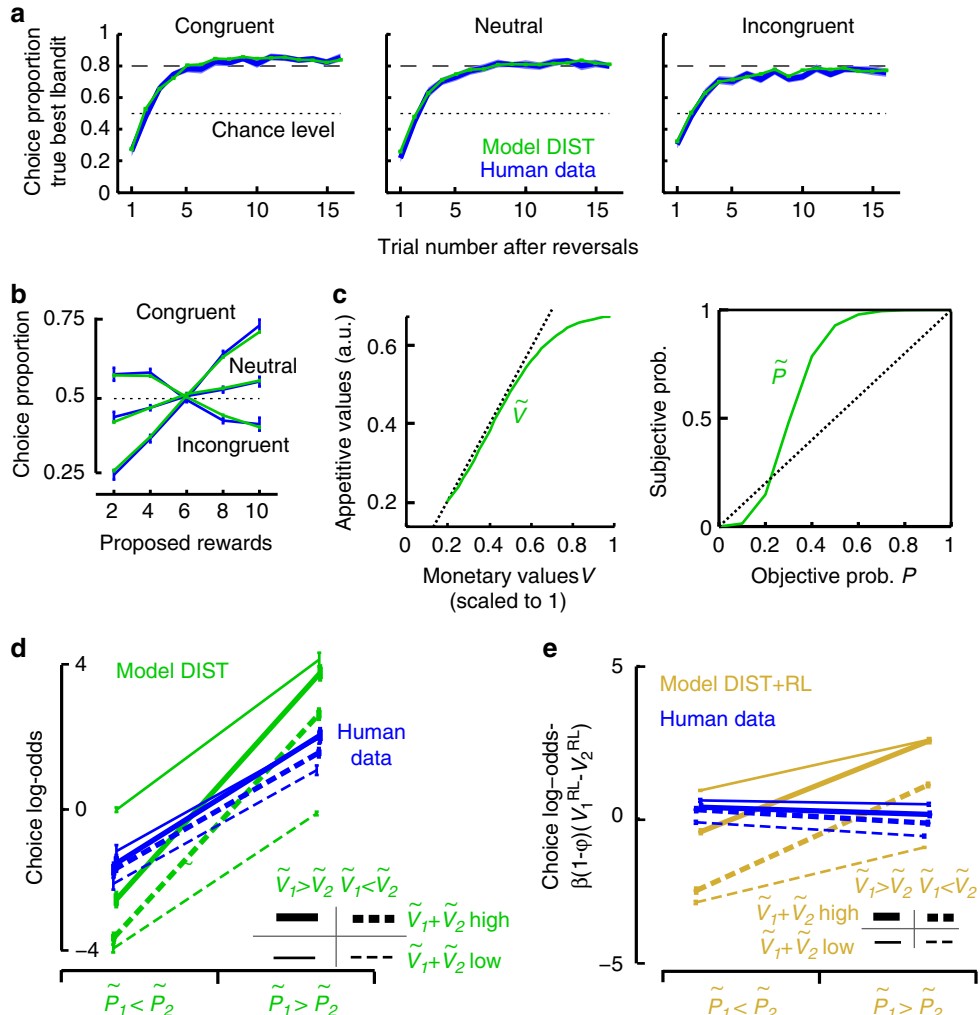

**Fig. 2** Model DIST fit to human performances. **a** Proportions of choosing true best bandits (maximising reward frequencies x proposed rewards) following reversals. **b** Proportions of choosing proposed rewards. In **a** and **b**, mean proportions are shown in the congruent, neutral and incongruent condition for fitted model DIST (green) and participants (blue, ±s.e.m.). See Supplementary Fig. 1 for (parameterised) model OPT fit. **c** Appetitive values of proposed monetary rewards $\tilde{V}$ (left) and subjective probabilities in inferred bandits' reward probabilities $\tilde{P}$ (right) in model DIST fitting human choices (lines correspond to distortion functions with mean parameters over participants). **d** Factorial analysis of choice log-odds over trials sorted according to subjective probabilities $(\tilde{P}_1 > \tilde{P}_2$ vs. $\tilde{P}_1 < \tilde{P}_2)$, appetitive values of proposed rewards $(\tilde{V}_1 > \tilde{V}_2$ vs. $\tilde{V}_1 < \tilde{V}_2)$ and total appetitive values $(\tilde{V}_1 + \tilde{V}_2$, median split) for fitted model DIST simulations (green) and human data (blue). Both model DIST and participants exhibited main effects of subjective probabilities and relative appetitive values (all $Fs(1,21) > 23.7$, $ps < 0.00001$) with no main effects of total appetitive values (both $Fs < 1$). However, model DIST unlike participants exhibited an interaction between subjective probabilities and total appetitive values (DIST: $F(1,21) = 2269$, $p < 0.00001$. participants: $F < 1$). **e** Same factorial analysis of choice log-odds adjusted for RL-values for fitted model DIST+RL and human data (see text). Model DIST+RL exhibited main effects of subjective probabilities, relative appetitive values and total appetitive values (all $Fs(1,21) > 7.8$, $ps < 0.01$), along with an interaction between subjective probabilities and total appetitive values ($F(1,21) = 691$, $p < 0.00001$). By contrast, participants exhibited only a main effect of relative appetitive values ($F(1,21) = 18.5$, $p < 0.0001$): participants exhibited no main effects of subjective probabilities ($F(1,21) = 3.0$, $p = 0.10$), and neither main nor interaction effects associated with total appetitive values (both $Fs(1,21) < 1.2$, $ps > 0.29$). See Supplementary Table 1 for model best-fitting parameters and Supplementary Fig. 2 for model-free analysis. Error bars are s.e.m. over participants ($N = 22$)

where $\varphi$ is a free weighting parameter, $V_i^{\text{proposed}}$ the monetary value of proposed rewards and $V_i^{\text{RL}}$ the RL-values associated with band*i*ts *i*.

**Behavioural results**. We fitted all models assuming choices to vary as a softmax function of decision variables with inverse temperature $\beta$ as free parameter. Consistent with previous studies about experience-based choices[19], model DIST exhibited major sigmoid-like distortions on subjective probabilities with under- and over-estimations of low and high probabilities, respectively (Fig. 2c, Supplementary Table 1). Appetitive values of proposed

rewards further varied as the standard concave utility function of monetary values posited in expected utility theory[29] (Fig. 2c). Importantly, model DIST simulations reproduced participants' choices in every condition both in choosing true best bandits along trial series following reversals (Fig. 2a) and in bandits' choices according to proposed rewards (Fig. 2b) (T-tests, all $ps > 0.05$). Moreover, model DIST fitted participants' behaviour better than all its reduced models comprising fewer free parameters (including model OPT), even when penalising for model complexity (Bayesian Information Criteria, paired T-tests, DIST vs. reduced models: all $Ts(21) > 8.0$, $ps < 10^{-7}$) (Methods).

Simulations of reduced models with no distortions significantly differed from participants' performances especially in the congruent and incongruent condition and with respect to proposed rewards (Supplementary Fig. 1).

Using a model falsification approach[23], we then tested a critical qualitative prediction from model DIST. This model assumes choice frequency log-odds $\log\frac{p_1}{p_2}$ between bandits to vary with the difference between subjective expected utilities of bandits:

$$\log\frac{p_1}{p_2} = \beta(\tilde{P}_1\tilde{V}_1 - \tilde{P}_2\tilde{V}_2) \tag{1}$$

or equivalently,

$$\log\frac{p_1}{p_2} = \frac{\beta}{2}\left[(\tilde{V}_1 + \tilde{V}_2)(\tilde{P}_1 - \tilde{P}_2) + (\tilde{P}_1 + \tilde{P}_2)(\tilde{V}_1 - \tilde{V}_2)\right] \tag{2}$$

where $p_1$, $p_2$ are choice frequencies, $\tilde{V}_1, \tilde{V}_2$ appetitive values (utilities) and $\tilde{P}_1, \tilde{P}_2$ subjective probabilities of proposed rewards associated with bandits 1 and 2, respectively. Equation (2) predicts log-odds $\left(\log\frac{p_1}{p_2}\right)$ to exhibit an interaction effect between total appetitive values $\tilde{V}_1 + \tilde{V}_2$ and relative probabilities $\tilde{P}_1 - \tilde{P}_2$, along with an additive effect of $(\tilde{V}_1 - \tilde{V}_2)(\tilde{P}_1 + \tilde{P}_2)$. To test this prediction, we entered choice frequency log-odds from participants and model DIST simulations in a $2\times 2\times 2$ factorial analysis including as within-subject factors: total values $(\tilde{V}_1 + \tilde{V}_2)$ (median split), relative probabilities $\tilde{P}_1 - \tilde{P}_2$ ($\tilde{P}_1 < \tilde{P}_2$ vs. $\tilde{P}_1 > \tilde{P}_2$) and relative values $\tilde{V}_1 - \tilde{V}_2$ ($\tilde{V}_1 < \tilde{V}_2$ vs. $\tilde{V}_1 > \tilde{V}_2$); due the protocol reversal structure, $\tilde{P}_1 + \tilde{P}_2$ was constant across these eight cells and consequently not included in the analysis). As shown in Fig. 2d, these log-odds exhibited additive main effects associated with relative probabilities and values but unlike model DIST, participants' log-odds exhibited no interactions between total appetitive values and relative probabilities. Thus, even though participants performed according to external reward contingencies as if they computed subjective expected utilities, their performance varied with subjective probabilities and appetitive values in a way contradicting this theoretical construct.

We then fitted model DIST+RL to participants' choices. The increased complexity of model DIST+RL, however, failed to improve the fit (BIC$_{\text{DIST}}$ < BIC$_{\text{DIST+RL}}$; $T(21) = 0.77$, $p = 0.44$). Moreover, model DIST+RL assumes choice log-odds to vary as follows:

$$\log\frac{p_1}{p_2} = \frac{\beta}{2}\left[(\tilde{V}_1 + \tilde{V}_2)(\tilde{P}_1 - \tilde{P}_2) + (\tilde{P}_1 + \tilde{P}_2)(\tilde{V}_1 - \tilde{V}_2)\right]\varphi + \beta(V_1^{\text{RL}} - V_2^{\text{RL}})(1 - \varphi) \tag{3}$$

with $\varphi$ a free parameter weighting subjective expected utilities and RL-values $V_i^{\text{RL}}$, model DIST+RL thus predicts the quantity $\left[\log\frac{p_1}{p_2} - \beta(1-\varphi)(V_1^{\text{RL}} - V_2^{\text{RL}})\right]$ to vary with appetitive values and subjective probabilities as in model DIST. Using the same factorial analysis as described above, we again found that in contrast to the prediction, this quantity varied in participants with no interactions between total appetitive values and subjective probabilities (Fig. 2e). Thus, neither models DIST/DIST+RL nor a fortiori all their reductions, including especially the standard mixture OPT+RL predicting the same interaction effects, accounted for how participants made choices. A model-free logistic regression analysis of participants' choices on protocol parameters confirmed these results (Supplementary Fig. 2).

We then fitted model MIX to participants' choices. As model DIST, model MIX simulations reproduced participants' choices in every condition, both in choosing true best bandits along trial series following reversals (Fig. 3a) and in bandits' choices according to proposed rewards (Fig. 3b) (T-tests; all $ps>0.05$), while the reduced models failed (Supplementary Fig. 2). Consistently, model MIX fitted participants' behaviour better than all its reduced models comprising fewer free parameters (including especially pure RL and no RL), even when penalising for model complexity (BIC, MIX vs. reduced models: $Ts(21) > 2.92$, $ps < 0.0082$) (Methods). As expected additionally, removing the divisive normalisation over utilities in model MIX while keeping the same variables (affective beliefs, proposed rewards and RL values) also degraded the fit (BIC$_{\text{MIX}}$ < BIC$_{\text{NoNorm}}$, $T(21) = 1.65$, $p = 0.05$, one-tailed).

Fitted parameters in model MIX (Supplementary Table 2) indicate that participants' choices strongly maximised the weighted sum of normalised utilities and affective state beliefs (inverse temperature $\beta = 54.9$, s.e.m = 9.6; lapse rates $\varepsilon = 2\%$, s.e.m. = 1%). Parameters accounting for proposed reward biases altering state beliefs across conditions matched their true values, suggesting that participants inferred affective state beliefs through optimal Bayesian inferences (as in model OPT) (Fig. 3c). These affective beliefs contributed to choices more strongly than normalised utilities, as weight parameter $\omega$ was larger than critical value $\omega_0 = 0.375$ ($\omega = 0.69$, s.e.m. = 0.06; one sample T-test, $T(21) = 4.84$, $p < 0.001$). This first indicates that as expected, participants were risk-adverse and preferentially chose safer bandits. Second, beliefs contributed to choices as an additional component of appetitive values rather than through marginalisation processes. As expected also, RL-values strongly altered proposed reward utilities ($1 - \varphi = 0.8$, s.e.m. = 0.07).

Using a model recovery procedure[23], we confirmed that even though both models MIX and DIST reproduced participants' performances according to external reward contingencies, they resulted in distinct detectable behaviours (Methods). Critically, model MIX fitted participants' choices better than model DIST and a fortiori, hybrid models DIST+RL and OPT+RL (BIC$_{\text{MIX}}$ < BIC$_{\text{DIST}}$; < BIC$_{\text{DIST+RL}}$; < BIC$_{\text{OPT+RL}}$: all $Ts(21) > 3.54$, $ps < 0.005$, non-nested models). Unlike DIST, model MIX indeed predicts choice log-odds $\log\frac{p_1}{p_2}$ to vary as the addition of relative normalised utilities and affective beliefs. As described above, we entered both MIX's and participants' choice log-odds in a factorial analysis including relative normalised utilities ($v_1 - v_2$) and affective beliefs ($B_1 - B_2$) from model MIX, along with total proposed rewards $V_1^{\text{pr}} + V_2^{\text{pr}}$ as within-subject factors. We found that both MIX's and participants' choice log-odds varied as the virtually identical addition of relative utilities and beliefs with no effects of total proposed rewards (Fig. 3d). Thus, model MIX accounted for how participants chose bandits from inferred beliefs, reward expectations and history.

Finally, both model DIST and MIX assume in agreement with the protocol that participants infer state beliefs rather than directly learn reward probabilities through RL processes. To validate this assumption, we fitted the corresponding RL variants of both models. As expected, we found that both model DIST and MIX outperformed these variants in predicting participants' choices (Likelihood$_{\text{DIST/MIX}}$>Likelihood$_{\text{any/variants}}$: all $Ts(21) > 10.23$, $ps < 10^{-8}$; BIC$_{\text{DIST/MIX}}$ > BIC$_{\text{any/variants}}$: all $Ts(21) > 7.39$, $ps < 10^{-6}$).

**fMRI activations**. Using fMRI, we investigated the PFC processes underpinning model MIX. As a necessary (but not sufficient) condition, brain regions involved in computing choices exhibit activations varying with decision entropy, i.e., activations increase when the decision variable becomes more ambiguous[30–33]. Model MIX thus predicts these activations to vary as an inverted, zero-centred U-shape function of decision variable

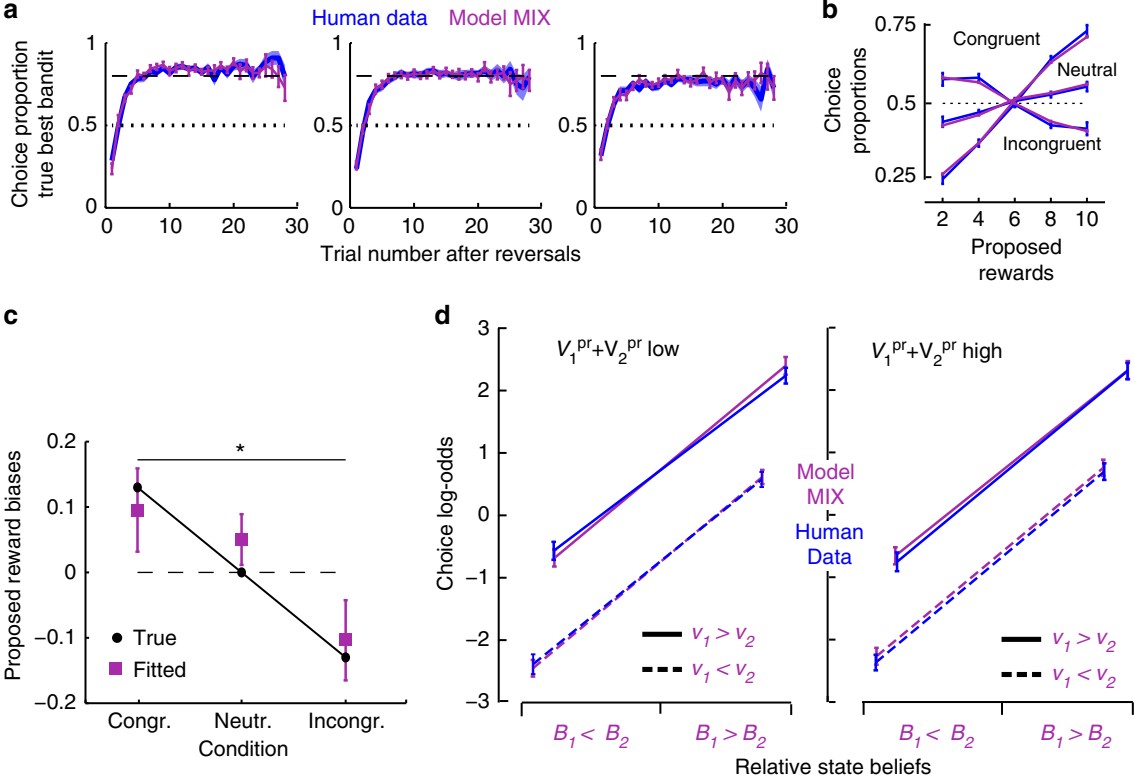

**Fig. 3** Model MIX fit to human performances. **a** Proportions of choosing true best bandits (maximising reward frequencies x proposed rewards) following reversals. **b** Proportions of choosing proposed rewards. In **a** and **b**, mean proportions over individuals (±s.e.m.) are shown in the congruent, neutral and incongruent condition for participants (blue), and fitted model MIX (purple) simulations. See Supplementary Fig. 1 for model RL fit. **c** Parameter estimates of proposed rewards biases in the congruent, neutral and incongruent conditions in model MIX fitting human choices, *$p < 0.05$ (T-tests). These estimates are similar to true values (±0.13, see Fig. 1b) indicating that according to model MIX, participants adequately inferred state beliefs. **d** Factorial analysis of choice log-odds over trials sorted according to inferred state beliefs ($B_1 > B_2$ vs. $B_1 < B_2$), normalised utilities ($v_1 > v_2$ vs. $v_1 < v_2$) and total proposed rewards ($V_1^{pr} + V_2^{pr}$, median split: left vs. right panel) for model MIX simulations (purple) and human data (blue). Both model MIX and human data exhibited main effects of beliefs and utilities (all $F$s(1,21) > 116.9, $p$s < 0.00001), no main effects of total proposed rewards (both $F$s(1,21) < 1.2, $p$s > 0.47 and no interactions between these factors (all $F$s(1,21) < 3.3, $p$s > 0.14). See Supplementary Table 2 for model best-fitting parameters. Error bars are s.e.m. over participants ($N = 22$)

$[(1 − ω)(v_1 − v_2) + ω(B_1 − B_2)]$ (Methods). To identify these activations, we therefore entered this decision variable as a quadratic regressor $[(1 − ω)(v_1 − v_2) + ω(B_1 − B_2)]^2$ locked at decision time (bandit onsets) in a multiple-regression analysis of brain activations factoring out no-interest variables (Methods). The whole brain analysis (Fig. 4, Supplementary Table 3) revealed activations in the dorsomedial PFC (dmPFC including the dorsal anterior cingulate cortex), bilateral PFC (Brodmann area 9/8), inferior parietal lobules, precuneus, right insular and frontopolar cortex. In the frontal lobes, however, only dmPFC and right PFC activations remained significant when factoring out reaction times (Fig. 4, Supplementary Fig. 3). The dmPFC and right PFC were thus the frontal regions potentially involved in computing choices.

We more closely examined frontal activations using significance thresholds set to $p = 0.05$ (T-tests) corrected for family-wise errors over the frontal lobes along with post hoc analyses removing selection biases[34] (Methods). Reaching a decision requires less processing resources when the decision-relevant evidence supporting actual choices increase, so that regions involved in choice computations exhibit decreasing activations when the decision-relevant evidence supporting actual choices increase[30–33,35]. Consequently, model MIX specifically predicts these regions to exhibit two additive decreasing effects as reflecting the linear combination of affective state beliefs and

normalised utilities guiding choices: activations should decrease when affective state beliefs or normalised utilities associated with chosen bandits increase, with no interactions between these effects (i.e., when $B_{chosen}$ and $v_{chosen}$ or equivalently, $B_{chosen} − B_{unchosen}$ and $v_{chosen} − v_{unchosen}$ increase, as these variables are normalised) (Methods). We then entered (relative) chosen beliefs and utilities as two linear regressors locked at decision time in a second multiple-regression analysis factoring out no-interest variables (Methods). A conjunction (intersection) analysis revealed only one PFC region exhibiting the addition of the two predicted negative linear effects, namely the dmPFC region reported above (Fig. 5, Supplementary Table 4): dmPFC activations decreased when (relative) chosen beliefs or utilities increased, with no interactions between these factors ($p > 0.05$, uncorrected). This finding along with the inverted quadratic effects associated with decision variable $(1 − ω)(v_1 − v_2) + ω(B_1 − B_2)$ provide evidence that the dmPFC guided choices based on additively combining state beliefs and normalised utilities, as predicted by model MIX.

To further investigate PFC regions encoding affective state beliefs and normalised utilities irrespective of choice computations, we followed the method described in ref. [31]. Accordingly, we included in the preceding regression analysis two additional quadratic regressors $(B_{chosen} − B_{unchosen})^2$ and $(v_{chosen} − v_{unchosen})^2$ (equal to $(v_1 − v_2)^2$ and $(B_1 − B_2)^2$). The rationale is that while the

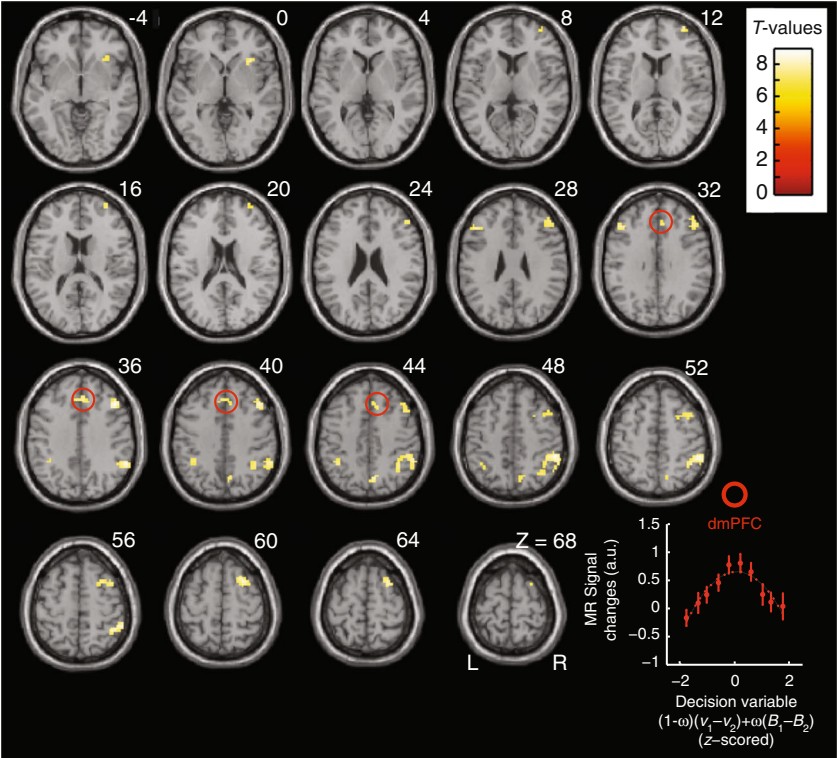

**Fig. 4** Activations associated with decision entropy. Yellow, brain activations associated with an inverted U-shape function of decision variable $(1 - \omega)$ $(v_1 - v_2) + \omega (B_1 - B_2)$ centred on zero from model MIX, overlaid on a canonical single-subject anatomical template (neurological convention). Axial slices are indexed by their MNI $Z$-coordinate. Red circles highlight dorsomedial prefrontal activations (dmPFC, Broadman's area 32). Voxel-wise threshold set at $p < 0.05$ (T-tests) corrected for family-wise errors for multiple comparisons over the whole brain. Cluster-wise threshold $p < 0.05$. See Supplementary Table 3 for MNI-coordinates of activation peaks. The graph shows the actual variations of dmPFC activations according to the decision variable irrespective of reaction times (see Supplementary Fig. 3). Error bars are s.e.m. over participants ($N = 21$)

linear regressors associated with $(B_\text{chosen} - B_\text{unchosen})$ and $(v_\text{chosen} - v_\text{unchosen})$ capture the activations reflecting choice computations, these quadratic regressors capture the residual activations coding for choice-relevant information, irrespective of actual choices. Indeed, regions coding for belief (utility, resp.) information are predicted to exhibit activations decreasing with belief entropy (utility entropy, resp.), i.e., varying as a positive quadratic function of $B_1 - B_2$ ($v_1 - v_2$, resp.). Such activations simply convey belief and utility information of one bandit compared with the other bandit, irrespective of chosen bandits (Methods). The analysis of participants' reaction times supports this approach (Fig. 6).

We found activations conveying utility information only in the dmPFC identified above (Fig. 7, Supplementary Table 4), whereas activations conveying belief information were only located in the ventromedial PFC (vmPFC) or medial orbitofrontal cortex (Fig. 7, Supplementary Table 4). Consistent with the dominance of affective state beliefs in model MIX, vmPFC activations were much larger than dmPFC activations. Across participants, moreover, vmPFC activations increased with weight $\omega$ favouring affective beliefs in model MIX ($r = 0.44$, T-test, $p = 0.048$). Thus, the dmPFC encoded comparative information about normalised utilities, while the vmPFC encoded comparative information about affective state beliefs, irrespective of choice computations. This dissociation further suggests that the dmPFC functionally interacts with the vmPFC so that as reported above, affective state beliefs contribute to choice computations in the dmPFC along with normalised utilities. We therefore analysed the functional connectivity between these regions (psychophysiological interactions) and consistently found that similar to the dmPFC activations reflecting choice computations (see Fig. 5), the

correlation between dmPFC and vmPFC activations decreased when (relative) chosen beliefs increased (Supplementary Fig. 4).

All the effects reported above remained significant when the analysis factored out reaction times. Consistent with previous findings (see Discussion), activations were also found to linearly increased with (relative) chosen utilities. These activations were located in the vmPFC (Fig. 8a). Importantly, all these results were properly obtained in full variance regression analyses (Supplementary Methods). As expected also, additional fMRI analyses revealed no significant brain activations associated with distorted subjective probabilities posited in model DIST ($p > 0.01$, uncorrected).

Finally, we investigated whether activations associated with normalised utilities reflect proposed rewards, RL-values or both. This third regression analysis was identical to the preceding one, except that the linear and quadratic regressor modelling normalised utilities were each broken down into two regressors modelling separately RL-values and proposed rewards (Methods). No dmPFC activations were associated with these component regressors (Supplementary Table 5), indicating the dmPFC activations reported above were associated with normalised utilities rather than these value components. By contrast, vmPFC along with superior left PFC activations (superior frontal gyrus, BA 8, Frontal Eyes Field) increased with chosen-relative-to-unchosen proposed rewards rather than RL-values (Fig. 8b), indicating that vmPFC activations were actually associated with proposed rewards rather than normalised utilities. There were no other significant effects in the PFC (Supplementary Table 5) except a positive quadratic effect associated with relative RL-values in the bilateral orbitofrontal cortex (laOFC) indicating that laOFC encoded bandits' relative RL-values, irrespective of choice computations (Fig. 8b).

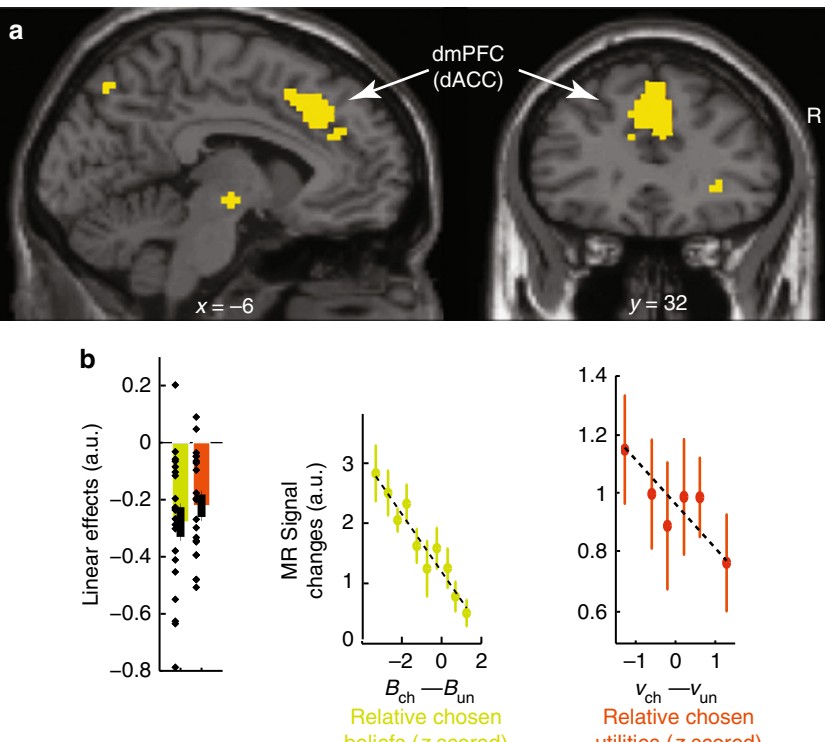

**Fig. 5** Prefrontal activations involved in computing choices. **a** Topography of frontal activations negatively correlated with relative chosen normalised utilities ($v_{chosen}-v_{unchosen}$) and with relative chosen state beliefs ($B_{chosen}-B_{unchosen}$) with no interactions between these factors ($p > 0.05$, uncorrected), from model MIX. Activations are thresholded at $p < 0.05$ (T-tests, voxel-wise corrected for family-wise errors over the frontal lobes: MNI coordinate $Y > 0$; cluster-wise threshold $p < 0.05$) and superimposed on sagittal and coronal anatomical slices (MNI template, $X = -6$ and $Y = 32$). Frontal activations are located in the dmPFC. See Supplementary Table 4 for coordinates of activation peaks. **b**, left: sizes of linear effects (betas) for relative chosen beliefs and normalised utilities (green and orange, resp.) averaged over the dmPFC activation cluster shown in **a** (leave-one-out procedure removing selection biases). **b**, right: activations averaged over the dmPFC activation cluster shown in **a** and plotted against relative chosen beliefs (no quadratic component factored out) and normalised utilities (the quadratic component being factored out is shown in Fig. 7), z-scored within each participant. All error bars are s.e.m. across participants ($N = 21$)

## Discussion

Our results confirm that as previously proposed, human adaptive behaviour in uncertain and changing environments is based on inferring state beliefs from past action outcomes and immediate contextual cues (proposed reward biases in the congruent/ incongruent condition)[1–4]. Our resuts further indicates that the expected utility theory[10] —requiring marginalising reward expectations over these beliefs—accounts reasonably well for human choices according to external contingencies, provided that one assumes choices to derive from *subjective* probabilities distorting objective probabilities. We replicated previous results showing that subjective probabilities built upon experience by under- and over-estimating low and high objective probabilities[19], respectively, while reward utilities (appetitive values) vary as a standard concave function of monetary values[29]. Consistently, pioneering studies often used this theoretical construct to investigate the neural bases of human decision-making[11,12,14]. However, we found that even though participants performed according to external contingencies "as if" their choices maximised subjective expected utilities, participants' choices actually varied with subjective probabilities and utilities in contradiction to this theoretical construct. Moreover, we observed no neural evidence supporting the notion of subjective probabilities.

Instead, our results support an alternative decision-making model (model MIX) based on two key features at variance with the standard utility theory. First, choices derived from *normalised* utilities stemming from an efficient coding mechanism of

context-dependent divisive value normalisation that makes utility values across choice options to be mutually dependent[24–26]. The mechanism was shown to account for human and monkey choices in static and deterministic environments[25,26,36]. Our findings expand this value-coding mechanism to human decision-making in changing and uncertain environments whereby utility values further depend upon the reward history and combine with state beliefs to guide choices. Consistently, we found the dmPFC involved in computing choices to exhibit activations associated with normalised utilities rather than its value components, namely reward expectations (proposed reward value) and reward history (RL-values).

Second, choices derived from linearly combining normalised utilities with undistorted affective state beliefs corresponding to optimal observer's beliefs regarding the currently, most frequently rewarded option. The linear combination generalises the optimal marginalization process over normalised utilities as the two processes become equivalent when the linear combination corresponds to equal contributions of beliefs and utilities to decision-making (Methods). However, we found affective state beliefs to outweigh normalised utilities and to be associated with larger PFC activations. Moreover, dmPFC activations guiding choices varied as the additive effects of normalised utilities and affective state beliefs. Thus, evidence was that choices derive from linearly combining normalised utilities and affective state beliefs rather than from marginalising the former over the latter. The dominant contribution of beliefs over utilities also reflects well-known

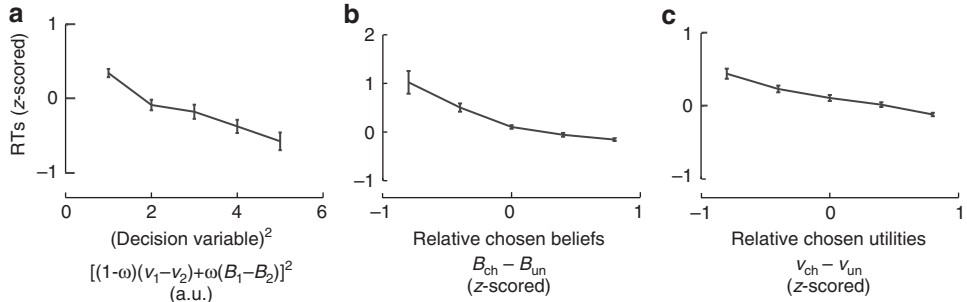

**Fig. 6** Reaction time analysis. Reaction times (RTs) plotted against the quadratic expansion of decision variable (**a**), relative chosen state beliefs (**b**, $z$-scored within participants) and relative chosen normalised utilities (**c**, $z$-scored within participants). Error bars are s.e.m. across participants ($N = 22$). These analyses of RTs, which constitute a dataset independent of both fMRI activations and choice data used for model fitting validate the approach used to investigate fMRI. Indeed, our first regression analysis of fMRI activations assumes that decision-making is more demanding when the decision variable becomes more ambiguous, thereby predicting RTs to vary as a quadratic function of decision variable (i.e., with $-[(1-\omega)(v_1-v_2)+\omega(B_1-B_2)]^2$). Consistently, RTs decreased with quadratic regressor $[(1-\omega)(v_1-v_2)+\omega(B_1-B_2)]^2$ (**a**: $T(21)=6.6$; $p<0.0001$). Our second regression analysis of fMRI activations further assumes that reaching a decision requires less processing resources, when the evidence supporting actual choices increases. Thus, model MIX predicts RTs to exhibit two additive decreasing effects: when given normalised utilities, relative chosen affective beliefs ($B_{chosen}-B_{unchosen}$) increase and, when given affective beliefs, relative chosen normalised utilities ($v_{chosen}-v_{unchosen}$) increase. We therefore entered RTs in a multiple linear regression analysis including these two regressors and their interaction as within-subject factors. As predicted, RTs decreased when ($B_{chosen} - B_{unchosen}$) or, to a lesser extent, ($v_{chosen}-v_{unchosen}$) increased, with no interactions between these factors (**b, c**; $B_{chosen}-B_{unchosen}$: $T(21) = 4.9$, $p < 0.0001$; $v_{chosen}-v_{unchosen}$: $T(21) = 2.94$, $p = 0.0078$; interaction: $F < 1$; difference in linear effects: $T(21) = 2.5$, $p < 0.05$,). As in fMRI analyses, we then included quadratic regressors ($B_{chosen}-B_{unchosen}$)$^2$ and ($v_{chosen}-v_{unchosen}$)$^2$ as within-subject factors in this analysis. The quadratic regressors presumably capture the encoding of decision-relevant information irrespective of choice computations and, are consequently predicted to have no influences on RTs. The results confirmed the additive decreasing effects of linear regressors ($B_{chosen}-B_{unchosen}$) and ($v_{chosen}-v_{unchosen}$) ($T(21) = 5.2$, $p < 0.0001$ and $T(21) = 2.3$, $p = 0.03$, respectively) with no significant effects of quadratic regressors (both $Ts(21) < 1.16$, $ps > 0.26$). In summary, RTs confirmed that affective state beliefs and normalised utilities contributed to decision-making independently with affective state beliefs dominating normalised utilities

risk-aversion effects in human reward-based decision-making[17,29]. Additionally, previous studies involving various protocols[37–39] incidentally found that based on BIC measures, decision variables linearly combining reward values and probabilities fitted human/monkey behavioural data better than those computing expected utilities (as in model OPT). Although none of these studies control for standard distortion and/or belief formation models along with possible RL influences on choices, these previous incidental findings suggest that the proposed model is likely to extend to various decision situations.

This model has important conceptual implications. First, beliefs about reward contingencies bear upon affective states corresponding to which choice option most surely leads to rewards (i.e., the safest one) rather than arbitrary-defined abstract states. Second, affective state beliefs act in decision-making as appetitive values rather than as abstract probability measures: choices derive from the independent (additive) contribution of multiple normalised value components, including normalised utilities and affective state beliefs. Third, the notion of subjective probability appears as a theoretical construct reflecting the sub-optimal combination of beliefs and utilities in decision-making rather than distorted cognitive/neural representations of objective probabilities. Fourth, risk aversion appears to stem from the dominant contribution of affective state beliefs to decision-making and not only from the concavity of utility function as in the standard expected utility theory[29]. Whether the present findings and particularly, the notion of affective state beliefs generalise to beliefs bearing upon task dimensions other than reward probabilities associated with choice options (e.g., reward magnitude, stimulus or option identity) remains an open question for future investigation.

Using this more adequate model, we clarified the functional role of prefrontal regions in decision-making. Over and above reaction times, first, normalised utilities were associated with dmPFC activations, whereas reward expectations (proposed rewards) and reward history (RL-values) composing utilities were associated with vmPFC and laOFC activations, respectively. Thus, the dmPFC builds and normalises over choice options an overall utility value of each option based on reward history and expectations rather than represents reward expectations and history per se. The result subsumes in a single account separate previous findings indicating that in protocols inducing action values to derive either from reward expectations or from reward history, these unique components of action values are associated with dmPFC activations[32,40,41]. Second, evidence was that the dmPFC is involved in computing choices by linearly combining normalised utilities and affective state beliefs. dmPFC activations further varied with the unsigned difference between normalised utilities (or equivalently, with the maximal normalised utility, see Methods) irrespective of actual choices. This suggests that the dmPFC compares normalised utilities between choice options and combines the comparison result with affective state beliefs to guide the selection. Alternatively but not exclusively, this also suggests that the dmPFC modulates the engagement in the decision process according to the maximal utility at stake in each choice, as previously shown in rule-based decision-making[42]. Thus, the dmPFC appears to collect multiple reward-related variables (proposed rewards, RL-values, affective state beliefs) that concurrently guide behaviour rather than to integrate these variables into an overall expected value of control guiding behaviour[43].

Our results reveal that in contrast to normalised utilities, affective state beliefs are encoded in the vmPFC rather dmPFC, irrespective of choice computations: vmPFC activations increased with the quadratic expansion of (relative) chosen beliefs, i.e., with the unsigned difference between beliefs or the (relative) maximal belief among choice options. This finding supports the idea that the vmPFC is involved in representing latent, hidden states determining action outcome contingencies[7,8,44]. We further note that vmPFC activations exhibited no linear effects associated with

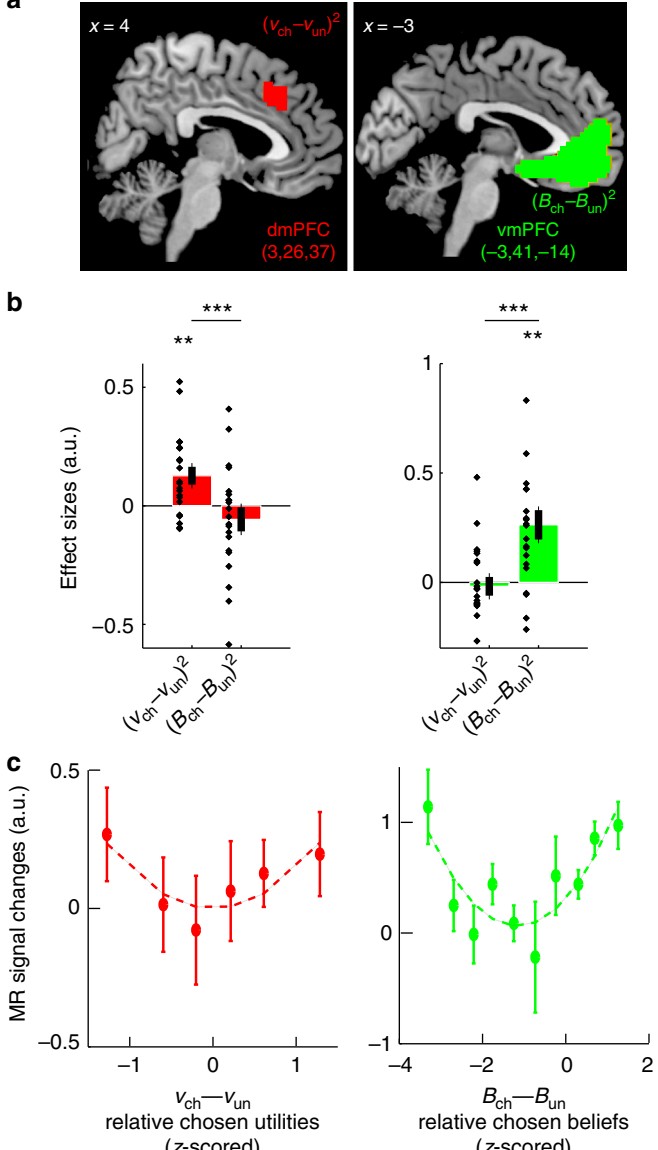

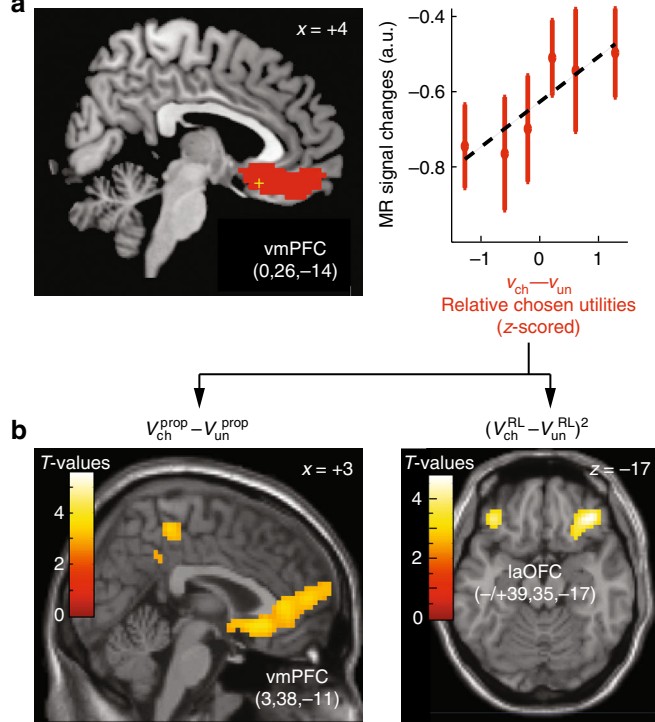

**Fig. 7** Prefrontal activations encoding normalised utilities and state beliefs. **a** Topography of fMRI activations associated with relative normalised utilities $(v_{chosen}-v_{unchosen})^2$ (red) and relative state beliefs $(B_{chosen}-B_{unchosen})^2$ (green) between bandits (Model MIX). Activations are thresholded at $p<0.05$ (T-tests, voxel-wise corrected for family-wise errors over the frontal lobes, MNI coordinate $Y>0$; cluster-wise threshold $p<0.05$) and superimposed on sagittal anatomical slices (MNI template). Numbers in brackets indicate MNI- coordinates of activation peaks (See Supplementary Table 4). **b** Quadratic effect sizes averaged over activation clusters shown in A (left: dmPFC; right: vmPFC; leave-one-out procedure removing selection biases). ** $p<0.01$; *** $p<0.001$ (T-tests). **c** Activations averaged over clusters shown in **a** and plotted against relative chosen normalised utilities ($v_{chosen}-v_{unchosen}$) (left, dmPFC, the linear component being factored out is shown in Fig. 5) and relative chosen beliefs ($B_{chosen}-B_{unchosen}$) (right, vmPFC, no linear component factored out) z-scored within participants. Error bars are s.e.m. across participants ($N=21$)

**Fig. 8** Prefrontal activations associated with value components of normalised utilities. a, left: topography of fMRI activations positively correlated with relative chosen normalised utilities $v_{ch}-v_{un}$ (red) from Model MIX (no other PFC activations exhibited this effect) and superimposed on sagittal anatomical slice $X=+4$ (MNI template, see Supplementary Table 4 for MNI-coordinates of activation peaks: yellow cross; results from the second regression analysis). a, right: MR signal changes averaged over the cluster shown on the left and plotted against relative chosen normalised utilities $v_{ch}-v_{un}$, z-scored within participants. Error bars are s.e.m. across participants ($N=21$). **b** When normalised utilities were broken down into its value components including proposed rewards $V^{prop}$ and reinforcement-Learning values $V^{RL}$ (third-regression analysis), the results showed that vmPFC activations actually correlated with relative chosen proposed rewards (left) and not RL-values (no other PFC activations exhibited this effect, except in a cluster located in the right dorsolateral cortex, BA 8). The only PFC activations associated with RL-values were located bilaterally in the orbitofrontal cortex (laOFC), which activations increased with the unsigned relative RL-values between bandits, irrespective of choices (positive quadratic effect). Activations were thresholded at $p<0.05$ (T-tests, voxel-wise corrected for family-wise errors over the frontal lobes: MNI coordinate $Y>0$; cluster-wise threshold $p<0.05$). Lighter colours indicate larger statistical T-values. Numbers in brackets indicate MNI-coordinates of activation peaks. See Supplementary Table 5 for MNI-coordinates of activation peaks

(relative) chosen beliefs. However, we found that consistent with previous studies using only linear regressors[5,12,45], vmPFC activations correlated with this linear regressor, only when the corresponding quadratic regressor was removed or equivalently,

projected onto this linear regressor. This operation arbitrary assigns the share variance between the two regressors to the linear one. This shared variance only stems from sampling biases due to participants' choices (choices more frequently matched the larger belief, otherwise the linear and quadratic regressor would be orthogonal). We then conclude that this linear correlation is likely to reflect a sampling artifact (see Supplementary Methods). Consistently, Fig. 7c shows that vmPFC activations varied as a pure quadratic function of (relative) chosen beliefs. Our results thus qualify previous findings and show that the vmPFC encodes the beliefs regarding which option most surely leads to rewards rather than the beliefs associated with the chosen option. Additionally, as the quadratic expansion of relative affective state

beliefs negatively scales with the risk to choose the less frequently rewarded option (Methods), our finding accords with previous results showing that vmPFC activations predict participants' confidence judgments[46,47].

vmPFC activations also increased with the value of chosen-relative-to-unchosen proposed rewards. This effect is consistent with previous studies[12,14,48,49] and the view that the vmPFC encodes the value of stimuli as potential action outcomes[50]. In the present protocol, however, this choice-dependent effect is unlikely to reflect decision processes computing choices, as choices derived predominantly from affective beliefs and normalised utilities rather than proposed rewards. Moreover, activations associated with the model decision variable were localised in the dmPFC rather than vmPFC. Previous results suggest that the vmPFC instead encode the appetitive value of attended relative to unattended stimuli[51]. We therefore interpret this choice-dependent effect associated with proposed rewards as reflecting the encoding of proposed rewards visually displayed on bandits modulated by participants' attention orienting towards the eventually chosen bandit. Supporting this interpretation, the frontal eyes field (superior lateral PFC, BA8) subserving visual attention[52,53] exhibited a similar effect.

By contrast, laOFC activations varied as the quadratic expansion of relative RL-values between bandits, indicating that laOFC encode the relative reward history of proposed bandits as cached variables, irrespective of choices. Consistently, the orbitofrontal cortex is involved in learning stimulus-reward associations[54–57]. Knowing that the functional segregation between laOFC and vmPFC has long been debated[50], the present results indicate that these regions functionally differ along at least two dimensions: (1) while the laOFC encodes the retrospective appetitive value, the vmPFC encodes the prospective appetitive value of available options; (2) the laOFC encodes values irrespective of actual choices, whereas the vmPFC encodes values with respect to actual choices. This segregation confirms that retrospective and prospective values are two functionally distinct components of option values.

In summary, the fMRI results provide evidence that the orbitofrontal cortex encodes three key components of appetitive values guiding decision-making. While laOFC encodes the reward history, the vmPFC encodes reward expectations of available options along with affective state beliefs regarding which option most surely leads to rewards. By contrast, the dmPFC including the dorsal anterior cingulate cortex is involved in computing choices based on these value components. The dmPFC translates the reward history and reward expectations into utility values normalised across choice options and commensurable to affective state beliefs. Normalised utilities and affective state beliefs then contribute to decision-making as two additive value components through the dmPFC. This additive contribution provides a mechanistic neural account of human choice suboptimality commonly theorised as deriving from irrational beliefs (subjective probabilities). Moreover, the dominant contribution of beliefs over utilities found here accounts for well-known risk aversion effects in human decision-making. Yet, a remaining issue is to identify the processes determining the relative contribution of utilities and beliefs to decision-making. This relative contribution might reflect an overall subjective scaling of outcome values and/ or the encoding strength of affective beliefs with respect to the instability (i.e., volatility) of external contingencies over time.

## Methods

**Participants.** We recruited 25 subjects (including 13 females and 12 males, aged 20–25 years and right-handed), who volunteered to participate to the study. Participants had no general medical, neurological, psychiatric or addictive history as assessed by medical examinations. Participants provided written informed consent,

and the present study was approved by the French National Ethics Committee (CPP, Inserm protocol #C07–28). Participants were paid for their participation. Each participant was tested in three MRI sessions (about 1.5 h each) administered in separate days. Three participants were excluded because they performed the task at chance level. An additional participant was excluded for fMRI analyses because of excessive head-motion. So the final sample included 22 individuals for behavioural analyses and 21 individuals for fMRI analyses.

**Behavioural protocol.** Participants were asked to make successive choices between two visually presented bandits, a square and a diamond bandit randomly presented on the left and right sides (Fig. 1a). One bandit led to rewards more frequently than the other one (frequency = 80% vs. 20%). This advantage reversed episodically and unpredictably (16, 20, 24 or 28 trials, pseudo-randomly). In every trial, each bandit proposed a potential monetary reward varying pseudo-randomly from 2 to 10 € (2, 4, 6, 8, 10 €, shown within each shape). These proposed rewards were drawn independently for each bandit. Both bandits were displayed until participants made their choice by pressing a hand-held response button (maximal duration 2.5 s). Participants chose the bandit on the left by pressing the left-handed button and vice versa. Following participants' choices, the chosen bandit outcome was visually revealed during 1 s with zero indicating no rewards. Otherwise, participants received the proposed reward approximately (the proposed reward with probability 0.5, the proposed reward plus 1 € with probability 0.25 and with probability 0.25, the proposed reward minus 1 €). The next trial then started after a delay. The time intervals between protocol events were uniformly jittered (from button presses to feedbacks onsets: range 0.1–4.1 s; from feedback offsets to next bandit onsets: range 0.5–4.5 s).

Participants performed the task in three experimental conditions administered in three separate sessions/days. (1) In the neutral condition, proposed rewards were independent of bandits' reward frequencies and were uniformly distributed (Fig. 1b). Thus, state beliefs could be inferred only from previous choice outcomes. (2) In the congruent condition, proposed rewards were biased towards higher values for the more frequently rewarded bandit according to an exponential-like distribution with slope $\gamma = 0.13$ (Fig. 1b, see computational modelling below). Conversely, proposed rewards were biased towards lower values for the less frequently rewarded bandit according to an exponential distribution with slope $-\gamma$. The exponential bias was chosen so that the information (in bits) conveyed by proposed rewards about bandit reward frequencies varied as a linear function of reward monetary values. (3) In the incongruent condition, proposed rewards were biased in the exact opposite direction ($\gamma = -0.13$ Fig. 1b). In both the congruent and incongruent conditions, thus, proposed rewards identically convey some additional information about state beliefs, i.e., bandits' reward frequencies. In both these conditions, state beliefs could be inferred from previous choice outcomes and proposed rewards.

Order of conditions/sessions was counterbalanced across participants. Each session comprised 400 trials and 19 reversals. Each session was broken down in four scanning runs separated by short breaks. Before each session, participants were only instructed that bandits had distinct reward frequencies and these frequencies reversed episodically. Additionally, participants were instructed that after each session, eight trials will be randomly drawn and the monetary rewards received during these trials will be added to their final pay-offs. Finally, participants were trained on the task before each session. The training protocol was identical to the experimental protocol, except that it includes only 50 trials and 2 reversals. Thus, participants could familiarise and learn bandits' and protocol contingencies. All experimental and training sessions were administered using the MATLAB PsychToolBox[58].

**Optimal model OPT.** We denote 1 and 2 the two bandits. Formally, the protocol consists of a sequence of hidden states $z_1, z_2, \ldots, z_t \in \{1, 2\}$, where $z_t = 1$ (=2, resp.) means that in trial $t$, bandit 1 (2, resp.) is the rewarded bandit with probability $q_M$ and bandit 2 (1, resp.) is the rewarded bandit with probability $q_m$ ($q_M$ and $q_m$ equal to 80% and 20%, respectively)·[27]. We denote $V_1(t)$ and $V_2(t)$ the proposed rewards (monetary values) in trial $t$ associated with bandit 1 and 2, respectively. In this protocol, the proposed rewards are drawn from distributions $p(V_1(t)|z_t)$ and $p(V_2(t)|z_t)$:

$$p(V_1(t)|z_t) \propto \begin{bmatrix} \exp(\gamma(V_1(t) - \bar{v})) & \text{if } z_t = 1, \\ \exp(-\gamma(V_1(t) - \bar{v})) & \text{otherwise,} \end{bmatrix}$$

$$p(V_2(t)|z_t) \propto \begin{bmatrix} \exp(\gamma(V_2(t) - \bar{v})) & \text{if } z_t = 2, \\ \exp(-\gamma(V_2(t) - \bar{v})) & \text{otherwise,} \end{bmatrix}$$

where, $\bar{v}$ denotes the mean reward (because only ratios between bandits are meaningful, quantity $\bar{v}$ has no influence on choices: see below). Parameter $\gamma$ is a constant: (i) $\gamma = 0$ corresponds to the neutral condition (proposed rewards are uninformative about hidden states); (ii) $\gamma > 0$ corresponds to the congruent condition (proposed rewards are biased towards higher values for the most frequently rewarded bandit and vice-versa); (iii) $\gamma < 0$ corresponds to the incongruent condition (proposed rewards are biased towards lower values for the most frequently rewarded bandit and vice versa).

In each trial $t$, the model chooses between two actions, $a_t = 1$ (choosing bandit 1) or $a_t = 2$ (choosing bandit 2). Then, the model receives a feedback $x_t \in \{0, 1\}$ indicating whether the proposed reward is obtained ($x_t = 1$) or not ($x_t = 0$). The model has beliefs $B_z(t)$ at trial $t$ onset about hidden states $z_t$, given all past observations. The model then observes proposed rewards $r = \{V_1(t), V_2(t)\}$ in trial $t$, and updates her/his beliefs $B_Z(t|r)$, which write as follows:

$$B_1(t|r) \propto e^{\gamma(V_1(t) - V_2(t))} B_1(t)$$

$$B_2(t|r) \propto e^{\gamma(V_2(t) - V_1(t))} B_2(t).$$

Normalization terms (so that $B_1(t|r) + B_2(t|r) = 1$) are omitted for simplicity. The model then computes reward probabilities associated with each bandit by marginalising over beliefs $B_Z(t|r)$. Accordingly, inferred reward probabilities associated with choosing bandit 1 (bandit 2), denoted $P_1(t)$, ($P_2(t)$, resp.) write as follows:

$$P_1(t) = q_M B_1(t|r) + q_m B_2(t|r)$$

$$P_2(t) = q_m B_1(t|r) + q_M B_2(t|r)$$

The model then computes bandits' expected utilities EU by integrating reward probabilities and values:

$$EU(a_t = 1) = V_1(t) P_1(t)$$

$$EU(a_t = 2) = V_2(t) P_2(t)$$

The model selects the bandit (soft)maximising expected utility: probabilities $p_1$ and $p_2$ of choosing bandit 1 and 2, respectively, write as the following softmax function:

$$p_1 = (1 - \varepsilon) \text{sigmoid} \beta(V_1(t) P_1(t) - V_2(t) P_2(t)) + \frac{\varepsilon}{2}$$

$$p_2 = 1 - p_1$$

where $\beta$ is the inverse temperature and $\varepsilon$ the lapse rate. The larger $\beta$ and the lower $\varepsilon$ the more optimal is the choice. After choosing a bandit, the model observes feedback $x_t$ and updates again her/is beliefs $B_Z(t|r, x)$ as follows:

$$B_1(t|r, x) \propto q_M^{I(t)} q_m^{1 - I(t)} B_1(t|r)$$

$$B_2(t|r, x) \propto q_M^{1 - I(t)} q_m^{I(t)} B_2(t|r),$$

where exponent $I(t) = x_t(2 - a_t) + (1 - x_t)(a_t - 1)$ is equal to 0 or 1 depending upon the chosen bandit ($a_t = 1$ or 2) and its outcome ($x_t = 0$ or 1). Finally, beliefs formed in trial $t$ serve to form beliefs in next trial $t+1$ according to the volatility of the environment:

$$B_1(t + 1) = (1 - \nu) B_1(t|r, x) + \nu B_2(t|r, x)$$

$$B_2(t + 1) = \nu B_1(t|r, x) + (1 - \nu) B_2(t|r, x)$$

where $\nu$ represents the volatility of the environment, i.e., the probability that a reversal occurs between two successive trials $(z_{t+1} \neq z_t)$[11]. Because episode lengths are pseudo-randomized within sessions, volatility $\nu$ is assumed to be a constant. In trials with no responses, beliefs were updated based on proposed rewards and volatility only.

In summary, beliefs evolve from trial to trial as follows:

$$\binom{B_1(t+1)}{B_2(t+1)} \propto \overbrace{\begin{pmatrix} 1 - \nu & \nu \\ \nu & 1 - \nu \end{pmatrix} \underbrace{\begin{pmatrix} q_M^{I(t)} q_m^{1-I(t)} & 0 \\ 0 & q_M^{1-I(t)} q_m^{I(t)} \end{pmatrix} \overbrace{\begin{pmatrix} e^{\gamma(V_1(t) - V_2(t))} & 0 \\ 0 & e^{\gamma(V_2(t) - V_1(t))} \end{pmatrix} \binom{B_1(t)}{B_2(t)}}^{B(t|r)}}_{B(t|r,x)}}^{B(t|r,x)},$$

inferred reward probabilities write as follows:

$$\binom{P_1(t)}{P_2(t)} = \begin{pmatrix} q_M & q_m \\ q_m & q_M \end{pmatrix} \binom{B_1(t|r)}{B_2(t|r)},$$

and choice probabilities as the following softmax function:

$$p_i = (1 - \varepsilon) \text{sigmoid} \beta(V_i P_i(t) - V_j P_j(t)) + \frac{\varepsilon}{2}, \ i = 1, 2; j = 3 - i.$$

Finally, model OPT predicts log-odds of choice frequencies to vary as follows

(assuming lapse rate $\varepsilon \ll 1$):

$$\log \frac{p_1}{p_2} \simeq \beta(V_1(t) P_1(t) - V_2(t) P_2(t))$$

$$= \frac{\beta}{2} [(P_1(t) + P_2(t))(V_1(t) - V_2(t)) + (V_1(t) + V_2(t))(P_1(t) - P_2(t))] \quad (4)$$

$$= \frac{\beta}{2} [(q_M + q_m)(V_1(t) - V_2(t)) + (V_1(t) + V_2(t))(P_1(t) - P_2(t))]$$

The third equation line is obtained using $P_1(t) + P_2(t) = q_M + q_m$. In the present protocol featuring constant bandits' reward frequencies $q_M$, $q_m$, choice frequency log-odds thus vary as the addition of relative reward values to the interaction between total reward values and relative inferred probabilities. Note that if $V_1(t), V_2(t)$ were normalised ($V_1(t) + V_2(t) = \text{constant}$), log-odds would vary as a linear combination of relative reward values and probabilities. Note finally that reward frequencies $(q_M, q_m)$ have a quadratic influence on how log-odds vary across two successive trials as a result of updating and marginalising over beliefs.

In the present protocol, quantities $q_M, q_m, \gamma, \nu$ are constant (true values: $q_M = 0.8; q_m = 0.2, \gamma_{\text{neutral}} = 0, \gamma_{\text{congruent}} = 0.13, \gamma_{\text{incongruent}} = -0.13, \nu = 0.05$). Consistent with the training phase preceding each experimental session, a more sophisticated model that further learns/infers these constants did not improve the fit of participants' data significantly. In the main text, model OPT conceptually refers to this optimal model with $q_M, q_m, \gamma_n, \gamma_c, \gamma_i, \nu$ set at their true values and the softmax set as hardmax ($\beta \to +\infty$ and $\varepsilon = 0$). When fitted to data, however, we treated $q_M, q_m = 1 - q_M, \gamma_n, \gamma_c, \gamma_i, \nu$ as free parameters along with softmax free parameters $\beta$ and $\varepsilon$. We refer to this parameterised version of model OPT as parameterised model OPT.

**Distortion model DIST.** Model DIST assumes inferred reward probabilities of bandits $P_1(t)$ and $P_2(t)$ are distorted as reflecting a misrepresentation of objective probabilities (subjective probabilities). As these reward probabilities are linear combinations of state beliefs $B_1(t|r)$ and $B_2(t|r)$, one may equivalently assume that distortions apply to beliefs.

We used the standard distortion function encompassing previously documented distortions (convex, concave, S-shaped and inverted S-shaped)[21], which writes as follows:

$$\log \frac{\tilde{P}}{1 - \tilde{P}} = \eta \log \frac{P}{1 - P} + (1 - \eta) \log \frac{P_0}{1 - P_0}$$

where $\tilde{P}$ denotes subjective probabilities and $P$ objective probabilities from model OPT. $\eta > 0$ and $P_0 (0 < P_0 < 1)$ are two parameters specifying distortions: $\eta = 1$ means no distortions. When $\eta > 1$, the subjective probability is an S-shaped function of objective probability. When $\eta < 1$, the subjective probability is an inverted S-shaped function of objective probability. In both cases, the equivalence point where subjective and objective probability are identical is $P_0$: when $P_0 \to 0$ or 1, the distortion function becomes simply convex or concave.

For completeness, we also used this distortion function (with distinct parameters) for monetary values of proposed rewards $V$ to account for the possibility that subjective utility (or appetitive values) of proposed rewards is not a linear function of monetary values (monetary values were rescaled for varying from 0 to 1). Accordingly, model DIST is identical to model OPT except that the computation of expected utilities is now based on distorted probabilities and distorted monetary values:

$$EU(a_t = 1) = \tilde{V}_1(t) \tilde{P}_1(t)$$

$$EU(a_t = 2) = \tilde{V}_2(t) \tilde{P}_2(t)$$

Compared with parameterised model OPT, model DIST comprises four additional free parameters: $\eta$ and $P_0$ for probabilities and values. Note that like parameterised model OPT, all parameters in model DIST were constant across the neutral, congruent and incongruent conditions, except proposed reward biases $\gamma$, of course.

Finally, we can deduce from choice frequency log-odds of model OPT (see above) that model DIST predicts log-odds of choice frequencies to vary as follows (assuming lapse rate $\varepsilon \ll 1$):

$$\log \frac{p_1}{p_2} \simeq \frac{\beta}{2} \left[ (\tilde{P}_1(t) + \tilde{P}_2(t))(\tilde{V}_1(t) - \tilde{V}_2(t)) + (\tilde{V}_1(t) + \tilde{V}_2(t))(\tilde{P}_1(t) - \tilde{P}_2(t)) \right]$$

**Reduced models from model DIST.** To validate the addition of distortion parameters to model OPT, we considered the three reduced models nested in model DIST: (1) one removing distortions on reward values; (2) one removing distortions on reward probabilities; (3) and one corresponding to parameterised model OPT, i.e., with no distortions on both values and probabilities.

**Hybrid model DIST+RL.** We also considered the extension of model OPT comprising an additional reinforcement-learning (RL) component altering subjective expected utilities. In this model, choice probabilities $p_i$ write as the following softmax function:

$$p_i = (1 - \varepsilon)\text{sigmoid}\beta\left(\varphi\left(\bar{V}_i(t)\bar{P}_i(t) - \bar{V}_j(t)\bar{P}_j(t)\right) + (1 - \varphi)(V_i^{\text{RL}}(t) - V_j^{\text{RL}}(t))\right) + \frac{\varepsilon}{2}$$

with $\varphi$ a weight parameter ($i = 1,2$; $j = 3-i$). $V_i^{\text{RL}}(t)$ denotes the RL value for bandit $i$ in trial $t$, deriving from the standard Rescorla & Wagner learning rule[28]:

$$V_{\text{ch}}^{\text{RL}}(t) = V_{\text{ch}}^{\text{RL}}(t - 1) + \alpha\left(R_{t-1} - V_{\text{ch}}^{\text{RL}}(t - 1)\right)$$

where $a$ is the learning rate, $R_{t-1}$ the actual reward received in trial $t$-1. Index ch indicates that the rule applies to the chosen bandit only. In trials with no responses, RL-values remained unchanged.

**Alternative model MIX.** Model MIX assumes no distortions. Hidden states $z_1, z_2, \ldots, z_t \in \{1, 2\}$ now represent affective states: $z_t = 1$ ($= 2$, resp.) means that in trial $t$, bandit 1 (2, resp.) is the most frequently rewarded bandit (corresponding to probability $q_M$ and bandit 2 (1, resp.) is the least rewarded bandit (corresponding to probability $q_m$). Model MIX assumes that choices derive from the additive contribution of normalised utilities and affective state beliefs. Consequently, choice probabilities $p_1$ and $p_2$ of choosing bandit 1 and 2, respectively, varies as a softmax function of the sum of normalised utilities $v_i(t)$ and state beliefs $B_i(t|r)$ as indicated below:

$$p_1 = (1 - \varepsilon)\text{sigmoid}\beta[(1 - \omega)(v_1(t) - v_2(t)) + \omega(B_1(t|r) - B_2(t|r))] + \frac{\varepsilon}{2}$$

$$p_2 = 1 - p_1,$$

where $\omega$ is a weighting parameter. As previously proposed[25,26], utilities $v_1$, $v_2$ of choice options corresponds to proposed rewards altered by the reward history and normalised across the two choice options:

$$v_i(t) = \frac{\varphi V_i^{\text{pr}}(t) + (1 - \varphi)V_i^{\text{RL}}(t)}{\sum_{i=1,2} \varphi V_i^{\text{pr}}(t) + (1 - \varphi)V_i^{\text{RL}}(t)}, \text{for } i = 1, 2.$$

In this expression, the first term $V_i^{\text{pr}}(t)$ represents the proposed monetary reward, while the second term $V_i^{\text{RL}}(t)$ represents the RL value for bandit $i$ in trial $t$. Constant $\varphi$ is simply a weighting parameter. Note that with no loss of generality, utilities are normalised to 1 (i.e. $v_1(t) + v_2(t) = 1$). Reinforcement values $V_i^{\text{RL}}(t)$ derive from the standard Rescorla & Wagner learning rule as in hybrid model DIST +RL (see above)[28].

Model MIX assumes that state beliefs are inferred as in model OPT. Compared with model parameterised OPT, consequently, model MIX comprises three additional free parameters: weighting constants $\omega$, $\varphi$, and learning rate $\alpha$. Note again that as in model OPT and DIST, all parameters in model MIX were constant across the neutral, congruent and incongruent conditions (except proposed-reward biases $\gamma$, of course). We further verified that, using Bayesian Information Criteria, fitting distinct weight parameters $\omega$, $\varphi$ across the three conditions led to no improvements in fitting experimental data.

Finally, model MIX predicts log-odds of choice frequencies to vary as the additive effects of normalised utilities and affective beliefs (assuming lapse rate $\varepsilon \ll 1$) (see Fig. 2c):

$$\log\frac{p_1}{p_2} \simeq \beta[(1 - \omega)(v_1 - v_2) + \omega(B_1(t|r) - B_2(t|r))]. \tag{5}$$

Note that in model MIX, reward frequencies ($q_M$, $q_m$) have a linear influence on how log-odds vary across two successive trials as a result of beliefs' updating only (see mathematical derivations for model OPT).

**Special case $\omega = \omega_0$.** Model MIX further features a special case when $\omega = \omega_0 = \frac{q_M - q_m}{q_M + q_m + q_M - q_m}$. In that case, model MIX is equivalent to model OPT with normalised utilities and affective state beliefs, and vice versa. Indeed, consider first choice log-odds from model OPT with normalised utilities, which write as follow (see Eq. (4) above):

$$\log\frac{p_1}{p_2} \simeq \frac{\beta}{2}[(P_1(t) + P_2(t))(v_1 - v_2) + (v_1 + v_2)(P_1(t) - P_2(t))] \tag{6}$$

where $P_1(t)$, $P_2(t)$ are inferred reward probabilities deriving from marginalizing over affective state beliefs $B_1(t|r)$, $B_2(t|r)$:

$$P_1(t) = q_M B_1(t|r) + q_m B_2(t|r)$$

$$P_2(t) = q_m B_1(t|r) + q_M B_2(t|r)$$

Note that $B_1(t|r) + B_2(t|r) = 1$ and consequently,

$$P_1(t) + P_2(t) = q_M + q_m$$

$$P_1(t) - P_2(t) = (q_M - q_m)(B_1(t|r) - B_2(t|r))$$

As $v_1 + v_2 = 1$, Eq. (6) is therefore equivalent to:

$$\log\frac{p_1}{p_2} \simeq \frac{\beta}{2}[(q_M + q_m)(v_1 - v_2) + (q_M - q_m)(B_1(t|r) - B_2(t|r))]$$

$$= \frac{\beta}{2}[(q_M + q_m) + (q_M - q_m)]\left[\frac{(q_M + q_m)}{(q_M + q_m) + (q_M - q_m)}(v_1 - v_2) + \frac{(q_M - q_m)}{(q_M + q_m) + (q_M - q_m)}(B_1(t|r) - B_2(t|r))\right]$$

$$= \beta q_M\left[\left(1 - \frac{(q_M - q_m)}{(q_M + q_m) + (q_M - q_m)}\right)(v_1 - v_2) + \frac{(q_M - q_m)}{(q_M + q_m) + (q_M - q_m)}(B_1(t|r) - B_2(t|r))\right]$$

Consequently, Eq. (6) writes as follow:

$$\log\frac{p_1}{p_2} \simeq \beta q_M[(1 - \omega_0)(v_1 - v_2) + \omega_0(B_1(t|r) - B_2(t|r))] \tag{7}$$

with $\omega_0 = \frac{(q_m - q_m)}{(q_M + q_m) + (q_M - q_m)}$. Eq. (7) is identical to Eq. (5) describing choice log-odds from model MIX with $\omega = \omega_0$ (with inverse temperature $\beta q_M$). Thus, model OPT with normalised utilities and affective state beliefs is equivalent to model MIX with $\omega = \omega_0$, and vice versa: maximising a weighted sum of normalised utilities and affective state beliefs with weight parameter $\omega_0$ is equivalent to maximising the expected values of normalised utilities.

**Reduced models from model MIX.** To validate the mixture comprising state beliefs, proposed rewards and RL values, we considered the six reduced models nested in model MIX: (1) one comprising only state beliefs ($\omega = 1$); (2) one comprising no state beliefs ($\omega = 0$); (3) one comprising only proposed rewards ($\omega = 0$; $\varphi = 1$); (4) one comprising no proposed rewards ($\varphi = 0$), i.e., comprising only RL-values and beliefs; (5) one comprising only RL values (pure RL); (6) and finally, one comprising no RL values ($\varphi = 1$), i.e., comprising only proposed rewards and beliefs.

**Relative affective state beliefs and confidence in choosing the more frequently rewarded option.** Lebreton et al.[47] show that subjects' confidence in their choices, which correlates with vmPFC activations, negatively scales with the variance of the distribution coding for the variable guiding choices. Here, we show that the unsigned difference between state beliefs, which correlated with vmPFC activations, matches this notion of confidence: namely, the quadratic expansion of relative affective state beliefs negatively scales with the variance between effective reward probabilities associated with bandits. Consider bandit 1 and its effective reward probability $P_1$, which is equal to $q_M$ with probability $B_1$ and $q_m$ with probability $B_2$. The expected value of $P_1$ denoted $E(P_1)$ is simply equal to $E(P_1) = q_M B_1 + q_m B_2$. We can therefore write $E(P_1 - P_2) = (q_M - q_m)(B_1 - B_2)$, where index 2 refers to bandit 2. The variance of $P_1$–$P_2$ then writes as follow:

$$\text{Var}(P_1 - P_2) = E(P_1 - P_2)^2 - (E(P_1 - P_2))^2$$

$$= (q_M - q_m)^2 B_1 + (q_m - q_M)^2 B_2 - (q_M - q_m)^2(B_1 - B_2)^2 = (q_M - q_m)^2\left[1 - (B_1 - B_2)^2\right]$$

Thus, the risk to choose the less frequently reward bandit, which varies with $\text{Var}(P_1 - P_2)$,
negatively scales with the quadratic expansion of relative state beliefs.

**Model fitting and comparison.** For adjusting model-free parameters, all models were fitted to experimental data separately for each participant by maximising the model log-likelihood (LLH):

$$\text{LLH} = \sum_t \log p_{\text{ch}}(t)$$

where $p_{\text{ch}}(t)$ is the model choice probability of participant's choice ch in trial $t$, given previous participant's choices. The three conditions were fitted together, with all parameters constant across conditions, except naturally reward distribution slope $\gamma = \gamma_n, \gamma_c, \gamma_i$. See Supplementary Tables 1 and 2 for best-fitting parameters in model DIST and MIX, respectively. To maximise LLH, we used a slice sampling procedure with uniform priors over large parameter ranges[59], which is computationally costly but appropriate for high-dimensional parameters spaces as providing estimates of each parameter posterior distribution. We ensured that samples were independent enough so that parameters' estimates were reliable. Finally, we carried out a gradient ascent starting from the best sample to get optimised estimates of parameters maximising LLH. We compared model fits by computing

Bayesian Information Criteria for each model and making pairwise $T$-tests with participants treated as random factors (Supplementary Table 6). BICs penalise for model complexity based on the number of free parameters[59].

**Model recovery procedure**. We also verified the ability of the present protocol and the validity of our fitting procedure to discriminate between model DIST and MIX by using a model recovery procedure. We generated synthetic data by simulating the performance of models DIST and MIX, using parameters fitted on every participant for these simulations. Then we fitted models DIST and MIX on these synthetic data as if they were participants' data, using the exact same fitting procedure described above. We found that as expected, model DIST fits its own performance data better than model MIX and vice-versa (LLH: both $Ts(21) > 4.98$, $ps < 0.00002$).

**fMRI data acquisition and processing**. fMRI volumes were acquired on a 3T Siemens Trio at the Centre de Neuroimagerie de Recherche (CENIR) in hospital La Pitié Salpêtrière, Paris, France. Functional images were acquired with parameters TR = 2 s, TE = 25 ms, Nb of repetitions/run = 431, Nb slices = 39, thickness = 2 mm, flip angle 75°, voxel size = 2.5 mm³. Before the first trial, two TR of baseline recording were acquired for subsequent slice-timing correction. Echo planar images were 30° tilted to minimise signal drop around the orbitofrontal cortex[60]. Stimuli were projected on a mirror settled on a 32-channels head coil. T1 anatomical images were also acquired before functional acquisitions. Each experimental condition (neutral, congruent and incongruent) was recorded in separate days and comprised four scanning runs.

MRI data were processed and analysed using SPM8 software package (http://www.fil.ion.ucl.ac.uk) using standard slice-timing, spatial realignment, normalisation to Montreal Neurological Institute echo planar imaging template (images resampled at 3 mm³) and Gaussian spatial smoothing (isotropic 6-mm kernel). Temporal correlations were estimated using restricted maximum likelihood estimates of variance components using a first-order autoregressive model. The resulting non-sphericity was used to form maximum likelihood estimates of the activations. Only head movements below 3–5 mm or 3–5° were accepted, thereby excluding one participant.

**fMRI statistical analyses**. Statistical parametric maps of local brain activations were computed in every subject using the standard general linear model (GLM). The model included separate event-related regressors, which convolved a series of delta function with the canonical haemodynamic response function (HRF) that estimated BOLD responses at stimulus and feedback onsets. Regressors of no interest included trials with no responses (0.6% of all trials), six motion parameters from the realignment procedure, along with regressors modelling each run. The three conditions were analysed with a unique GLM. Event-related regressors of interest were parametrically modulated by variables derived from model MIX fitted on behavioural data. Parametric modulations at stimulus onsets included linear and quadratic expansions of relative state beliefs and normalised utilities derived from model MIX. All parametric modulations were z-scored to ensure between-subjects and between-regressors comparability of regression coefficients (quadratic regressors were z-scored after applying the quadratic transformation). We ensured that Variance Inflation Factor assessing collinearity between all parametric modulators were small enough[61] to allow proper dissociations. Thus, shared variances (coefficient of determination $R^2$) between state beliefs and normalised utilities were equal to 0.17 and 0.02 for linear and quadratic expansions, respectively. Importantly, GLMs were performed in complete full variance with no orthogonalisation (by deactivating the SPM default orthogonalisation option), so that all shared variance across regressors were placed in residuals and observed activations were specific to each parametric modulation (Supplementary Methods). We systematically performed control analyses for reaction times: in all GLMs, we further included along with regressors of interest an additional parametric regressor factoring out reaction times. All results reported in the main text remained unchanged when including this additional regressor.

Following the standard SPM method, second-level parametric maps were then obtained for each contrast over the group of participants. Activated voxels were identified using a significance voxel-wise threshold set at $p < 0.05$ ($T$-tests) corrected for family-wise errors for multiple comparison over the search volumes (see below). Statistical inferences were based on cluster level: significant activations were identified as clusters of activated voxels with significance cluster-wise threshold set at $p < 0.05$. We removed selection biases from all post hoc analyses performed from activation clusters using a leave-one-out procedure[34]: for every GLM, the partial correlation coefficients (betas) of each participant were averaged over activation clusters identified in the N-1 remaining participants (using the significance thresholds indicated above); these coefficients were then entered in post hoc analyses across the sample of N participants. Three regression analyses were carried out:

<u>1-GLM</u>: Decision entropy $-[(1 - \omega)(v_1 - v_2) + \omega(B_1 - B_2)]^2$ between bandits 1 and 2 was the unique parametric modulation at stimulus (bandit) onsets. Parametric modulations at feedback onsets included relative chosen beliefs $B_{ch} - B_{un}$, RL-values $V_{ch}^{RL}$ and feedback values, where indexes ch and un refer to

the chosen and unchosen bandit in trial $t$, respectively. This analysis was performed over the whole brain. Given this search volume and the resulting FWE-correction over voxel-wise $T$-values, the cluster-wise threshold set at $p < 0.05$ then corresponded to a cluster size $k > 10$ voxels (0.27 cm³).

To identify regions involved in choice computations, we used the quadratic rather than modulus operator of decision variable (i.e. $[(1 - \omega)(v_1 - v_2) + \omega(B_1 - B_2)]^2$) based on the following rationale. Previous results show fMRI activity to vary with the area under evidence accumulation traces in sequential sampling models of decision-making[33]. As these traces form on average linear increases which slope scales with the decision variable, the area under these traces then vary as an inverted quadratic function of the decision variable. Anyway, using a modulus operator provided virtually identical results (Supplementary Fig. 5a).

<u>2-GLM</u>: parametric modulations at stimulus (bandit) onsets included relative chosen beliefs $B_{ch} - B_{un}$, relative chosen utilities $v_{ch} - v_{un}$, their interaction $(B_{ch} - B_{un})(v_{ch} - v_{un})$ and their quadratic expansion $(B_{ch} - B_{un})^2$ and $(v_{ch} - v_{un})^2$. Parametric modulations at feedback onsets were as in GLM#1 above. This analysis was performed over the frontal lobes only (MNI coordinate $Y > 0$). Given this search volume and the resulting FWE-correction over voxel-wise $T$-values, the cluster-wise threshold set at $p < 0.05$ then corresponded to a cluster size $k > 40$ voxels (1.08 cm³).

Note that as utilities and beliefs are normalised variables, one may indifferently use linear regressors $(B_{ch} - B_{un})$, $(v_{ch} - v_{un})$ or $B_{ch}$, $v_{ch}$. Indeed, $B_{chosen} = \frac{1}{2}(1 + B_{chosen} - B_{unchosen})$ and $v_{chosen} = \frac{1}{2}(1 + v_{chosen} - v_{unchosen})$. Similarly, quadratic expansions $(B_{ch} - B_{un})^2$ and $(v_{ch} - v_{un})^2$ modelling coding information or negative entropy associated with beliefs and utilities identically reflect the encoding of maximal beliefs $B_{max}$ and utilities $v_{max}$ or relative maximal beliefs $(B_{max} - B_{min})$ and utilities $(v_{max} - v_{min})$. Indeed, $X_{max} = \frac{1}{2}(1 + |X_1 - X_2|)$ and $X_{min} = \frac{1}{2}(1 - |X_1 - X_2|)$ when $X_1 + X_2 = 1$. In any case, activations correlating with these quadratic regressors convey belief and utility information of one bandit compared with the other bandit, irrespective of chosen bandits. We used quadratic expansions as they better approximate entropy function than modulus operators (unsigned differences between beliefs and between utilities). Using modulus operators provided virtually identical results (Supplementary Fig. 5b). See additional details in Supplementary Methods.

<u>3-GLM</u>: parametric modulations at stimulus (bandits) onsets included relative chosen beliefs $B_{ch} - B_{un}$, relative chosen proposed rewards $V_{ch}^{pr} - V_{un}^{pr}$, relative chosen RL-values $V_{ch}^{RL} - V_{un}^{RL}$ and their quadratic expansion $(B_{ch} - B_{un})^2$, $(V_{ch}^{pr} - V_{un}^{pr})^2$ and $(V_{ch}^{RL} - V_{un}^{RL})^2$. Parametric modulations at feedback onsets were as in GLM#1 above. As in GLM#2, this analysis was performed over the frontal lobes only (MNI coordinate $Y > 0$). Consequently, the cluster-wise threshold set at $p < 0.05$ again corresponded to a cluster size $k > 40$ voxels (1.08 cm³).

**Reporting summary**. Further information on experimental design is available in the Nature Research Reporting Summary linked to this article.

## Data availability
fMRI data are available at the NeuroVault data repository: neurovault.org, collection ID: MIX. URL: neurovault.org/collections/YFISTKGF. Behavioral data are available from the authors upon request. A reporting summary for this article is available as a Supplementary Information file.

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

## Acknowledgements

We thank Stefano Palminteri and Valentin Wyart for helpful comments and Anne-Dominique Lodeho-Devauchelle for technical assistance. Supported by a European Research Council Grant (ERC-2009-AdG #250106) to E.K.

## Author contributions

M.R., J.D. and E.K. designed the study. M.R., J.D. and E.K. built the computational models. M.R. ran the experiment. M.R. and E.K. analysed the data. M.R. and E.K. wrote the paper.

## Additional information

**Competing interests:** The authors declare no competing interests.

