## [Peer Review File · Nature Communications]

Reviewers' comments:

Reviewer #1 (Remarks to the Author):

The Study by Rouault et al investigates human decision making in an environment with outcome probability reversals and outcome magnitude fluctuations. The authors test a series of models in which state beliefs and utilities influence decisions. Rouault et al propose that choice results from the additive contributions of state beliefs and utility values, in contrast to the expected utility theory that integrates utilities over state beliefs to derive subjective probabilities for choice computation. They test the idea in a bandit task by fitting participants' behavior to those contrasting models. Using model comparison and falsification approaches the manuscript presents evidence in support of the hypothesis that beliefs and utilities have independent effects. Additionally, they demonstrate with fMRI that normalized utility value correlates with dmPFC activation, while its two additive components (proposed rewards and RL-values) correlate with activations in vmPFC and laOFC respectively. The study is well written and the analysis is thorough.

1. In the presented study the strength of the beliefs is confounded with the outcome probabilities. This might be a factor contributing to the fact that the MIX model performs well. If the state uncertainty would for instance come from perceptual uncertainty, and would be unrelated to the outcome probabilities, the best fitting model might be quite different. I would like to see evidence that the reported effects regarding belief states cannot be reframed as learned/updated outcome probabilities

2. I find the modelling as it is already quite extensive. But it is unclear how the work relates to RL models that have been shown to influence behaviour and brain activation in a number of studies. The core issue is that the RL models are specified in an unusual way.

2A) The unusual part is that the RL value is assumed to not take into account the proposed value, which is not necessarily what a RL model would do (this pertains both to the DIST and the MIX models). From an RL perspective, a diamond with a 6 inside should be treated as a different bandit than a diamond with a 3 inside (the numbers are relevant for predicting the outcomes, so they should be part of the state/bandit). In other words, the RL component is particularly simplistic, and it remains unclear to me if the additional contribution of the proposed reward in the MIX model is due to the oversimplified RL model or reflects a true non-RL influence on participants' choices.

2B) I am confused by the double influence of the outcome probabilities. It is stated that the beliefs are updated according to $B(t|r,x) \sim q_M^{I(t)} q_m^{(1-I(t))} B(t|r)$. Later the reward probabilities are computed again involving q_m/M : $P(t) = (q_m q_M) * B(t|r)$. I might be misunderstanding, but this seems to imply that the probabilities have a quadratic influence on choices?

2C) more generally, it would be important to know how models that incorporate either state uncertainty (belief state models, see e.g. Dayan & Daw 2008, *Cognitive, Affective, & Behavioral Neurosci*, 8, 429-453.) or belief updating (e.g. Nassar et al, 2010, *J Neurosci* 30; Hang et al., 2015, *J Neurosci.*, 35) would perform in this task (I am not an author on any of the listed papers).

3. The introduction reads very well, but results were a bit hard to follow because the models are introduced piecemeal in between behavioral effects. I would suggest to introduce the models in a separate section that emphasises which hypotheses they reflect. In this regard it would also be good to avoid using mathematical notation that overlaps with, but has a different meaning in, other models such as using 'Q' for choice history, although Q is quite established notation for expected state-action value in RL models.

Additional points:

4. I disagree with the statement about exploration in the introduction: isn't exploration still necessary, since outcome probabilities are changing? E seems more about dynamic/changing environments, rather than how much info one can get in each trial
5. Fig 1C: I would be good to see data until trial 28. Is there evidence that discrete reversal structure of occurrence at either 16/20/24/28 is picked up upon? Relatedly: Can our show how belief states develop as a function of trial history after the last reversal? What is the step size in belief updating?
6. It would be good to include information about the response deadline and the trial timing directly into figure 1
7. It would be good to have a table of BIC values for all the models, along with AIC
8. It might be worth referring to the fact that participants exhibit the well known effect of probability matching
9. How does the modeling account for trials with no responses?
10. What t test is done when conditions are compared: mean reward or only the last trials before a reversal? Mean over all trials, or only 1-16 as shown in the figure?
11. In Fig 2A, say what green is. In Fig 2C, it would be useful to have an identity line in the left panel too
12. Fig 2A: please overlay the participants' data for visual comparison.
13. Fig 2D would be easier to read if data is displayed using bar graphs. The current x axes ("-0.5" despite the categorical distinctions) do not make sense.
14. What does model OPT predict regarding the interaction of V and P? Is it really true that all reduced models of DIST are generally ruled out by its failure to capture some aspects of the data? I understand the logic, but what about for instance overfitting?
15. I think the RL part in DIST+RL could be explained better in the text
16. Line 188: I think the p value indicates that there is no evidence against the null that DIST == DIST+RL, not that it is "in favour of model DIST"
17. State what the p values refer to in general, and in particular when non-nested models are compared. NB that log likelihood ratio test cannot be used to compare non-nested models
18. Can you provide Fig 3d in the same format as 2d ?
19. The authors could more directly mention and discuss the relation to models of conflict/expected value of control and their findings regarding dACC
20. In all figure with color maps: please show the color scale.
21. In Figure 5 it wasn't entire clear whether the shown maps are corrected or uncorrected. Please use

one consistent correction that is introduced early and used in all analyses/figures.

22. In my opinion some effects found to be in vmPFC would be better characterised as being in medial OFC, in particular the quadratic belief effect. I am aware that the the definitions are overlapping, but OFC is an anatomically more established definition and this also links the findings to a relevant body of work

23. What's the difference in regressors associated with Fig. 5A and Fig. 7A? Both appear to be vchosen-vunchosen from model MIX, so does it activate both dmPFC and vmPFC but with negative correlation in the former and positive correlation in the latter?

24. Line 395: should it be "decreased"?

25. Line 400: should it be "increase"?

26. How did you factor out RTs for the fMRI analyses?

27. There seems to be an implication from the fact that within dmPFC you find a linear effect of relative chosen beliefs and at the same time a quadratic of the decision variable: is there an effect of the utilities of the form $x(x-1)$?

28. Is Fig. 7 from the 3-GLM? The 3-GLM doesn't include the relative chosen normalized utilities vch-vun as a regressor, so which GLM is Fig. 7A associated with?

29. What do your findings say about the differentiation of probabilities and rewards, see L Hunt's work?

30. Line 662: in $Bz(t|r)$, is r the reward received?

31. Please report DFs in all stats

32. The manuscript seems inconsistent regarding the use of brit. vs amer. english

33. Line 789: Reduced MIX models: what's the difference between 4 and 5? Also between 3 and 6?

34. Line 891 and 897: in 2-GLM and 3-GLM, the regressor appears in both stimulus onsets and feedback onsets, is that correct?

Reviewer #2 (Remarks to the Author):

Using computational modelling and fMRI the authors investigated how expected rewards and state beliefs are combined to influence value-based choices. The main computational contribution of this work is the proposition that reward expectations and state beliefs are processed as separate components that contribute independently to choice behavior – a finding that contradicts the more mainstream view that these two components are integrated linearly into a single source of evidence that is used to maximize expected utility. The fMRI data offered evidence of the proposed formulation with regions of the vmPFC appearing to encode the two separate representations while the dmPFC integrates these quantities to guide choice.

I think the work is methodologically quite rigorous and the paper itself makes a potentially important contribution to the field. I have a few comments however that ought to help further validate these

findings and rule out alternative interpretations.

My main concern is with regards to the potential RT confound as response times will inevitably scale with changes in the main variable of interest (in fact it would be good to report explicitly the correlation between these two quantities). The authors state in the Methods that they performed an analysis similar to the Grinband et al., 2008 Neuron paper but no further details are given. I think this is a critical point and it deserves a well fleshed out explanation and a clear presentation of the outcome of the chosen analysis.

If I recall correctly, the Grinband paper used an approach whereby only a small subset of trials in which RT variance was small (those on the core of the individual subject RT distributions) were used in a separate analysis to reproduce the main findings. Is this what was done here? This analysis inevitably discards the bulk of the data and as such it's problematic. The more straight forward way to address this issue is to include a separate parametric regressor in the original GLM (along with the main regressor of interest) and show that the dmPFC continues to appear in the decision variable regressor and do in a significantly higher capacity than the RT regressor.

Related to this it would be nice to show the dmPFC BOLD activity for the decision entropy analysis in different bins to convince us that it indeed resembles an inverted V-pattern. This was done for relevant ROIs in the other fMRI analyses and it would be good to see this depicted in Figure 4 as well.

More generally anything that scales with decision uncertainty would present a confound. The presence of other regions in the main regressor of interest (e.g. insula) also point to this. The author's attempt to further titrate the network by testing predictions relating to the additive decreasing effects of beliefs and utilities (as in Fig. 5) is a step in the right direction. In this spirit, how about testing the extent to which dmPFC activity scales with the "omega" parameter in the model *across* individuals (since omega presumably captures inter-individual differences in "risk-taking" tendencies). I believe this will be a further validation for both the neurobiological validity of the model and the relevant involvement of the dmPFC.

A connectivity analysis using the dmPFC as seed would be appropriate to further validate the role of this region as an integrator for belief states and reward expectations. One might expect that the region(s) of the vmPFC identified to encode these quantities would covary with the dmPFC in a task-dependent manner.

I believe unsigned quantities (e.g. decision entropy – fig4 and later analyses on belief/utility differences – fig6) should be captured using a modulus operator rather than introducing quadratic regressors. There is some mention that this was attempted for the data shown in Fig. 4 but it would be interesting to do the same for the developments presented in Fig. 6 and in both cases present the maps to allow direct comparisons with the current results.

More minor points:

The description of the computational modeling under the Results section is currently quite dense and at times too technical. As this is a general interest journal you might consider presenting the relevant model parameters and various model instantiations in a more intuitive fashion (to be comprehensible by a non-expert). There is a bit of this in the text already but it tends to get lost in the technical details.

For consistency, could the modeling results currently presented in Fig. 3d be shown in the same way as those in Fig. 2d (i.e. further divided up into V1+V2 high and low).

Reviewer #3 (Remarks to the Author):

This paper seeks to distinguish between two explanations for suboptimal probabilistic decision-making: (1) the optimal combination of distorted beliefs (i.e. subjective probabilities and utilities) or (2) suboptimal combination of undistorted beliefs and utilities. Participants performed a reversal learning task. Two bandits had reward probabilities (0.8 and 0.2) that periodically reversed, together with randomized, explicitly cued reward magnitudes. Based on the implementation of multiple behavioral models, the authors argue the data are best explained by a model that additively (and thus suboptimally) combines two attributes: "affective beliefs" (related to each bandit's probability of being rewarded), and "utility" (related to the bandit's face value and reward history). Model-based fMRI analyses showed that dmPFC activity scaled with the difficulty of decisions, whereas vmPFC activity scaled with the weight of evidence favoring a given bandit.

This paper reflects a thorough and thoughtful approach to building and testing computational models of decision making, and the fMRI results point to interesting potential distinctions between and within dmPFC and vmPFC. However, I felt the paper also had some significant weaknesses relating to the clarity of presentation and the grounding of conclusions in evidence, which made it difficult to be confident about its scientific contribution.

I. Comments about the computational modeling.

1. Figure 1 shows that behavior differs qualitatively from an optimal model, in that performance was worst in the "incongruent" condition. Figures 2 and 3 then show that the "DIST" and "MIX" models can reproduce this pattern. This is confusing (and perhaps misleading) given that a parameterized version of the optimal model could also fit the same pattern (Fig. S2). So, does the good visual match between model and data in Fig. 2A and 3A have anything to do with the unique features of the DIST and MIX models?

2. Around line 159, the authors are making what I think boils down to a fairly straightforward point: the difference in probabilities should theoretically matter more when the face-values of the bandits are high, and this qualitative pattern was absent in the behavioral data. The description is a little hard to follow, in part because of the run-on sentence that extends from line 164 to 170. More substantively, I didn't understand why it was necessary to binarize the data with a median split in order to perform this analysis. I would have thought a continuous logistic regression would be more straightforward (and in either case, it would be appropriate to include a statistical test of these claims). I also found the visualizations in Fig. 2D-E quite hard to interpret. In 2D, the y-axis scaling makes it hard to verify the effect of relative value and the lack of an interaction in the behavioral data.

3. I had some trouble following the motivation for the various features of the MIX model (which is the model the authors ultimately favor). The model has quite a number of parts, including (1) combining the face-values of the bandits with RL-based values, (2) normalizing the resulting utilities across the choice set, and (3) further combining the result with an estimate of affective beliefs.

- I didn't see a strong justification for the normalization step. The closest the authors come to explaining this is on line 202, where they invite the reader to work through a substitution in Equation 2, but this falls short of a satisfactory explanation. None of the reduced models seem to address whether normalization improves the model fit.

- In general, the model would be better justified if it were tested and validated across multiple data sets. Because the model is both novel and fairly elaborate, there would seem to be some risk of its being tailored to the idiosyncrasies of the data set on which it was developed. This is particularly important because the MIX model is reified and used as the premise for the fMRI analyses.

- It's an interesting proposal that people might treat reward probability and magnitude as separate, additive attributes (rather than properly multiplying them). The distinction is muddled, however, since the "utility" attribute, which encodes reward magnitude, seems likely to also contain information about probability via the RL mechanism. This reduces my confidence that the fMRI analyses successfully dissociated the factors of utility and probabilistic beliefs.

4. Some of the modeling is presented in a needlessly verbose manner that obfuscates the main points. For example, the standard softmax choice function is written as a system of 2 or 3 equations (repeated on lines 671, 745, and 757) where one simpler equation would more clearly express the same meaning. Other related points:

- I didn't follow the derivation of the special-case value for the weighting parameter ω_0 , either in the text on line 223 or in the methods on line 780. Please try to clarify.

- I also didn't understand the point being made in lines 796-806.

- it's unclear what the variable "r" refers to (line 662), is it the set of both nominal reward amounts $\{V_1, V_2\}$? It's also unclear how r was used as a modulator in GLM 1 (line 887).

II. Comments about the fMRI analyses and results.

5. It is not clear what method was used to correct for multiple comparisons. There is mention of a cluster-size threshold (Fig. 4 legend and line 879), and it sounds like this was applied subsequent to an unspecified correction at the voxel level. And why was a more stringent cluster-size threshold used for the analyses restricted to frontal cortex ($k > 40$) and a more lenient threshold used for the whole-brain analyses ($k > 10$)? (line 895)

6. The authors state they used "post-hoc analyses removing selection biases" (line 298), and add in the methods section that this involved a leave-one-out procedure (line 881), but more detail about this would be needed to understand how circularity was avoided in extracting data from the significant clusters.

7. The fMRI analysis introduced starting at line 303 is not described very clearly. I think this passage could be made a lot more succinct. As best I can tell, the prediction is less brain activity when either beliefs or utilities favor the chosen option by a larger margin. The result is described as "the addition of the two negative linear effects" (line 318), which confused me because the authors talk about this result as if it were a conjunction of effects. To support the conclusions, it would be necessary to report the intersection of the two thresholded maps, not their additive combination.

8. The next fMRI analysis looks for activity that increases with larger differences in either beliefs or utilities. It's an interesting finding that vmPFC shows more activity when there is more confidence about which bandit has the higher reward probability. However, the dmPFC data are presented confusingly. For positive values of chosen-minus-unchosen utility (which is presumably the range that contains a majority of the trials), the same dmPFC region is shown to be positively modulated in Fig. 6C but negatively modulated in Fig. 5B. This is because Fig. 5B subtracts the quadratic trend from the data, whereas Fig. 6C subtracts the linear trend. I think it would be clearer to directly show the true pattern of data for this cluster.

9. The authors argue that the fMRI results support the validity of the MIX model (e.g. line 501). This claim might need to be softened, as it isn't supported by any direct comparison between different models' ability to fit the fMRI data (and it seems plausible that other models might produce correlated estimates of decision difficulty and evidence strength). The statement that the DIST model yielded no significant fMRI effects (line 368) is relevant, but not a strong basis for this claim.

10. On line 494, the authors conclude that dmPFC builds a representation of each option's utility in a manner prescribed by the MIX model. But how can the alternative possibility be excluded that dmPFC merely responds to the difficulty of the comparison between values computed elsewhere?

III. General and minor comments.

11. The paper needs general proofreading for grammar and clarity.

12. The introduction assumes background knowledge and familiarity with the task structure that isn't provided. For instance, lines 35-57 introduce a hypothesis about how state beliefs interact with reward expectations. But this hypothesis is specific to a certain kind of choice context that hasn't yet been introduced. This issue also impacts the generality of the paper's conclusions. While it may be that state beliefs exert a direct effect on choice during reversal learning, the connection between latent states and rewards in other situations might not be so straightforward. It's not at all clear how this model would provide a general account of risk aversion (line 470).

13. Some important information about the methods would be worth elevating to the main text:

- The task timing isn't clear from Fig. 1 legend. It gives a range for the "stimulus onset asynchrony" without saying which two stimulus onsets it applies to.
- Please clarify in the main text that the 3 covariance conditions were administered on separate days of scanning.

14. It would be helpful to clarify some aspects of the instructions that were given to participants:

- Were they explicitly informed that the reward probabilities for the two bandits were 80% and 20%?
- Were they given any information about whether/how the task might differ across the 3 sessions?

Point-to-point response to referees' comments

We thank the reviewers for their time and consideration of our original manuscript, and especially for their insightful comments that helped us to clarify the paper and strengthen its content and conclusion. Below is a point-to-point response to all your comments and a description of how we revised the manuscript accordingly.

NB1: the exact changes we made in the revised manuscript are indicated in **red**.

NB2: to keep the **main text** as short as possible, we moved the analysis of RTs from the **main text** to the legend of **Fig. 8**, and suppressed several redundant sentences throughout the manuscript.

Contents:

Reviewer 1: your comments and our responses: pages 1-14

Reviewer 2: your comments and our responses: pages 15-23

Reviewer 3: your comments and our responses: pages 24-39.

Reviewer #1 (Remarks to the Author):

The Study by Rouault et al investigates human decision-making in an environment with outcome probability reversals and outcome magnitude fluctuations. The authors test a series of models in which state beliefs and utilities influence decisions. Rouault et al propose that choice results from the additive contributions of state beliefs and utility values, in contrast to the expected utility theory that integrates utilities over state beliefs to derive subjective probabilities for choice computation. They test the idea in a bandit task by fitting participants' behavior to those contrasting models. Using model comparison and falsification approaches the manuscript presents evidence in support of the hypothesis that beliefs and utilities have independent effects. Additionally, they demonstrate with fMRI that normalized utility value correlates with dmPFC activation, while its two additive components (proposed rewards and RL-values) correlate with activations in vmPFC and laOFC respectively. The study is well written and the analysis is thorough.

1. In the presented study the strength of the beliefs is confounded with the outcome probabilities. This might be a factor contributing to the fact that the MIX model performs well. If the state uncertainty would for instance come from perceptual uncertainty, and would be unrelated to the outcome probabilities, the best fitting model might be quite different. I would like to see evidence that the reported effects regarding belief states cannot be reframed as learned/updated outcome probabilities

Your comment raises the issue whether subjects' performances might simply derive from learning and updating outcome probabilities from feedbacks without developing any state beliefs. As mentioned in the original manuscript, our protocol was built to induce the formation of state beliefs: in the congruent and incongruent condition, proposed rewards conveyed some information about how reward frequencies map onto bandits. The optimal performance (model OPT) therefore requires using this information to better identify

when this mapping reverses, *i.e.* to develop state beliefs. So we presumed that subjects develop such beliefs as postulated in model DIST (distortion model) and MIX (linear mixture model). Yet, we acknowledge that our original manuscript omits to provide clear empirical evidence supporting this basic assumption.

We directly tested this hypothesis using a variant of model DIST and a variant of model MIX, whereby outcome probabilities are learned from feedbacks (rewards vs. no rewards) instead of developing any state beliefs. In both variants, outcome probabilities are learned through reinforcement learning [$q_{t+1(\text{chosen})} = q_{t(\text{chosen})} + \alpha(1_r - q_{t(\text{chosen})})$, with $1_r = 1$ in case of rewards, $1_r = 0$ otherwise], which in absence of state beliefs, closely approximates Bayesian inference over q_t (with beta distribution as priors over q_t). The fitting results show that both variants are poor predictors of subjects' performance compared to both model DIST and MIX. In terms of log-likelihood (LLH), AIC and BIC, both model DIST and MIX largely outperformed any of these variants (LLH: $T(21) > 10.23$, $p < 10^{-8}$; AIC: $T(21) > 9.42$, $p < 10^{-8}$; BIC: $T(21) > 7.39$, $p < 10^{-6}$). These results provide evidence that as presumed in the original manuscript and consistent with the protocol, subjects developed state beliefs about how reward frequencies map onto bandits. In the revised manuscript, we reported these results as follows:

(Results, Behavioural results, p. 10, second para.)

“Finally, both model DIST and MIX assume in agreement with the protocol that participants infer state beliefs rather than directly learn reward probabilities through RL processes. To validate this assumption, we fitted the corresponding RL variants of both models. As expected, we found that both model DIST and MIX outperformed these variants in predicting participants' choices (Likelihood_{DIST/MIX} > Likelihood_{any/variants}: all $Ts(21) > 10.23$, $ps < 10^{-8}$; BIC_{DIST/MIX} > BIC_{any/variants}: all $Ts(21) > 7.39$, $ps < 10^{-6}$).”

2. I find the modelling as it is already quite extensive. But it is unclear how the work relates to RL models that have been shown to influence behaviour and brain activation in a number of studies. The core issue is that the RL models are specified in an unusual way.

2A) The unusual part is that the RL value is assumed to not take into account the proposed value, which is not necessarily what a RL model would do (this pertains both to the DIST and the MIX models). From an RL perspective, a diamond with a 6 inside should be treated as a different bandit than a diamond with a 3 inside (the numbers are relevant for predicting the outcomes, so they should be part of the state/bandit). In other words, the RL component is particularly simplistic, and it remains unclear to me if the additional contribution of the proposed reward in the MIX model is due to the oversimplified RL model or reflects a true non-RL influence on participants' choices.

First of all, RL-values in Model DIST and MIX corresponds to the standard RL model in the literature, whereby RL-values are bandits' *retrospective* values reflecting the influence of *past rewards* onto current behaviour through the Rescorla & Wagner's rule. In the present protocol as you mention, such RL-values alone are unlikely to account for subjects' performances, which also depends upon proposed rewards (*i.e.* bandits'

prospective values). For that reason, **Supplementary Fig. 2** shows the fitting of the RL model linearly combining RL-values and proposed rewards (which failed to account for human data). Second, model MIX also linearly combines RL-values and proposed rewards. As you rightly note, model MIX consequently raises the issue (while remaining agnostic) of whether proposed rewards should be conceptually considered as an additional RL component or not. Our fMRI data provide evidence that proposed rewards and RL-values are processed as two functionally distinct components of option values: proposed rewards were encoded in the vmPFC with respect to actual choices, whereas RL-values were encoded in the laOFC, irrespective of actual choices. In the revised manuscript, we highlighted this point in the Discussion as follows:

(Discussion, p. 17, first para.)

“[...] Knowing that the functional segregation between laOFC and vmPFC has long been debated (Rushworth et al., 2011), the present results indicate that **these regions functionally differ along at least two dimensions: (1) while the laOFC encodes the retrospective appetitive value, the vmPFC encodes the prospective appetitive value of available options; (2) the laOFC encodes retrospective values irrespective of actual choices, whereas the vmPFC encodes prospective values with respect to actual choices. This segregation confirms that retrospective and prospective values are two functionally distinct components of option values.**”

In the legend of **Supplementary Fig. 2**, we also clarified that the simulated RL model comprises both RL-values and proposed rewards:

(**Supplementary Fig. 2**, legend)

“**Sup. Fig. 2 (related to Figs. 2 & 3). Simulations of fitted models RL (with proposed rewards) and OPT (parameterized).**

Proportions of choosing true best bandits (maximizing reward frequency x proposed reward monetary value) according to trial number after reversals (A) and choice proportion of proposed rewards (B) for participants (blue, same data as in Fig. 1), model OPT (parameterized, grey) and model RL (**including proposed rewards**, red). Data are shown separately for the conditions congruent, neutral and incongruent. Parameterized model OPT corresponds to model DIST with no distortions on reward probabilities and values, and was fitted to human data with the same free parameters except the four parameters specifying distortions (see Supplementary Table S2). Model RL shown here included four free parameters: inverse temperature β , lapse rate ϵ , learning rate α , and contribution of proposed rewards to RL values ϕ at decision time. Error bars are s.e.m. across participants. Note that none of the models succeed in reproducing human data both **along trial series following reversals and according to proposed rewards.**”

*2B) I am confused by the double influence of the outcome probabilities. It is stated that the beliefs are updated according to $B(t/r,x) \sim q_M^{I(t)} q_m^{(1-I(t))} B(t/r)$. Later the reward probabilities are computed again involving q_m/M : $P(t) = (q_m q_M) * B(t/r)$. I might be misunderstanding, but this seems to imply that the probabilities have a quadratic influence on choices?*

In the optimal model (model OPT), bandits' reward frequencies (q_M, q_m) have a

quadratic influence on *how the decision variable varies across two successive trials*. This quadratic effect results from the integration of two linear influences: (1) a first linear influence in updating state beliefs from one trial to the next according to chosen actions and actual feedbacks; (2) a second linear influence in deriving the decision variable from state beliefs through marginalization processes. This quadratic influence stems from the formation of state beliefs reflecting the uncertainty upon how reward frequencies (q_M , q_m) map onto bandits. If there were no uncertainty (e.g. beliefs $B_1(t)=1$ and $B_2(t)=0$, implying $B_1(t+1)=1$ and $B_2(t+1)=0$), reward frequencies would have only a linear influence onto the decision variable as these certain state beliefs remain constant. By contrast, model MIX assumes no marginalization processes, so that reward frequencies (q_M , q_m) have only a linear influence on how the decision variable varies across successive trials due to beliefs' updating (point 1 above). Following your comment, we added the following two clarification statements in the Methods:

(Methods, computational modelling: optimal model OPT, p.22, first para.)

“Note finally that reward frequencies (q_M, q_m) have a quadratic influence on how log-odds vary across two successive trials as a result of updating and marginalizing over beliefs.”

(Methods, computational modelling: alternative model MIX, p.25, second para.)

“Note that in model MIX, reward frequencies (q_M, q_m) have a linear influence on how log-odds vary across two successive trials as a result of beliefs' updating only (see mathematical derivations for model OPT).”

2C) more generally, it would be important to know how models that incorporate either state uncertainty (belief state models, see e.g. Dayan & Daw 2008, Cognitive, Affective, & Behavioral Neurosci, 8, 429-453.) or belief updating (e.g. Nassar et al, 2010, J Neurosci 30; Hang et al., 2015, J Neurosci., 35) would perform in this task (I am not an author on any of the listed papers).

Models OPT, DIST and MIX are based on the same belief model, which corresponds to the formulation of belief state models described in Dayan & Daw (2008) and applied to our protocol. As you point out, we should have cited this reference in our original manuscript. Belief updating/inference in Nassar et al (2010) is also based on the same probabilistic computational framework described in Dayan & Daw (2008): Nassar et al. simply derive a delta-rule algorithm that approximates the optimal probabilistic (i.e. Bayesian) model inferring state beliefs. In our protocol, the optimal belief inference model remains quite simple as reward frequencies (q_M, q_m) were constant and consequently, was used in models OPT, DIST and MIX. Accordingly, Nassar et al.'s algorithm was not useful and would provide the same results for two reasons: (1) it is a fairly good approximation of the optimal model; (2) our results discriminate models OPT, DIST and MIX, although these models are based on the same belief inference model. Thus, our results do not bear upon how state beliefs are updated but how state beliefs are integrated with reward values in decision-making.

Furthermore, Dayan & Daw (2008) outline two general issues that we addressed in our study: (1) the construction of a relevant state space; (2) the elaboration of relevant

approximations in computing behavioural policies. Compared to models OPT and DIST, model MIX addresses these two issues jointly. Regarding approximations first, model MIX linearly combines reward values and state beliefs rather than combines them in a multiplicative way through a cross-product marginalization processes. This approximation is meaningful only because model MIX assumes *affective latent states* yielding to affective state beliefs (as we named them): namely, “bandit 1 is the most frequently rewarded option” vs. “bandit 2 is the most frequently rewarded option”. By contrast model OPT and DIST assumes only abstract latent states: namely, “Paired bandits (1,2) is rewarded with frequencies (q_1, q_2), respectively ” vs. “Paired bandits (1,2) is rewarded with frequencies (q_2, q_1), respectively ”, where q_1, q_2 are indifferently equal to reward frequencies q_m or q_M .

To sum up, models OPT, DIST and MIX all belongs to the class of belief state models described in Dayan & Daw (2008). Nassar et al.’s approximation of belief updating applied to this class of models will lead to the same results. Accordingly, we revised our manuscript by adding the missing reference to Dayan & Daw (2008) when we describe the optimal model OPT including state beliefs and from which both models DIST and OPT derive:

(Main text, Results, Behavioural protocol, p. 5, second para.):

“In every trial, the optimal performance model (referred to as model OPT) forms probabilistic beliefs from previous trials **about how reward frequencies map onto bandits**, updates these beliefs according to proposed rewards and finally, chooses the bandit maximizing the (objective) expected utility by marginalizing reward expectations over beliefs (**Dayan & Daw 2008; Methods**)”

(Methods, Computational modelling: optimal model OPT, p. 19, last para.)

“We denote 1 and 2 the two bandits. Formally, the protocol consists of a sequence of hidden states $z_1, z_2, \dots, z_t \in \{1,2\}$, where $z_t = 1$ (=2, resp.) means that in trial t , bandit 1 (2, resp.) is the rewarded bandit with probability q_M and bandit 2 (1, resp.) is the rewarded bandit with probability q_m (q_M and q_m equal to 80% and 20%, respectively) (**Dayan & Daw, 2008**).

3. The introduction reads very well, but results were a bit hard to follow because the models are introduced piecewise in between behavioral effects. I would suggest to introduce the models in a separate section that emphasizes which hypotheses they reflect. In this regard it would also be good to avoid using mathematical notation that overlaps with, but has a different meaning in, other models such as using ‘Q’ for choice history, although Q is quite established notation for expected state-action value in RL models.

We followed your suggestion. In the revised manuscript, we moved and rewrote the description of the different models in a separate section within Results entitled “Computational modelling”. This new section minimizes mathematical details for improving readability and focusing on concepts. The section appears before presenting the model fitting results and behavioural effects. As you also suggest, we used “ p ” instead of “ Q ” for choice probabilities. Here is the new section “Computation modelling”:

(Results, pp. 5-7)

“Computational modelling

Distortion models of human decision-making. To account for human suboptimal performances, we first considered the standard *distortion hypothesis* we referred to as model DIST (Savage, 1954; Hertwig and Erev, 2009; Kahneman and Tversky, 1979; Zhang and Maloney, 2012). This model is identical to model OPT, except that choices maximize the *subjective* expected utility of choice options. Subjective utilities involve (1) subjective probabilities distorting objective probabilities that derive from marginalizing reward expectations over state beliefs; (2) subjective appetitive values distorting monetary values of proposed rewards. We used the standard distortion function encompassing all previously documented distortions (convex, concave, S-shape and inverted S-shape) on reward probabilities derived from state beliefs as well as on proposed rewards modelling subjective appetitive values (Zhang and Maloney, 2012) (**Methods**). Model DIST thus includes four additional free parameters characterizing such distortions with model OPT as a special case.

Suboptimal performances might also stem from reinforcement learning (RL) whereby the reward history of past choices biases current choices. To assess this effect, we considered the extension of model DIST (referred to as model DIST+RL) comprising an additional RL component based on the Rescorla & Wagner’s rule (Rescorla and Wagner, 1972) that alter the subjective expected utilities of bandits (**Methods**).

Mixture models of human decision-making. As outlined in the introduction, the distortion hypothesis might reflect that state beliefs and reward expectations are actually not integrated as posited in the expected utility theory (i.e. as in models OPT and DIST) (Beck et al., 2012). First, the distortion over monetary values of proposed rewards might actually reflect the recently proposed idea that option utilities are encoded in the brain according to a context-dependent divisive normalization mechanism (Louie et al., 2015; 2013), so that utilities of choices’ options are mutually dependent. Second, the notion of subjective probabilities distorting objective probabilities might actually reflect that no marginalization processes occur over abstract state beliefs such as “*bandit 1 (2 resp.) is rewarded with frequency q_1 (q_2 , resp.)*” (as in models OPT and DIST). Instead, beliefs might bear upon affective states such as “*bandit 1 (2 resp.) is the most frequently rewarded bandit*”, so that utilities and beliefs act as two normalized appetitive value components contributing to choices independently, *i.e.* additively. These two hypotheses, utility-normalization and affect-additivity, are tightly linked because: (1) affect-additivity requires making utilities commensurable to probabilistic beliefs, thereby leading to utility-normalization; (2) model OPT with normalized utilities and affective state beliefs is equivalent to maximizing a weighted sum of normalized utilities and affective state beliefs with specific weights denoted $1 - \omega_0$ and ω_0 , respectively (**Methods**).

Accordingly, we proposed the more general decision-making model (we refer to as model MIX) assuming that choices maximize a linear combination of normalized utilities and affective state beliefs with weighting parameters $1 - \omega$ and ω , respectively (**Methods**). If ω significantly differs from critical value ω_0 , choices are unlikely to derive from computing bandits' expected normalized utilities based on marginalization processes over affective state beliefs. Moreover, $\omega > \omega_0$ vs. $\omega < \omega_0$ implies more belief-based (risk-averse) vs. more utility-based (risk-prone) choices, respectively. Model MIX includes no distortion functions but, according to the context-dependent divisive normalization (Louie et al., 2015; 2013), utilities v_1, v_2 of choice options correspond to proposed rewards possibly altered by the reward history and normalized across the two choice options:

$$v_i = \frac{\varphi V_i^{proposed} + (1-\varphi)V_i^{RL}}{\sum_{i=1,2}(\varphi V_i^{proposed} + (1-\varphi)V_i^{RL})},$$

where φ is a free weighting parameter, $V_i^{proposed}$ the monetary value of proposed rewards and V_i^{RL} the RL-values associated with bandits i ."

Additional points:

4. I disagree with the statement about exploration in the introduction: isn't exploration still necessary, since outcome probabilities are changing? E seems more about dynamic/changing environments, rather than how much info one can get in each trial

Our statement was indeed misleading. We agree that exploration is unrelated to *how much* info one can get in each trial. Instead, exploration is not required if you get *the same information whatever the options you choose*. If so, the behaviour that maximizes rewards *over trial series* is to choose the option maximizing rewards in *every* trial. In the present protocol, due to the reversal/symmetrical structure of the protocol, the task required no exploration for maximizing rewards because subjects get the same information whatever the option they choose in every trial. Accordingly, we changed the original statement with the following one:

(Results, behavioural protocol, p. 5, first para.)

"Note also that due to the reversal/symmetrical structure of the protocol, the task required no exploration for maximizing rewards: **participants got the same information about bandits' reward frequencies, whatever the option they choose in every trial.**"

5. Fig 1C: I would be good to see data until trial 28. Is there evidence that discrete reversal structure of occurrence at either 16/20/24/28 is picked up upon? Relatedly: Can our show how belief states develop as a function of trial history after the last reversal? What is the step size in belief updating?

Following your suggestion, we included human performance until trial 28 in **Fig. 3A** (i.e.

when these data are compared to model MIX’s performance, see below). Note that after trial #16, data become noisier as the mean is computed over fewer episodes. As you can further notice, there is no evidence that participants picked up upon discrete reversal structure of occurrence at either trial 16/20/24/28. Following your comment, we carried many analyses on the time course of state beliefs to try to identify possible discrete reversals in human performance, for instance realigning participants’ performances on when beliefs reversed. We found no evidence of discrete switches in participants’ performances.

(Fig. 3A)

6. It would be good to include information about the response deadline and the trial timing directly into figure 1

We included timing information directly into **Fig. 1**, as you suggest.

7. It would be good to have a table of BIC values for all the models, along with AIC

We added **Supplementary Table 6** showing the Likelihood, BIC and AIC for the main tested models (MIX, DIST, DIST+RL, parameterized OPT, parameterized OPT+RL, RL+proposed reward). The table is referred in **Methods**, section **model fitting and comparison**.

8. It might be worth referring to the fact that participants exhibit the well-known effect of probability matching

In the revised manuscript, we added the following sentence:

(Results, behavioural protocol, p. 5, second para.)

“(corresponding to “probability matching”)”

9. How does the modeling account for trials with no responses?

In trials with no responses, RL-values remain unchanged, while beliefs were updated only based on proposed rewards and volatility. We added the following sentences in **Methods**:

(Methods, computational modelling: optimal model OPT, p. 21, first para.)

“In trials with no responses, beliefs were updated based on proposed rewards and volatility only.”

(Methods, computational modelling: Model DIST. Hybrid model DIST+RL, p. 24, first para.)

“In trials with no responses, RL values remained unchanged.”

10. What t test is done when conditions are compared: mean reward or only the last trials before a reversal? Mean over all trials, or only 1-16 as shown in the figure?

Using paired T-tests, we compared the plateau across the three conditions, i.e. the mean over trials from trial 10 to next reversals. We clarified as follow:

(Results, Behavioural protocol, p. 5, second para.)

(mean over trials from trial#10, paired T-tests, both $T_s(21) > 2.97$, $P_s < 0.01$)

11. In Fig 2A, say what green is. In Fig 2C, it would be useful to have an identity line in the left panel too

We indicated what green is in **Fig. 2A**. We added the identity line in **Fig. 2C**, left panel.

12. Fig 2A: please overlay the participants' data for visual comparison.

Participants' data were overlaid in the original **Fig. 2A**. Following your suggestion in point #11, this should be now clear.

13. Fig 2D would be easier to read if data is displayed using bar graphs. The current x axes ("-0.5" despite the categorical distinctions) do not make sense.

We changed x-axes as you suggest (we removed labels -0.5, 0, 0.5). We found bar graphs were not easier to read, so we kept original **Fig. 2D**.

14. What does model OPT predict regarding the interaction of V and P? Is it really true that all reduced models of DIST are generally ruled out by its failure to capture some aspects of the data? I understand the logic, but what about for instance overfitting?

All reduced models from model DIST, including model OPT, predict exactly the same additive and interaction effects as model DIST (with or without distorted variables) (see the log-odds equation for model DIST in the main text). This is a constitutive feature of these models stemming from multiplying probabilities and values, independent of free parameters.

Model DIST including more distortion parameters would be potentially (though unlikely) better at removing these effects (which are absent in the data). Our analysis confirms that as expected, fitting model DIST failed to remove these effects. The conclusion is therefore that none of these reduced models can remove these effects and accounts for the data. Thus, the inference is correct for two reasons: (1) all models predict the same effects as a constitutive feature, independent of free parameters; (2) model DIST comprises more free parameters. We clarified this point in the revised manuscript as follow:

(Results, behavioural results, p. 8, last para.)

“Thus, neither models **DIST/DIST+RL** nor *a fortiori* all their reductions, including especially the standard mixture **OPT+RL predicting the same interaction effects**, accounted for how participants made choices.”

15. I think the RL part in DIST+RL could be explained better in the text

In response to your comment #3, we rewrote the presentation of computational models in a separate section. We believe the RL part in DIST+RL is now better explained in the revised text:

(Results, Computational modelling, Distortion models of human decision-making, p. 6, second para.):

“Suboptimal performances might also stem from reinforcement learning (RL) whereby the reward history of past choices biases current choices. To assess this effect, we considered the extension of model **DIST** (referred to as model **DIST+RL**) comprising an additional RL component based on the Rescorla & Wagner’s rule (Rescorla and Wagner, 1972) that alter the subjective expected utilities of bandits (**Methods**).”

*16. Line 188: I think the p value indicates that there is no evidence against the null that $DIST = DIST+RL$, not that it is “in favour of model **DIST**”*

We agree our statement was ambiguous. We actually used “in favour of model **DIST**” to mention that the BIC was lower (i.e. better) (though not significant) for model **DIST** than **DIST+RL**. In the revised manuscript, we clarified the statement as follows:

(Results, behavioural results, p. 8, last para.)

“The increased complexity of model **DIST+RL**, however, **failed to improve the fit** ($BIC_{DIST} < BIC_{DIST+RL}$; paired T-test, $T(21)=0.77$ $p=0.44$).”

17. State what the p values refer to in general, and in particular when non-nested models are compared. NB that log likelihood ratio test cannot be used to compare non-nested models

In the revised manuscript, we indicated that p-values refer to paired T-tests, unless specified otherwise. We further indicated when the comparison involves non-nested models (MIX vs. DIST).

(Results, behavioural results, p. 9, last para.)

“($BIC_{MIX} < BIC_{DIST} < BIC_{DIST+RL} < BIC_{OPT+RL}$: all $T_s(21) > 3.54$, $p_s < 0.005$, **non-nested models**)”

18. Can you provide Fig 3d in the same format as 2d ?

In the revised manuscript, we provided **Fig. 3D** in the same format as **Fig. 2D** (see below). For clarity, we only split in two graphs the median split between lower and higher values of total proposed rewards $V_1^{Pf} + V_2^{Pf}$. The graphs clearly show that there

were neither main nor interaction effects of total proposed rewards $V_1^{pr}+V_2^{pr}$, but only two additive main effects of relative normalized utilities and state beliefs. We reported the statistics in the figure legend:

(Fig. 3D)

Fig. 3. Model MIX fit to human performances. **A**, proportions of choosing true best bandits (maximising reward frequencies \times proposed rewards) following reversals. **B**, proportions of choosing proposed rewards. In **A** & **B**, mean proportions over individuals (\pm s.e.m) are shown in the congruent, neutral and incongruent condition for participants (blue), and fitted model MIX (purple) simulations. See **Supplementary Fig. 2** for model RL fit. **C**, Parameter estimates of proposed reward biases in the congruent, neutral and incongruent conditions in model MIX fitting human choices, $*p < 0.05$. These estimates are similar to true values (± 0.13 , see **Fig. 1B**) indicating that according to model MIX, participants adequately inferred state beliefs. **D**, factorial analysis of choice log-odds over trials sorted according to inferred state beliefs ($B_1 > B_2$ vs. $B_1 < B_2$), normalized utilities ($v_1 > v_2$ vs. $v_1 < v_2$) and total proposed rewards ($V_1^{pr} + V_2^{pr}$, median split: left vs. right panel) for model MIX simulations (purple) and human data (blue). Both model MIX and human data exhibited main effects of beliefs and utilities (all $F_s(1,21) > 116.9$, $p_s < 0.00001$), no main effects of total proposed rewards (both $F_s(1,21) < 1.2$, $p_s > 0.47$) and no interactions between these factors (all $F_s(1,21) < 3.3$, $p_s > 0.14$). Error bars are s.e.m. over participants. See **Supplementary Table 1** for model best-fitting parameters.

19. The authors could more directly mention and discuss the relation to models of conflict/expected value of control and their findings regarding dACC

Following your suggestion, we added the following sentence in the revised manuscript: (Discussion, p. 15, second para.)

“Overall, our findings suggest that the dmPFC collects multiple reward-related variables (proposed rewards, RL-values, affective state beliefs) that concurrently guide behaviour rather than integrates these variables into an overall expected value of control guiding behaviour (Shenhav et al., 2016)”

20. In all figure with color maps: please show the color scale.

We showed the colour scale in the figures with colour maps: i.e. **Figs. 4 & 7** and **supplementary Fig. 3**.

21. In Figure 5 it wasn't entire clear whether the shown maps are corrected or uncorrected. Please use one consistent correction that is introduced early and used in all analyses/figures.

All statistical thresholds in all analyses/figures were set as follows: $p < 0.05$, voxel-wise, FWE-corrected for multiple comparisons over search volumes (whole brain in 1-GLM; frontal lobes in 2-GLM & 3-GLM) along with cluster-wise threshold set at $p < 0.05$. This latter threshold corresponds to different cluster-sizes according to voxel-wise FWE-corrected thresholds ($k > 10$ voxels in 1-GLM and $k > 40$ in 2-GM & 3-GLM). In the revised text, we followed your suggestion for avoiding confusion: we mentioned throughout all figure and table legends that voxel-wise thresholds were set to $p < 0.05$, FWE-corrected through search volumes and cluster-wise thresholds were set to $p < 0.05$.

22. In my opinion some effects found to be in vmPFC would be better characterised as being in medial OFC, in particular the quadratic belief effect. I am aware that the definitions are overlapping, but OFC is an anatomically more established definition and this also links the findings to a relevant body of work

Indeed, the vmPFC and mOFC definition are overlapping. We agree that our vmPFC activations may also be characterized as mOFC activations. Yet, the vmPFC label is more often used in the neuroimaging literature. So we preferred to keep the vmPFC label throughout the paper and in its first occurrence, to mention it may also refer to medial OFC:

(Results, fMRI activations, p. 12, first para.):

“Activations conveying utility information were found only in the dmPFC identified above (**Fig. 6, Supplementary Table 4**), whereas activations conveying belief information were found only in the ventromedial PFC (vmPFC) **or medial orbitofrontal cortex (Fig. 6, Supplementary Table 4).**”

23. What's the difference in regressors associated with Fig. 5A and Fig. 7A? Both appear to be vchosen-vunchosen from model MIX, so does it activate both dmPFC and vmPFC but with negative correlation in the former and positive correlation in the latter?

You are right. This is explicitly mentioned in the legends of **Fig. 5A** and **7A**, and in the main text.

24. Line 395: should it be “decreased”?

Right. We corrected.

25. Line 400: should it be “increase”?

Right. We corrected.

26. How did you factor out RTs for the fMRI analyses?

RTs were factored out using an additional parametric regressor at the first level in all regression analyses. In the revised manuscript, we clarified this point as follows (we

removed the reference to Grinband, 2008, Neuron, which was misleading as this paper uses another method to remove this confounding factor):

(Methods, fMRI statistical analyses, p. 28, last para.)

“We systematically performed control analyses for reaction times: in all GLMs, we further included along with regressors of interest an additional parametric regressor factoring out reaction times. All results reported in the main text remained unchanged when including this additional regressor.”

27. There seems to be an implication from the fact that within dmPFC you find a linear effect of relative chosen beliefs and at the same time a quadratic of the decision variable: is there an effect of the utilities of the form $x(x-1)$?

Our results show that dmPFC vary with $-\beta_1 x + \beta_2 x^2$ with $x = v_{\text{chosen}} - v_{\text{unchosen}}$ and $\beta_1, \beta_2 > 0$ the partial correlation coefficients. There was no significant evidence that β_1 differs from β_2 , so we cannot reject the null hypothesis that dmPFC vary with $x(x-1)$. But this is a null result. Moreover, regressors x and x^2 are not fully orthogonal, so that the shared variance is captured in the GLM residuals: residuals may partly vary with x and/or x^2 . Consequently, accepting this null hypothesis remains highly speculative. What the results clearly show is that both x and x^2 can be linearly decoded from dmPFC activations: i.e. dmPFC activations encode both chosen-relative-to-unchosen utilities (reflecting decision processes) and unsigned relative utilities (reflecting utility comparison).

28. Is Fig. 7 from the 3-GLM? The 3-GLM doesn't include the relative chosen normalized utilities $v_{\text{ch}} - v_{\text{un}}$ as a regressor, so which GLM is Fig. 7A associated with?

Fig. 7A is related to 2-GLM, whereas **Fig. 7B** is related to 3-GLM. In the revised manuscript, we explicitly mentioned this information in the figure legend.

29. What do your findings say about the differentiation of probabilities and rewards, see L Hunt's work?

In Hunt's work (Hunt et al, 2012, Nature Neurosci; Jocham et al., 2012, Nature Neurosci; Chau et al., 2014, Nature Neurosci; Jocham et al., 2014, Neuroimage), outcome probabilities and values are explicitly delivered in every trial in a standard risky gamble task (proposed rewards along with proposed probabilities). These studies show that both proposed outcome probabilities and values influence subjects' choices and correlate with vmPFC activations. As mentioned in our original manuscript (citing his primary paper Hunt et al., 2012, Nature Neurosci), these findings are broadly consistent with our results indicating that both proposed rewards and probabilistic beliefs are encoded in vmPFC. However, Hunt's work neither investigates the formation of state beliefs nor how outcome probabilities and values are combined to guide choices. These studies were based on the standard model whereby probabilities are multiplied with values as in models OPT and DIST, but no comparisons were made with alternative integration models like our model MIX. Thus, the links between our study and Hunt's work remain so far quite tenuous. So we believe that further discussing Hunt's work is beyond the scope of the present paper.

30. Line 662: in $B_z(t|r)$, is r the reward received?

Thank you for pointing out that we omitted to mention what “ r ” refers to. “ r ” refers to proposed rewards, *i.e.* $r = \{V_1, V_2\}$. Accordingly, $B_z(t|r)$ denotes the beliefs updated according to proposed rewards. We corrected this omission in the revised manuscript:

(Methods, computational modelling: optimal model OPT, p. 20, second para.)

“The model then observes proposed rewards $r = \{V_1(t), V_2(t)\}$ in trial t , and updates her/his beliefs $B_z(t|r)$, which write as follows: [...]”

31. Please report DFs in all stats

We reported all DFs throughout the revised manuscript.

32. The manuscript seems inconsistent regarding the use of brit. vs amer. english

We corrected the manuscript using British English.

33. Line 789: Reduced MIX models: what’s the difference between 4 and 5? Also between 3 and 6?

We clarified in the revised manuscript as follows:

(Methods, Computational modelling: alternative model MIX, p. 26, first para.):

“*Reduced models from model MIX.* To validate the mixture comprising state beliefs, proposed rewards and RL values, we considered the six reduced models nested in model MIX: (1) one comprising only state beliefs ($\omega=1$); (2) one comprising no state beliefs ($\omega=0$); (3) one comprising only proposed rewards ($\omega=0; \varphi=1$); (4) one comprising no proposed rewards ($\varphi=0$), *i.e. comprising only RL-values and beliefs*; (5) one comprising only RL values (pure RL); (6) and finally, one comprising no RL values ($\varphi=1$), *i.e. comprising only proposed rewards and beliefs.*”

34. Line 891 and 897: in 2-GLM and 3-GLM, the regressor appears in both stimulus onsets and feedback onsets, is that correct?

In 2-GLM and 3-GLM, the regressors at stimulus onsets (regressors of interest) were as specified. At feedback onsets, the regressors were exactly as in 1-GLM (*i.e.* including relative chosen beliefs $B_{ch} - B_{un}$, RL-values V_{ch}^{RL} and feedback values), which was indeed not clearly specified. In the revised manuscript, we clarified as follows:

(Methods, fMRI statistical analyses, p. 30, first para.):

“2-GLM: [...]. Parametric modulations at feedback onsets were as in GLM#1 above. [...]”

(Methods, fMRI statistical analyses, p. 30, last para.)

“3-GLM: [...]. Parametric modulations at feedback onsets were as in GLM#1 above. [...]”

Thank you for your review.

Reviewer #2 (Remarks to the Author):

Using computational modelling and fMRI the authors investigated how expected rewards and state beliefs are combined to influence value-based choices. The main computational contribution of this work is the proposition that reward expectations and state beliefs are processed as separate components that contribute independently to choice behavior – a finding that contradicts the more mainstream view that these two components are integrated linearly into a single source of evidence that is used to maximize expected utility. The fMRI data offered evidence of the proposed formulation with regions of the vmPFC appearing to encode the two separate representations while the dmPFC integrates these quantities to guide choice.

I think the work is methodologically quite rigorous and the paper itself makes a potentially important contribution to the field. I have a few comments however that ought to help further validate these findings and rule out alternative interpretations.

My main concern is with regards to the potential RT confound as response times will inevitably scale with changes in the main variable of interest (in fact it would be good to report explicitly the correlation between these two quantities). The authors state in the Methods that they performed an analysis similar to the Grinband et al., 2008 Neuron paper but no further details are given. I think this is a critical point and it deserves a well fleshed out explanation and a clear presentation of the outcome of the chosen analysis. If I recall correctly, the Grinband paper used an approach whereby only a small subset of trials in which RT variance was small (those on the core of the individual subject RT distributions) were used in a separate analysis to reproduce the main findings. Is this what was done here? This analysis inevitably discards the bulk of the data and as such it's problematic. The more straight forward way to address this issue is to include a separate parametric regressor in the original GLM (along with the main regressor of interest) and show that the dmPFC continues to appear in the decision variable regressor and do in a significantly higher capacity than the RT regressor.

Thank you for raising this important point. Actually, we exactly did what you suggest and not what was done in Grinband (2008, Neuron). Thanks for pointing out that the reference to Grinband was misleading (we apologize for the mistake: we only referred to this paper to highlight the importance of controlling for RTs). In the revised manuscript we removed the reference to Grinband and we detailed what we did as follow:

(Methods, fMRI statistical analyses, p. 28, last para.)

“We systematically performed control analyses for reaction times: in all GLMs, we further included along with regressors of interest an additional parametric regressor factoring out reaction times. All results reported in the main text remained unchanged when including this additional regressor.”

As you request, we also provided in the revised manuscript an additional figure as supplementary information (**Supplementary Fig. 3**, see below) showing the results of the decision variable analysis (in **Fig. 4**) factoring out reaction times. Compared to **Fig. 4**,

Supp. Fig. 3 particularly shows that dmPFC activations remained *unchanged* whether reaction times were factored out or not.

(Supplementary Fig. 3)

In the revised manuscript, we further referred to **Supp. Fig. 3** as follow:

(Results, fMRI activations, p. 10, last para.):

“The whole brain analysis (**Fig. 4, Supplementary Table 3**) revealed that these activations were located in the dorsomedial PFC (dmPFC including the dorsal anterior cingulate cortex), bilateral PFC (Brodmann Area 9/8), inferior parietal lobules, precuneus, right insular and frontopolar cortex. **In the frontal lobes, however, only dmPFC and right PFC activations remained significant when factoring out reaction times (Fig. 4 and supplementary Fig. 3).** The dmPFC and right PFC were thus the frontal regions potentially involved in computing choices based on linearly combining affective state beliefs and normalized utilities.”

Related to this it would be nice to show the dmPFC BOLD activity for the decision entropy analysis in different bins to convince us that it indeed resembles an inverted V-pattern. This was done for relevant ROIs in the other fMRI analyses and it would be good to see this depicted in Figure 4 as well.

We added the plot you mention into **Fig. 4** and changed the legend accordingly. The plot clearly shows that the dmPFC exhibited an inverted U-pattern:

*More generally anything that scales with decision uncertainty would present a confound. The presence of other regions in the main regressor of interest (e.g. insula) also point to this. The author’s attempt to further titrate the network by testing predictions relating to the additive decreasing effects of beliefs and utilities (as in Fig. 5) is a step in the right direction. In this spirit, how about testing the extent to which dmPFC activity scales with the “omega” parameter in the model *across* individuals (since omega presumably captures inter-individual differences in “risk-taking” tendencies). I believe this will be a further validation for both the neurobiological validity of the model and the relevant involvement of the dmPFC.*

Note first that the dmPFC was the only region that exhibited activations consistent with the computation of choices predicted by model MIX: namely, (1) activations varying as an inverted U-shape function of the decision variable (factoring out reaction times, see our response above); (2) activations decreasing with both chosen beliefs and utilities with no interactions between these factors. Second, we carried out the analysis you suggest: consistent with model MIX, the (negative) linear effect of affective beliefs relative to normalized utilities in dmPFC tended to increase with weight “omega” favouring affective beliefs ($r=0.25$, $p=0.27$). The lack of statistical power, unfortunately, prevents us to find a significant result for this 2nd order interaction. Yet, we found the following significant main effect: vmPFC activations associated with the encoding of affective beliefs (quadratic effects) significantly increased with weight “omega” ($r=0.44$, $p=0.048$). This suggests that weight “omega” reflects the encoding strength of affective beliefs in vmPFC. Accordingly, we added this result in the revised manuscript as follows:

(Results, fMRI activations, p. 11, first para.)

“Consistent with the dominance of affective state beliefs in model MIX, these vmPFC activations were much larger than dmPFC activations. **Across participants, moreover, these vmPFC activations increased with weight ω**

favoring affective beliefs in model MIX ($r=0.44$, $p=0.048$).”

(Discussion, p. 17, last para.)

“We speculate that this relative contribution might reflect an overall subjective scaling of outcome values and/or the encoding strength of affective beliefs with respect to the instability (*i.e.* volatility) of external contingencies over time.”

A connectivity analysis using the dmPFC as seed would be appropriate to further validate the role of this region as an integrator for belief states and reward expectations. One might expect that the region(s) of the vmPFC identified to encode these quantities would covary with the dmPFC in a task-dependent manner.

We agree that functional connectivity analyses would be interesting extensions of the activation results presented in the paper. As you note, one might expect belief state information and proposed reward values encoded in the vmPFC along with RL-values encoded in LaOFC to be transferred to the dmPFC. Our feeling however is that these additional analyses will add little to the main results/conclusions of the paper both in terms of validation and information. Indeed, the current paper provides clear evidence supporting its main claims: namely (1) state beliefs, prospective rewards and RL-values are encoded in the vmPFC and LaOFC; (2) the dmPFC linearly combines all these variables through a normalization mechanism rather than computes expected subjective utilities (as posited in the expected utility theory) to guide choices. Given that our paper already presents a complex series of analyses and is quite long and dense, we believe that including an additional series of functional connectivity analyses will unnecessarily dilute the key findings. So we would prefer not to report these analyses unless, of course, you strongly believe the converse.

I believe unsigned quantities (e.g. decision entropy – fig4 and later analyses on belief/utility differences – fig6) should be captured using a modulus operator rather than introducing quadratic regressors. There is some mention that this was attempted for the data shown in Fig. 4 but it would be interesting to do the same for the developments presented in Fig. 6 and in both cases present the maps to allow direct comparisons with the current results.

We modelled the coding of belief/utility information using quadratic regressors, as information coding varies negatively with belief/utility entropy between choice options. Indeed, the quadratic function approximates this belief/utility entropy better than the modulus operator. Regarding the decision variable, fMRI activity was previously shown to vary with the area under evidence accumulation traces in sequential sampling models (Pisauro et al., 2017, Nature Comm.). As these traces form *on average* linear increases which slopes scale with the decision variable, the area under these traces vary as an inverted quadratic function of the decision variable. Thus we also chose the quadratic expansion of model MIX’s decision variable rather than the modulus operator to identify regions involved in choice computations.

Following your suggestion, anyway, we added an additional figure as supplementary

information in the revised paper (namely **Supplementary Fig. 4** shown below) showing the activation maps obtained when replacing quadratic with modulus operators in analyses reported in **Fig. 4** and **Fig. 6**. **Supp. Fig 4** reveals that the results were similar with modulus instead of quadratic operators. This certainly pertains to the low signal-to-noise ratio in fMRI, which prevents from discriminating between two close parametric regressors like the modulus and quadratic operator.

(Supplementary Fig 4)

activations along with activation peaks are similar to those in Fig. 6. Voxel-wise threshold $p < 0.05$ FWE-corrected for multiple comparisons over the frontal lobes (MNI coordinate $Y > 0$). Cluster-wise threshold $p < 0.05$.

We further mentioned in **Methods** the rationale for using quadratic rather than modulus operators:

(Methods, fMRI statistical analyses, 1-GLM, p. 29, fourth para.)

“To identify regions involved in choice computations, we used the quadratic rather than modulus operator of decision variable (*i.e.* $[(1 - \omega)(v_1 - v_2) + \omega(P_1 - P_2)]^2$) based on the following rationale. Previous results show fMRI activity to vary with the area under evidence accumulation traces in sequential sampling models of decision-making (Pisauro et al., 2017, Nature Comm.). As these traces form *on average* linear increases which slopes scale with the decision variable, the area under these traces then vary as an inverted quadratic function of the decision variable. Anyway, using a modulus operator provided virtually identical results (**Supplementary Fig. 4A**).”

(Methods, fMRI statistical analyses, 2-GLM, p. 30, second para.)

[...]. Similarly, quadratic expansions $(B_{ch} - B_{un})^2$ and $(v_{ch} - v_{un})^2$ modelling coding information or negative entropy associated with beliefs and utilities identically reflect the encoding of maximal beliefs B_{max} and utilities v_{max} or relative maximal beliefs $(B_{max} - B_{min})$ and utilities $(v_{max} - v_{min})$. Indeed, $X_{max} = \frac{1}{2}(1 + |X_1 - X_2|)$ and $X_{min} = \frac{1}{2}(1 - |X_1 - X_2|)$ when $X_1 + X_2 = 1$). In any case, such activations convey belief and utility information of one bandit compared to the other bandit, irrespective of chosen bandits. We used quadratic expansions as they better approximate entropy function than modulus operators (unsigned differences between beliefs and between utilities). Using modulus operators provided virtually identical results (**Supplementary Fig. 4B**).”

More minor points:

The description of the computational modeling under the Results section is currently quite dense and at times too technical. As this is a general interest journal you might consider presenting the relevant model parameters and various model instantiations in a more intuitive fashion (to be comprehensible by a non-expert). There is a bit of this in the text already but it tends to get lost in the technical details.

We followed your suggestion. We moved and rewrote the description of the different models in a separate section within Results entitled “Computational modelling”. The new section minimises mathematical details to improve readability and to focus on concepts. The section appears before presenting the model fitting and behavioural results as follows:

(Results, pp. 5-7)

“Computational modelling

Distortion models of human decision-making. To account for human suboptimal performances, we first considered the standard *distortion hypothesis* we referred to as model DIST (Savage, 1954; Hertwig and Erev, 2009; Kahneman and

Tversky, 1979; Zhang and Maloney, 2012). This model is identical to model OPT, except that choices maximize the *subjective* expected utility of choice options. Subjective utilities involve (1) subjective probabilities distorting objective probabilities that derive from marginalizing reward expectations over state beliefs; (2) subjective appetitive values distorting monetary values of proposed rewards. We used the standard distortion function encompassing all previously documented distortions (convex, concave, S-shape and inverted S-shape) on reward probabilities derived from state beliefs as well as on proposed rewards modelling subjective appetitive values (Zhang and Maloney, 2012) (**Methods**). Model DIST thus includes four additional free parameters characterizing such distortions with model OPT as a special case.

Suboptimal performances might also stem from reinforcement learning (RL) whereby the reward history of past choices biases current choices. To assess this effect, we considered the extension of model DIST (referred to as model DIST+RL) comprising an additional RL component based on the Rescorla & Wagner’s rule (Rescorla and Wagner, 1972) that alter the subjective expected utilities of bandits (**Methods**).

Mixture models of human decision-making. As outlined in the introduction, the distortion hypothesis might reflect that state beliefs and reward expectations are actually not integrated as posited in the expected utility theory (i.e. as in models OPT and DIST) (Beck et al., 2012). First, the distortion over monetary values of proposed rewards might actually reflect the recently proposed idea that option utilities are encoded in the brain according to a context-dependent divisive normalization mechanism (Louie et al., 2015; 2013), so that utilities of choices’ options are mutually dependent. Second, the notion of subjective probabilities distorting objective probabilities might actually reflect that no marginalization processes occur over abstract state beliefs such as “*bandit 1 (2 resp.) is rewarded with frequency q_1 (q_2 , resp.)*” (as in models OPT and DIST). Instead, beliefs might bear upon affective states such as “*bandit 1 (2 resp.) is the most frequently rewarded bandit*”, so that utilities and beliefs act as two normalized appetitive value components contributing to choices independently, i.e. additively. These two hypotheses, utility-normalization and affect-additivity, are tightly linked because: (1) affect-additivity requires making utilities commensurable to probabilistic beliefs, thereby leading to utility-normalization; (2) model OPT with normalized utilities and affective state beliefs is equivalent to maximizing a weighted sum of normalized utilities and affective state beliefs with specific weights denoted $1 - \omega_0$ and ω_0 , respectively (**Methods**).

Accordingly, we proposed the more general decision-making model (we refer to as model MIX) assuming that choices maximize a linear combination of normalized utilities and affective state beliefs with weighting parameters $1 - \omega$ and ω , respectively (**Methods**). If ω significantly differs from critical value ω_0 , choices are unlikely to derive from computing bandits’ expected normalized utilities based on marginalization processes over affective state beliefs. Moreover,

$\omega > \omega_0$ vs. $\omega < \omega_0$ implies more belief-based (risk-averse) vs. more utility-based (risk-prone) choices, respectively. Model MIX includes no distortion functions but, according to the context-dependent divisive normalization (Louie et al., 2015; 2013), utilities v_1, v_2 of choice options correspond to proposed rewards possibly altered by the reward history and normalized across the two choice options:

$$v_i = \frac{\varphi V_i^{proposed} + (1-\varphi)V_i^{RL}}{\sum_{i=1,2}(\varphi V_i^{proposed} + (1-\varphi)V_i^{RL})},$$

where φ is a free weighting parameter, $V_i^{proposed}$ the monetary value of proposed rewards and V_i^{RL} the RL-values associated with bandits i .”

For consistency, could the modeling results currently presented in Fig. 3d be shown in the same way as those in Fig. 2d (i.e. further divided up into VI+V2 high and low).

In the revised manuscript, we provided **Fig. 3D** in the same format as **Fig. 2D** (see below). For clarity, we only split in two graphs the median split between lower and higher values of total proposed rewards $V_1^{pr}+V_2^{pr}$. The graphs clearly show that there is neither main nor interaction effects of total proposed rewards $V_1^{pr}+V_2^{pr}$, but only two additive main effects of relative normalized utilities and state beliefs. We reported the statistics in the figure legend:

(Fig. 3D) see next page.

Fig. 3. Model MIX fit to human performances. **A**, proportions of choosing true best bandits (maximising reward frequencies \times proposed rewards) following reversals. **B**, proportions of choosing proposed rewards. In **A** & **B**, mean proportions over individuals (\pm s.e.m) are shown in the congruent, neutral and incongruent condition for participants (blue), and fitted model MIX (purple) simulations. See **Supplementary Fig. 2** for model RL fit. **C**, Parameter estimates of proposed rewards biases in the congruent, neutral and incongruent conditions in model MIX fitting human choices, $*p < 0.05$. These estimates are similar to true values (± 0.13 , see **Fig. 1B**) indicating that according to model MIX, participants adequately inferred state beliefs. **D**, factorial analysis of choice log-odds over trials sorted according to inferred state beliefs ($B_1 > B_2$ vs. $B_1 < B_2$), normalized utilities ($v_1 > v_2$ vs. $v_1 < v_2$) and total proposed rewards ($V_1^{pr} + V_2^{pr}$, median split: left vs. right panel) for model MIX simulations (purple) and human data (blue). Both model MIX and human data exhibited main effects of beliefs and utilities (all $F_s(1,21) > 116.9$, $p_s < 0.00001$), no main effects of total proposed rewards (both $F_s(1,21) < 1.2$, $p_s > 0.47$) and no interactions between these factors (all $F_s(1,21) < 3.3$, $p_s > 0.14$). Error bars are s.e.m. over participants. See **Supplementary Table 1** for model best-fitting parameters.

Thank you for your review.

Reviewer #3 (Remarks to the Author):

This paper seeks to distinguish between two explanations for suboptimal probabilistic decision-making: (1) the optimal combination of distorted beliefs (i.e. subjective probabilities and utilities) or (2) suboptimal combination of undistorted beliefs and utilities. Participants performed a reversal learning task. Two bandits had reward probabilities (0.8 and 0.2) that periodically reversed, together with randomized, explicitly cued reward magnitudes. Based on the implementation of multiple behavioral models, the authors argue the data are best explained by a model that additively (and thus suboptimally) combines two attributes: "affective beliefs" (related to each bandit's probability of being rewarded), and "utility" (related to the bandit's face value and reward history). Model-based fMRI analyses showed that dmPFC activity scaled with the difficulty of decisions, whereas vmPFC activity scaled with the weight of evidence favoring a given bandit.

This paper reflects a thorough and thoughtful approach to building and testing computational models of decision making, and the fMRI results point to interesting potential distinctions between and within dmPFC and vmPFC. However, I felt the paper also had some significant weaknesses relating to the clarity of presentation and the grounding of conclusions in evidence, which made it difficult to be confident about its scientific contribution.

I. Comments about the computational modeling.

1. Figure 1 shows that behavior differs qualitatively from an optimal model, in that performance was worst in the "incongruent" condition. Figures 2 and 3 then show that the "DIST" and "MIX" models can reproduce this pattern. This is confusing (and perhaps misleading) given that a parameterized version of the optimal model could also fit the same pattern (Fig. S2). So, does the good visual match between model and data in Fig. 2A and 3A have anything to do with the unique features of the DIST and MIX models?

The unique feature of models DIST and MIX is to replicate **both** the variation of subjects' performances across conditions (congruent, neutral and incongruent, Figs. 2A and 3A) **and** the variation of subjects' performances according to bandits' proposed rewards (Figs. 2B and 3B). By contrast, the parameterized version of model OPT succeeds to fit the variation of subjects' performances along trial series across conditions (Fig. S2A) *only by failing* to fit the variation of subjects' performances according to bandits' proposed rewards (Fig. S2B)(note that the converse failing effect might have been found). We agree that the main text was not enough clear regarding this point. To clarify, we highlighted the point in the revised main text and supplementary Fig. 2 as follows:

(Main text, Results, Behavioural results, p. 7, second para.):

“Importantly, model DIST simulations *reproduced* participants' choices in every condition *both* in choosing true best bandits along trial series following reversals (**Fig. 2A**) *and* in bandits' choices according to proposed rewards (**Fig. 2B**).

Moreover, model DIST fitted participants' behaviour better than all its reduced models comprising fewer free parameters (including model OPT), even when penalizing for model complexity (Bayesian Information Criteria, paired T-tests, DIST vs. reduced models: all $T_s(21) > 8.0$, $p_s < 10^{-7}$) (Methods). **Reduced models with no distortions failed to reproduce participants' choices across all conditions both along trial series following reversals and according to proposed rewards (Supplementary Fig. 2A,B).**"

(Main text, Results, Behavioural Results, p. 9, first para.):

We fitted model MIX to participants' choices. As model DIST, model MIX simulations *reproduced* participants' performances according to external reward contingencies in every condition *both along trial series and according to proposed rewards* (Fig. 3A,B), while the reduced pure RL model failed especially in the congruent and incongruent condition (Supplementary Fig. 2).

(Supplementary Fig. 2, legend)

"Sup. Fig. 2 (related to Figs. 2 & 3). Simulations of fitted models RL (with proposed rewards) and OPT (parameterized).

Proportions of choosing true best bandits (maximizing reward frequency \times proposed reward monetary value) according to trial number after reversals (A) and choice proportion of proposed rewards (B) for participants (blue, same data as in Fig. 1), model OPT (parameterized, grey) and model RL (including proposed rewards, red). Data are shown separately for the conditions congruent, neutral and incongruent. Parameterized model OPT corresponds to model DIST with no distortions on reward probabilities and values, and was fitted to human data with the same free parameters except the four parameters specifying distortions (see Supplementary Table S2). Model RL shown here included four free parameters: inverse temperature β , lapse rate ϵ , learning rate α , and contribution of proposed rewards to RL values ϕ at decision time. Error bars are s.e.m. across participants. Note that none of the models succeed in reproducing human data both *along trial series following reversals and according to proposed rewards.*"

2. Around line 159, the authors are making what I think boils down to a fairly straightforward point: the difference in probabilities should theoretically matter more when the face-values of the bandits are high, and this qualitative pattern was absent in the behavioral data. The description is a little hard to follow, in part because of the run-on sentence that extends from line 164 to 170. More substantively, I didn't understand why it was necessary to binarize the data with a median split in order to perform this analysis. I would have thought a continuous logistic regression would be more straightforward (and in either case, it would be appropriate to include a statistical test of these claims). I also found the visualizations in Fig. 2D-E quite hard to interpret. In 2D, the y-axis scaling makes it hard to verify the effect of relative value and the lack of an interaction in the behavioral data.

A continuous logistic regression analysis based on model variables would be redundant with model fitting. For this reason, our original paper reports the logistic regression

analysis of participants' choices that you have in mind, but according to protocol parameters rather than model variables resulting from model fitting (see **Supplementary Fig. 1**). The corresponding results clearly show that participants' choices were independent of expected value terms (reward frequencies x proposed rewards). Such regression analyses, however, can only indicate that a model poorly fit the data without constituting a falsification test (see Palminteri et al., 2017, *Trends Cog. Sci.*: “the importance of falsification in computational cognitive modelling”).

By contrast, the factorial analyses reported in the main text directly tested the core feature of model DIST/DIST+RL (and its reduced models), namely the product between subjective probabilities and appetitive values of proposed rewards in choice log-odds. We acknowledge that our description of this analysis was rather clumsy. In the revised text, we clarified the description as follows:

(Results, Behavioural results, p. 8, first para.)

“Equation (E2) predicts log-odds ($\log \frac{p_1}{p_2}$) to exhibit an interaction effect between total appetitive values $\tilde{V}_1 + \tilde{V}_2$ and relative probabilities $\tilde{P}_1 - \tilde{P}_2$, along with an additive effect of $(\tilde{V}_1 - \tilde{V}_2)(\tilde{P}_1 + \tilde{P}_2)$. To test this prediction, we entered choice frequency log-odds from participants and model DIST simulations in a 2x2x2 factorial analysis including as within-subject factors: total values ($\tilde{V}_1 + \tilde{V}_2$) (median split), relative probabilities $\tilde{P}_1 - \tilde{P}_2$ ($\tilde{P}_1 < \tilde{P}_2$ vs. $\tilde{P}_1 > \tilde{P}_2$) and relative values $\tilde{V}_1 - \tilde{V}_2$ ($\tilde{V}_1 < \tilde{V}_2$ vs. $\tilde{V}_1 > \tilde{V}_2$; due the protocol reversal structure, $\tilde{P}_1 + \tilde{P}_2$ was constant across these eight cells and consequently not included in the analysis). As shown in **Fig 2D**, these log-odds exhibited additive main effects associated with relative probabilities and values but unlike model DIST, participants' log-odds exhibited no interactions between total appetitive values and relative probabilities.”

As you suggest, furthermore, we provided the statistical tests supporting the main text in **Fig. 2, legend**, and we improved the readability of panels **D-E** in **Fig. 2**:

(**Fig. 2**)

analysis. **le 2** for model best-fitting parameters and **Supplementary Fig. 1** for model-free **Fig. 2**

We also did the same for Fig. 3 regarding model MIX:
(Fig. 3)

Fig. 3. Model MIX fit to human performances. **A**, proportions of choosing true best bandits (maximising reward frequencies \times proposed rewards) following reversals. **B**, proportions of choosing proposed rewards. In A & B, mean proportions over individuals (\pm s.e.m) are shown in the congruent, neutral and incongruent condition for participants (blue), and fitted model MIX (purple) simulations. See **Supplementary Fig. 2** for model RL fit. **C**, Parameter estimates of proposed rewards biases in the congruent, neutral and incongruent conditions in model MIX fitting human choices, $*p < 0.05$. These estimates are similar to true values (± 0.13 , see **Fig. 1B**) indicating that according to model MIX, participants adequately inferred state beliefs. **D**, factorial analysis of choice log-odds over trials sorted according to inferred state beliefs ($B_1 > B_2$ vs. $B_1 < B_2$), normalized utilities ($v_1 > v_2$ vs. $v_1 < v_2$) and total proposed rewards ($V_1^{pr} + V_2^{pr}$, median split: left vs. right panel) for model MIX simulations (purple) and human data (blue). Both model MIX and human data exhibited main effects of beliefs and utilities (all $F_s(1,21) > 116.9$, $p_s < 0.00001$), no main effects of total proposed rewards (both $F_s(1,21) < 1.2$, $p_s > 0.47$) and no interactions between these factors (all $F_s(1,21) < 3.3$, $p_s > 0.14$). Error bars are s.e.m. over participants. See **Supplementary Table 1** for model best-fitting parameters.

3. I had some trouble following the motivation for the various features of the MIX model (which is the model the authors ultimately favor). The model has quite a number of parts, including (1) combining the face-values of the bandits with RL-based values, (2) normalizing the resulting utilities across the choice set, and (3) further combining the result with an estimate of affective beliefs.

- I didn't see a strong justification for the normalization step. The closest the authors come to explaining this is on line 202, where they invite the reader to work through a substitution in Equation 2, but this falls short of a satisfactory explanation. None of the reduced models seem to address whether normalization improves the model fit.

Following the suggestion of other reviewers for revising our manuscript, we moved and rewrote the description of the different models in a separate section within Results entitled "Computational modelling". The new section minimizes mathematical details to improve readability and to focus on concepts. In particular, the section clarifies the rationales supporting the normalization step in model MIX as follow:

(Results, Computational modelling, pp. 5-7)

“Mixture models of human decision-making. As outlined in the introduction, the distortion hypothesis might reflect that state beliefs and reward expectations are actually not integrated as posited in the expected utility theory (i.e. as in models OPT and DIST) (Beck et al., 2012). First, the distortion over monetary values of proposed rewards might actually reflect the recently proposed idea that option utilities are encoded in the brain according to a context-dependent divisive normalization mechanism (Louie et al., 2015; 2013), so that utilities of choices’ options are mutually dependent. Second, the notion of subjective probabilities distorting objective probabilities might actually reflect that no marginalization processes occur over abstract state beliefs such as “bandit 1 (2 resp.) is rewarded with frequency q_1 (q_2 , resp.)” (as in models OPT and DIST). Instead, beliefs might bear upon affective states such as “bandit 1 (2 resp.) is the most frequently rewarded bandit”, so that utilities and beliefs act as two normalized appetitive value components contributing to choices independently, i.e. additively. These two hypotheses, utility-normalization and affect-additivity, are tightly linked because: (1) affect-additivity requires making utilities commensurable to probabilistic beliefs, thereby leading to utility-normalization; (2) model OPT with normalized utilities and affective state beliefs is equivalent to maximizing a weighted sum of normalized utilities and affective state beliefs with specific weights denoted $1 - \omega_0$ and ω_0 , respectively (Methods).

Accordingly, we proposed the more general decision-making model (we refer to as model MIX) assuming that choices maximize a linear combination of normalized utilities and affective state beliefs with weighting parameters $1 - \omega$ and ω , respectively (Methods). If ω significantly differs from critical value ω_0 , choices are unlikely to derive from computing bandits’ expected normalized utilities based on marginalization processes over affective state beliefs. [...]

We hope this clarifies why model MIX includes normalized utilities. Without normalization, a linear model would be odd, as utilities are compared to affective state beliefs without being commensurable to state beliefs (the latter being normalized). Consequently, even though such a linear model remains close to original model MIX (weight parameter ω might correct the lack of normalization), one expects model MIX to remain a better predictor of human performance with than without normalizations. We carried out this analysis (the two models comprise the same parameters) and found that as expected, model MIX remains the best predictor of participants’ choices ($BIC_{MIX} < BIC_{noNorm}$; $T=1.65$, $p=0.05$, one-tailed). As expected also, fitted parameter ω for the noNorm model increased to 0.94 for correcting the lack of normalization. This result provides additional evidence supporting the normalization step in model MIX. We reported this result in the revised paper:

(Results, Behavioural results, p. 9, first para.)

“As expected additionally, removing the divisive normalization over utilities in model MIX while keeping the same variables (affective beliefs, proposed rewards and RL values) also degraded the fit ($BIC_{MIX} < BIC_{NoNorm}$, $T(21)=1.65$, $p=0.05$, one-tailed).”

Note that the result is consistent with previous studies (Louie et al., 2015; 2013).

- In general, the model would be better justified if it were tested and validated across multiple data sets. Because the model is both novel and fairly elaborate, there would seem to be some risk of its being tailored to the idiosyncrasies of the data set on which it was developed. This is particularly important because the MIX model is reified and used as the premise for the fMRI analyses.

We know at least three published papers reporting as an incidental finding from BIC measures, that a linear combination of reward values and probabilities fitted human/monkey behavioural data better than the computation of expected utilities (as in model OPT). One monkey study used a protocol similar to our neutral condition (Donahue et al., 1013 *Neuron*, 80). Two human studies (Scholl et al., 2015, *J. Neurosci.* 35; Faulkner et al., 2016, *Intl. J. Neuropsychopharma.* 20) used a protocol where bandits' reward probabilities were explicitly delivered in every trial rather than inferred as in our present protocol. However, none of these studies addressed the issue of how reward values and probabilities are combined to guide choices: these studies neither investigated standard distortion models as model DIST nor the formation of state beliefs. Moreover, none investigated possible contribution of RL-values to choice suboptimality. Accordingly, none could conclude in favour of the linear model. We precisely elaborated our protocol to disentangle these models. Conversely, these previous findings provide evidence that our results are unlikely to stem from idiosyncrasies of the present protocol. To respond to your comment, we briefly mentioned these previous findings in our discussion:

(Discussion, p. 14, second para.)

“Additionally, previous studies involving various protocols (Donahue et al., 2013, Scholl et al., 2015, Faulkner et al., 2016) incidentally found that based on BIC measures, decision variables linearly combining reward values and probabilities fitted human/monkey behavioural data better than computing expected utilities (as in model OPT). Although none of these studies control for standard distortion and/or belief formation models along with possible RL influences on choices, these previous incidental findings provide evidence that the proposed model is likely to extend to various decision situations.”

- It's an interesting proposal that people might treat reward probability and magnitude as separate, additive attributes (rather than properly multiplying them). The distinction is muddled, however, since the "utility" attribute, which encodes reward magnitude, seems likely to also contain information about probability via the RL mechanism. This reduces my confidence that the fMRI analyses successfully dissociated the factors of utility and probabilistic beliefs.

As you note, the ability of GLM analyses to dissociate activations associated with two regressors depends upon the shared variance between these regressors: the shared variance belongs to the residuals of GLMs and can be indifferently attributed to any of these regressors, thereby degrading potential dissociations. In response to your comment, we reported in the revised manuscript the shared variance between regressors utility and

beliefs, *i.e.* the coefficient of determination R^2 (R squared) between these factors. Between linear regressors coding for chosen-relative-to-unchosen utilities and beliefs, R^2 was equal to 0.17 (mean over subjects). This indicates that in addition to dissociated dmPFC activations between these regressors, the dmPFC might exhibit additional activations confounding both factors representing *at most* 17% of the formers. Thus, *at least* 83% of dmPFC activations (potentially attributable to chosen-relative-to-unchosen utilities or beliefs) reflected a “true” dissociation between these factors.

Moreover, R^2 was equal to 0.02 between quadratic regressors coding for (unsigned) relative utilities and beliefs. Accordingly, in addition to dmPFC activations associated with relative utilities, the dmPFC might exhibit additional activations associated with relative beliefs representing *at most* 2% of the formers. Conversely, in addition to vmPFC activations associated with relative beliefs, the vmPFC might exhibit additional activations associated with relative utilities and representing *at most* 2% of the formers. Thus *at least* 98% of vmPFC and dmPFC activations (potentially attributable to relative utilities or beliefs) reflected a “true” regional dissociation between these factors. With respect to fMRI standards, these values are considered as very strong dissociations.

Note that as mentioned in the original manuscript, we precisely built our protocol with three conditions (congruent, neutral, incongruent) to obtain such strong dissociations between beliefs and RL-values values, and consequently utilities (Results, behavioural protocol, first para.: “To properly dissociate belief probabilistic inferences from reinforcement learning processes, the protocol included two additional condition [...]. Thus, the protocol properly dissociated belief inferences from reinforcement learning processes over trials [...]. Overall, the protocol was as simple as possible while inducing the formation of beliefs dissociable from reinforcement learning processes”). To clarify this point, we reported the coefficients of determination R^2 above in the revised manuscript:

(Methods, fMRI statistical analyses, p. 28, second para.)

“We ensured that Variance Inflation Factor assessing collinearity between all parametric modulators were small enough (Hair et al., 2006) **to allow proper dissociations. Thus, shared variances (coefficient of determination R^2) between state beliefs and normalized utilities were equal to 0.17 and 0.02 for linear and quadratic expansions, respectively**”.

4. Some of the modeling is presented in a needlessly verbose manner that obfuscates the main points. For example, the standard softmax choice function is written as a system of 2 or 3 equations (repeated on lines 671, 745, and 757) where one simpler equation would more clearly express the same meaning.

Following your suggestion, we replaced equations lines 671, 745 and 757 in the original manuscript with single equations using the standard softmax sigmoid function (see revised text, Methods, computational modelling, pp. 21, 24 and 24).

Other related points:

- I didn't follow the derivation of the special-case value for the weighting parameter ω_0 , either in the text on line 233 or in the methods on line 780. Please try to clarify.

Sorry for the lack of clarity. To clarify, we replaced the sentence “In that case, the linear combination becomes equivalent to the model assuming that choices maximize the *expected* normalized utility by marginalizing reward expectations over state beliefs B_1 and B_2 (see **Methods**)” (line 233 in the original manuscript) with the following one:

(Results, computational modelling, mixture models of human decision-making, p. 6, third para.)

“These two hypotheses, utility-normalization and affect-additivity, are tightly linked because: (1) affect-additivity requires making utilities commensurable to probabilistic beliefs, thereby leading to utility-normalization; (2) model OPT with normalized utilities and affective state beliefs is equivalent to maximising a weighted sum of normalized utilities and affective state beliefs with specific weights denoted $1 - \omega_0$ and ω_0 , respectively (**Methods**).”

In the Methods, similarly, we replaced the original sentence “In that case, model MIX is equivalent to the model maximizing the expected normalized utility” with the following one:

(Methods, computational modelling, alternative model MIX, p. 25, fourth para.)

“*Special case* $\omega = \omega_0$. Model MIX further features a special case when $\omega = \omega_0 = \frac{q_M - q_m}{q_M + q_m + q_M - q_m}$. In that case, model MIX is equivalent to **model OPT with normalized utilities**.”

We hope this clarifies.

- I also didn't understand the point being made in lines 796-806.

The point specifically refers to the paper from Lebreton et al. (2015, Nature Neurosci.). This paper shows that subjects' confidence in their choices, which correlates vmPFC activations, negatively scales with the variance of the distribution coding for the variable guiding choices. Here we show that the quadratic expansion of relative state beliefs, which correlated with vmPFC activations, matches this notion of confidence. To clarify the point in the revised text, we rewrote this paragraph as follows:

(Methods, Computational modelling: alternative model MIX, p. 26, second para.):

“*Relative affective state beliefs and confidence in choosing the more frequently rewarded option.* Lebreton et al. (2015) show that subjects' confidence in their choices, which correlates with vmPFC activations, negatively scales with the variance of the distribution coding for the variable guiding choices. Here we show that the unsigned difference between state beliefs, which correlated with vmPFC activations, matches this notion of confidence: namely, the quadratic expansion of relative affective state beliefs negatively scales with the variance between effective reward probabilities associated with bandits. Consider bandit 1 and its effective reward probability P_1 , [...]”

- it's unclear what the variable "r" refers to (line 662), is it the set of both nominal reward amounts $\{V_1, V_2\}$? It's also unclear how r was used as a modulator in GLM 1 (line 887).

You are right: “*r*” denotes {*V*₁, *V*₂}. We apologize to have omitted to mention it. We added it in the revised text (Methods). In GLM#1 as you point out, we also incorrectly used “*r*” to refer to feedback values. In the revised text, we therefore replaced this “*r*” with “**feedback values**”.

II. Comments about the fMRI analyses and results.

5. It is not clear what method was used to correct for multiple comparisons. There is mention of a cluster-size threshold (Fig. 4 legend and line 879), and it sounds like this was applied subsequent to an unspecified correction at the voxel level. And why was a more stringent cluster-size threshold used for the analyses restricted to frontal cortex ($k > 40$) and a more lenient threshold use for the whole-brain analyses ($k > 10$)? (line 895)

We used the standard SPM method to correct for multiple comparisons: in all analyses, voxel-wise activation thresholds were set at $p < 0.05$, FWE-corrected for multiple comparisons. The corresponding T-values depend upon the number of comparisons, i.e. upon the search volume: namely, the whole brain in the first analysis (Fig. 4; GLM#1; Sup. Table 3); the frontal lobes (MNI coordinates $Y > 0$) in the subsequent analyses (Figs. 5, 6 & 7; GLM#2 & GLM#3; Sup. Tables 4 & 5). Following the standard SPM method, these FWE-corrected cluster-wise thresholds were applied along with cluster-wise activation thresholds also set to $p < 0.05$. The latter corresponds to a number of voxels, which depends upon voxel-wise T-values and therefore, search volumes. In the first analysis, (Fig. 4; GLM#1; Sup. Table 3), this cluster size threshold was $k > 10$; in the subsequent analyses (Figs. 5, 6 & 7; GLM#2 & GLM#3; Sup. Tables 4 & 5). this cluster size threshold was $k > 40$. This explains why the cluster-wise thresholds differ across the whole brain analysis and the subsequent analysis restricted to the frontal cortex. In the revised manuscript, we made the following clarifications:

(Methods, fMRI statistical analyses, p. 29, second para.)

“**Following the standard SPM method**, second-level parametric maps were then obtained for each contrast over the group of participants with significance voxel-wise threshold set at $p < 0.05$ corrected for family-wise errors for multiple comparison over the search volumes **and with significance cluster-wise threshold set at $p < 0.05$** (see below)”.

(Methods, fMRI statistical analyses, 1-GLM, p. 29, third para.):

“This analysis was performed over the whole brain (**cluster-wise threshold set at $p < 0.05$: $k > 10$ voxels**).

6. The authors state they used "post-hoc analyzes removing selection biases" (line 298), and add in the methods section that this involved a leave-one-out procedure (line 881), but more detail about this would be needed to understand how circularity was avoided in extracting data from the significant clusters.

We used the standard leave-one-out method described in (Kriegeskorte et al., 2009) . Following your suggestion, we added the following description in Methods:

(Methods, fMRI statistical analyses, p. 29, second para.):

“We removed selection biases from all post-hoc analyses performed from activation clusters using a leave-one-out procedure (Kriegeskorte et al., 2009): for every GLM, the partial correlation coefficients (betas) of each participant were averaged over activation clusters identified in the N-1 remaining participants (using the significance thresholds indicated above); these coefficients were then entered in post-hoc analyses across the sample of N participants”

7. The fMRI analysis introduced starting at line 303 is not described very clearly. I think this passage could be made a lot more succinct. As best I can tell, the prediction is less brain activity when either beliefs or utilities favor the chosen option by a larger margin. The result is described as "the addition of the two negative linear effects" (line 318), which confused me because the authors talk about this result as if it were a conjunction of effects. To support the conclusions, it would be necessary to report the intersection of the two thresholded maps, not their additive combination.

We actually carried out the conjunction/intersection analysis you suggest. In the revised manuscript, we clarified this point as follow:

(Results, fMRI activations, p. 11, first para.)

“A conjunction (intersection) analysis revealed only one PFC region exhibiting the addition of the two predicted negative linear effects, namely the dmPFC region reported above (along with right premotor and insular activations)”

Following your suggestion, we also made the description of this analysis more succinct (moving technical details to Methods) as follows:

(Results, fMRI activations, p. 11, first para.)

“Consequently, model MIX specifically predicts these regions to exhibit two additive decreasing effects as reflecting the linear combination of affective state beliefs and normalized utilities guiding choices: activations should decrease when affective state beliefs or normalized utilities associated with chosen bandits increase, with no interactions between these effects (*i.e.* when B_{chosen} and v_{chosen} or equivalently, $B_{\text{chosen}} - B_{\text{unchosen}}$ and $v_{\text{chosen}} - v_{\text{unchosen}}$ increase, as these variables are normalized)(Methods).”

(Methods, fMRI statistical analyses, 2-GLM, p. 30, second para.)

“Note that as utilities and beliefs are normalized variables, one may indifferently use regressors $(B_{\text{ch}} - B_{\text{un}})$, $(v_{\text{ch}} - v_{\text{un}})$ or B_{ch} , v_{ch} . Indeed, $B_{\text{chosen}} = \frac{1}{2}(1 + B_{\text{chosen}} - B_{\text{unchosen}})$ and $v_{\text{chosen}} = \frac{1}{2}(1 + v_{\text{chosen}} - v_{\text{unchosen}})$.”

8. The next fMRI analysis looks for activity that increases with larger differences in either beliefs or utilities. It's an interesting finding that vmPFC shows more activity when there is more confidence about which bandit has the higher reward probability. However, the dmPFC data are presented confusingly. For positive values of chosen-minus-unchosen utility (which is presumably the range that contains a majority of the trials), the same dmPFC region is shown to be positively modulated in Fig. 6C but negatively modulated in Fig. 5B. This is because Fig. 5B subtracts the quadratic trend from the data, whereas Fig. 6C subtracts the linear trend. I think it would be clearer to directly

show the true pattern of data for this cluster.

Figs. 5 and 6 follow the logic of our GLM analysis. Our GLM analysis shows that both x and x^2 ($x=v_{\text{chosen}}-v_{\text{unchosen}}$, v denoting normalized utilities) can be linearly decoded from dmPFC activations: *i.e.* dmPFC activations encode both chosen-relative-to-unchosen utilities (reflecting selection processes) and unsigned relative utilities (reflecting utility comparison). In contrast, the relative value of partial regression coefficients (beta) between x and x^2 has no relevance in this analysis. Consequently, the overall variation of dmPFC activations, which combines linear and quadratic expansions of x , has no clear meanings and may even mislead the reader regarding the true meaning of this analysis. For this reason, we preferred to keep Figs 5 & 6 as presented in the original manuscript.

Note that for clarifying this results description, we shortened the corresponding paragraph by moving the technical details to Methods:

(Results, fMRI activations, p. 11, last para.)

“To further investigate PFC regions encoding affective state beliefs and normalized utilities irrespective of choice computations, we followed the method described in (Duverne and Koechlin, 2017). Accordingly, we included in the preceding regression analysis two additional quadratic regressors $(B_{\text{chosen}} - B_{\text{unchosen}})^2$ and $(v_{\text{chosen}} - v_{\text{unchosen}})^2$ (equal to $(v_1 - v_2)^2$ and $(B_1 - B_2)^2$). The rationale is that while the linear regressors associated with $(B_{\text{chosen}} - B_{\text{unchosen}})$ and $(v_{\text{chosen}} - v_{\text{unchosen}})$ capture and factor out the activations reflecting choice computations, these quadratic regressors capture the residual activations coding for choice-relevant information, irrespective of actual choices. Indeed, regions coding for belief (utility, resp.) information are predicted to exhibit activations *decreasing with belief entropy (utility entropy, resp.)*, *i.e.* varying as a positive quadratic function of $B_1 - B_2$ ($v_1 - v_2$, resp.). Such activations simply convey belief and utility information of one bandit compared to the other bandit, irrespective of chosen bandits (**Methods**).

(Methods, fMRI statistical analyses, 2-GLM, p. 30, second para.)

“[...] Similarly, quadratic expansions $(B_{\text{ch}} - B_{\text{un}})^2$ and $(v_{\text{ch}} - v_{\text{un}})^2$ modelling coding information or negative entropy associated with beliefs and utilities identically reflect the encoding of maximal beliefs B_{max} and utilities v_{max} or relative maximal beliefs $(B_{\text{max}} - B_{\text{min}})$ and utilities $(v_{\text{max}} - v_{\text{min}})$. Indeed, $X_{\text{max}} = \frac{1}{2}(1 + |X_1 - X_2|)$ and $X_{\text{min}} = \frac{1}{2}(1 - |X_1 - X_2|)$ when $X_1 + X_2 = 1$. In any case, such activations convey belief and utility information of one bandit compared to the other bandit, irrespective of chosen bandits. We used quadratic expansions as they better approximate entropy function than modulus operators (unsigned differences between beliefs and between utilities). Using modulus operators provided virtually identical results (**Supplementary Fig. 4B**).”

9. The authors argue that the fMRI results support the validity of the MIX model (e.g. line 501). This claim might need to be softened, as it isn't supported by any direct comparison between different models' ability to fit the fMRI data (and it seems plausible that other models might produce correlated estimates of decision difficulty and evidence strength).

The statement that the DIST model yielded no significant fMRI effects (line 368) is relevant, but not a strong basis for this claim.

As you suggest, we softened the statement line 501 “Furthermore, our results provide evidence that the dmPFC is involved in computing choices by combining normalized utilities and affective state beliefs, as model MIX predicts”. We replaced it with the following statement:

(Discussion, p. 15, second para.):

“Second, our results provide evidence that the dmPFC is involved in computing choices by linearly combining normalized utilities and affective state beliefs”.

10. On line 494, the authors conclude that dmPFC builds a representation of each option's utility in a manner prescribed by the MIX model. But how can the alternative possibility be excluded that dmPFC merely responds to the difficulty of the comparison between values computed elsewhere?

The difficulty of comparing utilities varies with the similarity between utilities and with Reaction Times (RTs). Two pieces of evidence reported in the paper rule out the possibility that dmPFC activations only reflects the difficulty of comparing utilities: (1) dmPFC activations remained associated with options' utilities, even when RTs were factored out; (2) dmPFC activations exhibited a quadratic effect related to relative utilities (maximal activations when utility values maximally differ) along with a linear effect (maximal activations when unchosen utilities maximally overcome chosen utilities) Consequently, dmPFC activations cannot only reflect the difficulty of comparing utilities. In the revised text, we clarified this point as follows:

(Discussion, P. 15, second para.):

“Over and above reaction times, first, normalized utilities were associated with dmPFC activations, [...]”

Point (2) is already mentioned later in the same paragraph.

III. General and minor comments.

11. The paper needs general proofreading for grammar and clarity.

We clarified and corrected the text as far as possible.

12. The introduction assumes background knowledge and familiarity with the task structure that isn't provided. For instance, lines 35-57 introduce a hypothesis about how state beliefs interact with reward expectations. But this hypothesis is specific to a certain kind of choice context that hasn't yet been introduced. This issue also impacts the generality of the paper's conclusions. While it may be that state beliefs exert a direct effect on choice during reversal learning, the connection between latent states and rewards in other situations might not be so straightforward. It's not at all clear how this model would provide a general account of risk aversion (line 470).

As you mention, our study addresses the issue of state beliefs *about reward contingencies*. We agree that in the original manuscript, the introduction was not clear

enough about this point. In the revised manuscript, we clarified the introduction as follow:

(Introduction , P. 3, first and second para.):

“Everyday life features uncertain and changing situations associated with distinct reward contingencies. In such environments, optimal adaptive behaviour requires detecting changes in external situations, which relies on making probabilistic inferences about the latent causes or hidden states generating the external contingencies the agent experiences. Previous studies show that humans make such inferences, *i.e.* they develop *state beliefs* to guide their behaviour in uncertain and changing environments (Collins and Koechlin, 2012; Collins and Frank, 2013; Gershman et al., 2010; Redish et al., 2007). More specifically, the prefrontal cortex (PFC) that subserves **reward-based** decision-making is involved in inferring state beliefs **about how reward contingencies map onto choice options** (Donoso et al., 2014; Collins et al., 2014; Koechlin, 2014; Schuck et al., 2016). Optimal decision-making for driving behaviour then requires integrating **these** state beliefs and reward expectations **through probabilistic marginalization processes** (Beck et al., 2011). **This integration is required to derive reward probabilities associated with choice options** and to choose the option maximizing the expected utility (Neumann and Morgenstern, 1947). Consistently, PFC regions involved in inferring these state beliefs also exhibit activations associated with reward expectations (Behrens et al., 2007; Boorman et al., 2009; Damasio et al., 1996; Hunt et al., 2012; Kennerley et al., 2011; Knutson et al., 2005).

However, human choices often differ from optimal choices systematically (Kahneman and Tversky, 1979) raising the open issue how the PFC combines **these** state beliefs and reward expectations to drive behaviour. A common hypothesis is that these quantities are integrated as posited in the expected utility theory but choice computations **derive** from distorted representations of **reward** probabilities, usually named *subjective* probabilities (Savage, 1954; Hertwig and Erev, 2009; Johnson and Busemeyer, 2010; Kahneman and Tversky, 1979; Zhang and Maloney, 2012). Yet the origin of subjective probability remains unclear. As marginalization processes are complex cross-product processes (Beck et al., 2011), the notion of subjective probability might then reflect that state beliefs and reward expectations are actually combined in a suboptimal way at variance with the expected utility theory (Beck et al., 2012). Thus, an alternative plausible hypothesis is that state beliefs **about reward contingencies are processed as an additional value component that contribute to choices independently of reward expectations** rather than through marginalization processes: *i.e.* state beliefs **about reward contingencies** act in decision-making as affective values that combine linearly with the appetitive value of rewards expectations.

Here we address this open issue using computational modelling and functional magnetic resonance imaging (fMRI). ~~We scanned 22 participants while they were making successive choices between two visually presented one armed bandits. In every trial, each bandit proposed a potential monetary reward. Bandits' reward~~

frequencies were constant across trials but reversed episodically, so that state beliefs corresponded to the probabilistic mapping between bandits and reward frequencies. We confirm here that participants **make decisions** “as if” they marginalise reward expectations over state beliefs and compute choices based on distorted subjective probabilities. Using a model falsification approach (Palminteri et al., 2017), however, we show that participants’ performance vary with these subjective probabilities in a way contradicting this theoretical construct. We then provide evidence that participants’ choices actually derive from the independent contribution of state beliefs regarding the most frequently rewarded option and reward expectations based on an efficient coding mechanism of context-dependent value normalization (Carandini and Heeger, 2012; Louie et al., 2015; 2013). We identify the PFC regions involved in this decision process combining linearly these state beliefs and reward expectations which at variance with the standard expected utility theory, results in (1) the mutual dependence of option utilities and (2) the processing of state beliefs as affective values rather than probability measures in decision-making.”

Following your suggestion, we also amended the statement line 470 in the original manuscript as follow:

(Discussion, p. 14, second para.)

“The dominant contribution of the latter over the former also **reflects well-known risk-aversion effects** in human **reward-based** decision-making (Bernoulli, 1954; Kahneman and Tversky, 1979).”

And we altered the abstract as follow:

(abstract)

“This neural mechanism accounts for ~~the well-known phenomenon of~~ risk aversion **effects** in human decision-making”

13. Some important information about the methods would be worth elevating to the main text:

- The task timing isn't clear from Fig. 1 legend. It gives a range for the "stimulus onset asynchrony" without saying which two stimulus onsets it applies to.

Thank you for raising the confusion. We replaced "stimulus onset asynchronies" in **Fig. 1**, legend, with "**response-feedback onset asynchronies**"

- Please clarify in the main text that the 3 covariance conditions were administered on separate days of scanning.

We clarified as follow (Results section, behavioural protocol, p. 4, last para.):

“To properly dissociate belief probabilistic inferences from reinforcement learning processes, the protocol included two additional conditions (**administered in separate days**):”

14. It would be helpful to clarify some aspects of the instructions that were given to

participants:

- Were they explicitly informed that the reward probabilities for the two bandits were 80% and 20%?

- Were they given any information about whether/how the task might differ across the 3 sessions?

Before each session, participants were simply instructed that bandits had distinct reward frequencies and that these frequencies reversed episodically. The exact probabilities as well as differences across sessions were not given explicitly but as mentioned in the original manuscript, participants were trained on the protocol before each session to learn reward probabilities, reversals and information conveyed by proposed rewards about external state contingencies. Following your suggestion, we added the following statement in the Methods section:

(Methods, Behavioural Protocol, p. 19, second para.):

”Before each session, participants were only instructed that bandits had distinct reward frequencies and these frequencies reversed episodically.”

Thank you for your comments.

Reviewers' comments:

Reviewer #1 (Remarks to the Author):

Rouault and colleagues investigated human decision making while outcome probability reversed and outcome magnitude fluctuated. Please refer to my original review for a full description of the study. The bottom line is that Rouault et al propose that choice results from the additive contributions of state beliefs and utility values, in contrast to the expected utility theory that integrates utilities over state beliefs to derive subjective probabilities for choice computation.

The study makes a major contribution to the literature and offers an interesting perspective on "irrational" decision making. It is also a great example of how model falsification approaches can and should be used. The authors had done an overall excellent job in addressing all concerns raised in my previous review, except one point which I will clarify below:

In my previous review I mentioned concerns relating to the fact that in the present study the beliefs states relate to reward probabilities (and hence are often termed affective belief states by the authors themselves) I think the authors have misunderstood my previous point, so I will reiterate here: As per equation for $B1(t|r)$ listed on page 20 in the paper, the 'belief' that a particular state might be true is related to the value difference for that state compared to other states. Doesn't that mean that if this state is particularly worth choosing (because $V1-V2$ is big), then the belief in state 1 goes up? Belief states, however, can signify many more dimensions of the task. Based on *sensory* evidence, rather than reward evidence, a belief state could for instance reflect that a presented stimulus is more likely to be green than to be red, while green is generally the less rewarding option. Another issue to consider is that belief states do not necessarily need to map onto choice options as in the present paper, but could relate to cues that specify which options are high or low in value (i.e. they are used to compute Q values that specify the value of a choice given the state). It seems unclear how reward probabilities would be calculated in this case.

If my understanding above is correct, my concerns is whether the particular finding presented here, i.e. additive contributions of state beliefs and utility values, is confined to cases in which the belief states are related to outcome probabilities in the way described above. In such a case, the authors should make clear that their findings might be limited in this regard.

Reviewer #2 (Remarks to the Author):

The authors have largely addressed my original concerns and I believe the manuscript has improved further. I still think the issue of connectivity would provide further support for the main conclusions and will get us a step closer to arguing more concretely about the proposed mechanism/computations (e.g. through the interactions of the identified nodes). Otherwise one could still argue that the reported clusters merely reflect an echo of the relevant variables without necessarily computing anything. I leave it up to the editors and authors to decide how to best deal with this issue.

Reviewer #3 (Remarks to the Author):

I think the revised paper has both strengths and weaknesses.

Strengths:

- As before, the paper reports an interesting new test of computational decision models in the context of reversal learning, and reports interesting dissociations between dmPFC and vmPFC.
- The paper now provides a clearer description of the rationale for the utility normalization step (to put the belief and utility attributes on commensurable scales so they can be integrated in a weighted sum), and shows this is helpful for fitting behavioral data.
- Many of the other additions also strengthen the paper, including the clarification of statistical tests, additional comparisons with reduced models, the new fMRI figure controlling for reaction times, and the description of the leave-one-out procedure.

Weaknesses:

My overarching reservation is that there are still weaknesses in the clarity of presentation, which make it difficult to assess certain aspects of the scientific contribution. To give examples:

- An important part of the paper is the derivation of the critical parameter value ω_0 , which makes the weighted-additive model equivalent to a multiplicative model. Unfortunately I still was not able to follow the authors' explanation of how they arrived at this result (line 729). The revisions serve to assert the result more strongly, but not to clarify how it was obtained. (I'm happy to be overruled on this point if the editor and other reviewers felt the derivation was clear.)
- I'm still unclear on the basic question of how fMRI results were corrected for multiple comparisons. If results were thresholded at voxelwise FWE-corrected $p < 0.05$, then it shouldn't be necessary to add a cluster size threshold to obtain results corrected at $p < 0.05$. It's not clear what cluster-forming threshold was used, how the cluster size thresholds ($k > 10$ or $k > 40$, for different analyses) were obtained, or why a more restricted search volume would require a higher cluster-size threshold to reach the same corrected alpha level.
- I didn't understand the response to point #1 from my earlier review, where I noted that the parameterized OPT model seemed to capture qualitative features of the data. The authors contend that Fig. S2B shows the model failing to match the data, but the model and data in that figure seem to be in rather good qualitative agreement (all the trends that are present in behavior seem to be present in the model results), even if the fit is quantitatively not as good as the other models. Furthermore, the mention of this in the main text is misleading (line 186), maybe just because of ambiguous wording. It says, "Reduced models with no distortions failed to reproduce participants' choices across all conditions both along trial series following reversals and according to proposed rewards." But the authors say in their response letter that the model did not fail at both of these. They argue it succeeded in the former case by failing in the latter case.
- I remain somewhat confused about how to understand the different theoretical significance of the linear and quadratic versions of the "chosen minus unchosen" regressors (my original point #8). These two terms would be expected to share a lot of variance because people tend to choose the better option, meaning the chosen-minus-unchosen difference tends to be positive—the authors refer to this as a sampling artifact (line 346) but it's quite fundamental. On the other hand, if the quadratic transformation is applied to a z-scored version of the chosen-minus-unchosen parameter (as implied by the x-axis labels in Fig. 6C), then the zero level no longer has a clear meaning.

Point-to-point response to reviewers' comments

We thank the reviewers for re-reviewing our paper. Below we describe how we revised our manuscript in response to each comment. We provided the additional data and made the clarifications suggested to further improve the paper.

NN: the exact changes made in the paper are shown **in red**.

Reviewer #1 (Remarks to the Author):

Rouault and colleagues investigated human decision making while outcome probability reversed and outcome magnitude fluctuated. Please refer to my original review for a full description of the study. The bottom line is that Rouault et al propose that choice results from the additive contributions of state beliefs and utility values, in contrast to the expected utility theory that integrates utilities over state beliefs to derive subjective probabilities for choice computation.

The study makes a major contribution to the literature and offers an interesting perspective on “irrational” decision making. It is also a great example of how model falsification approaches can and should be used. The authors had done an overall excellent job in addressing all concerns raised in my previous review, except one point which I will clarify below:

In my previous review I mentioned concerns relating to the fact that in the present study the beliefs states relate to reward probabilities (and hence are often termed affective belief states by the authors themselves) I think the authors have misunderstood my previous point, so I will reiterate here:

*As per equation for $BI(t/r)$ listed on page 20 in the paper, the ‘belief’ that a particular state might be true is related to the value difference for that state compared to other states. Doesn’t that mean that if this state is particularly worth choosing (because $V1-V2$ is big), then the belief in state 1 goes up? Belief states, however, can signify many more dimensions of the task. Based on *sensory* evidence, rather than reward evidence, a belief state could for instance reflect that a presented stimulus is more likely to be green than to be red, while green is generally the less rewarding option. Another issue to consider is that belief states do not necessarily need to map onto choice options as in the present paper, but could relate to cues that specify which options are high or low in value (i.e. they are used to compute Q values that specify the value of a choice given the state). It seems unclear how reward probabilities would be calculated in this case.*

If my understanding above is correct, my concerns is whether the particular finding presented here, i.e. additive contributions of state beliefs and utility values, is confined to cases in which the belief states are related to outcome probabilities in the way described above. In such a case, the authors should make clear that their findings might be limited in this regard.

We agree about the importance of properly identifying the potential limitations of the present study. First of all, our study addresses the general issue of how the beliefs about reward frequencies/probabilities associated with choice options combine with

reward values in decision-making. As you note, the beliefs about other dimensions (e.g. beliefs about reward magnitude, stimulus or option identity) were not manipulated and are consequently out of the scope of the present study: whether our findings generalize to these beliefs indeed remains an open question for future research.

As explained below, the question is unrelated to using value differences between proposed rewards to influence state beliefs. It relates to the notion of affective state beliefs defined here for latent states mapping reward frequencies onto choice options. Future investigations will determine whether this notion and the present findings extend to other dimensions.

Generally speaking, indeed, the beliefs about reward frequencies evolve across trials according to two types of events the person observes/experiences: (1) getting a reward vs. no rewards after choosing one option; (2) the occurrence of any external cues informing about the reward frequencies associated with choice options. In our study, cases (1) occurred in all experimental conditions (congruent, neutral, incongruent): getting a reward after choosing one option increases the belief that this option is associated with a larger reward frequency, while getting no rewards decreases this belief. Cases (2) occurred only in the congruent and incongruent condition. In these conditions, indeed, the value difference between the rewards that choice options propose acts as an informative cue about the associated reward frequencies: in the congruent condition, this value difference increases the belief that the option proposing the *larger* reward is associated with a larger reward frequency; in the incongruent condition, this value difference increases the belief that the option proposing the *lower* reward is associated with a larger reward frequency. The optimal observer model (model OPT) as well as the distortion and mixture model (models DIST and MIX, respectively) assume that as posterior probabilities over latent states, these beliefs evolve according to cases (1) and (2) through Bayesian inferences. Accordingly, beliefs evolve in the same way in the congruent and incongruent condition, i.e. irrespective of the valence of the value difference between proposed rewards. The value difference thus acts as a standard quantitative cue modulating the updating of beliefs across trials.

More importantly, models OPT/DIST assume these beliefs to act in decision-making as *probability measures* implicated in marginalization processes to compute expected utilities, while model MIX assumes these beliefs to act in decision-making as *value components* that add up to utility values. This difference is made possible because beliefs in model MIX bear upon a notion of latent states more restrictive than in models OPT/DIST. In models OPT/DIST, indeed, latent states are simply defined as “option#1 and option#2 are associated with reward frequencies q_M and q_m , respectively” and vice-versa. In model MIX, latent states need to be defined as “option#1 and option#2 are associated with the *larger and lower* reward frequency q_M and q_m , respectively” and vice-versa. For that reason, the manuscript refers to these latent states as *affective states*. Future research is thus required to generalise the notion of affective states to task dimensions other than the mapping between choice options and reward frequencies.

We revised our manuscript to make more explicit the potential limitation of the present study as follows:

(Abstract)

“The results reveal that the irrationality of human choices commonly theorized as deriving from optimal computations over false beliefs actually stems from suboptimal neural heuristics over rational beliefs **about reward contingencies.**”

(Discussion, fourth para.; p. 15)

“This model has important conceptual implications. First, beliefs about reward contingencies bear upon *affective* states corresponding to which choice option most surely leads to rewards (i.e. the safest one) rather than arbitrary-defined abstract states. Second, affective state beliefs act in decision-making as appetitive values rather than as abstract probability measures: choices derive from the independent (additive) contribution of multiple normalized value components including normalized utilities and affective state beliefs. Third, the notion of *subjective* probability appears as a theoretical construct reflecting the suboptimal combination of beliefs and utilities in decision-making rather than distorted cognitive/neural representations of objective probabilities. Fourth, risk aversion appears to stem from the dominant contribution of affective state beliefs to decision-making and not only from the concavity of utility function as in the standard expected utility theory (Bernoulli, 1954). **Whether the present findings and particularly, the notion of affective state beliefs generalize to beliefs bearing upon task dimensions other than reward probabilities associated with choice options (e.g. reward magnitude, stimulus or option identity) remains an open question for future investigation.**”

Thank you for your comment.

Reviewer #2 (Remarks to the Author):

The authors have largely addressed my original concerns and I believe the manuscript has improved further. I still think the issue of connectivity would provide further support for the main conclusions and will get us a step closer to arguing more concretely about the proposed mechanism/computations (e.g. through the interactions of the identified nodes). Otherwise one could still argue that the reported clusters merely reflect an echo of the relevant variables without necessarily computing anything. I leave it up to the editors and authors to decide how to best deal with this issue.

Following your suggestion, we conducted the functional connectivity analysis to provide additional support to the main conclusion of the paper.

Our original analyses provide evidence that: (1) the dmPFC encodes normalized utilities of choice options; (2) the dmPFC is involved in computing choices by linearly combining these utilities with *affective state beliefs encoded in the vmPFC*. This result predicts that the dmPFC functionally interacts with the vmPFC so that in addition to normalized utilities, state beliefs concurrently contribute to choice computations in the dmPFC. Accordingly, psychophysiological interactions (PPI) between the dmPFC and vmPFC should reflect the involvement of state beliefs in choice computations occurring in the dmPFC. Specifically, the correlation between

dmPFC and vmPFC activity should vary with state beliefs in the same way as dmPFC activations reflect the involvement of state beliefs in choice computations, i.e. *decrease* when (relative) chosen beliefs $B_{\text{chosen}} - B_{\text{unchosen}}$ increase.

To test this prediction, we performed the PPI analysis you suggested, using as seed region the dmPFC region identified in choice computations and especially exhibiting a *negative* linear effect of relative chosen beliefs (**Fig. 5**): dmPFC activity and the PPI “dmPFC activity x relative chosen beliefs” were then included as additional regressors in GLM#2 (comprising all regressors of interest: relative chosen beliefs and utilities along with their quadratic expansion in a full variance analysis). The prediction is that the vmPFC region encoding state beliefs should exhibit activations *negatively* associated with the PPI regressor described above.

Consistent with the prediction, we found that within this vmPFC region, a cluster of 95 voxels (2.6 cm³) exhibited activations negatively associated with the PPI regressor ($p < 0.05$. Peak at -6,50,-11, $T = 2.92$, $p = 0.004$) (all other effects being factored out). This finding provides evidence that the dmPFC functionally interacts with the vmPFC so that state beliefs encoded in the vmPFC concurrently contribute to choice computations in the dmPFC.

We added this analysis and these data to the revised manuscript in a new supplementary figure:

(Supplementary Fig. 5)

“Supplementary Fig. 5. Psycho-Physiological Interactions associated with relative chosen beliefs with the dmPFC as seed region. Green, vmPFC activations encoding relative beliefs irrespective of choice computations, i.e. associated with $(B_{\text{chosen}} - B_{\text{unchosen}})^2$ (same data as in **Fig. 6**). Blue, significant Psycho-Physiological Interactions (PPIs) within these vmPFC activations: the correlation between dmPFC and vmPFC activity in the blue voxels decreased when relative chosen beliefs increased ($p < 0.05$, cluster size: 95 voxels or 2.6 cm³, peak: $T = 2.92$, $p = 0.004$, Peak MNI coordinates: $x = -6$, $y = 50$, $z = -11$). Activations are superimposed on coronal and sagittal anatomical slices from MNI template (indexed by their MNI coordinate, neurological convention). PPIs were analyzed using the standard SPM8 method. The seed region (ROI) was the dmPFC activation reflecting choice computations shown in **Fig. 5**. Time series of ROI activity were extracted from single-subject activation peaks (associated with the negative linear effect of relative chosen beliefs) within this ROI and adjusted for factors of no interest (scanning runs, movements, no response trials). ROI activity regressors and PPI regressors modeling the interaction between ROI activity and relative chosen beliefs ($B_{\text{chosen}} - B_{\text{unchosen}}$) were then formed and added to GLM#2 (see **Methods**), which included all regressors of interest (relative chosen beliefs, relative

chosen utilities and their quadratic expansion) in a full variance analysis.”

Finally, we referred to these results in the Results section as follows:

(Results, fMRI activations, 4th para.; p. 12)

“[...] Thus, the dmPFC encoded comparative information about normalized utilities, while the vmPFC encoded comparative information about affective state beliefs, irrespective of choice computations. This dissociation further suggests that the dmPFC functionally interacts with the vmPFC so that as reported above, affective state beliefs contribute to choice computations in the dmPFC along with normalized utilities. We therefore analysed the functional connectivity between these regions (psychophysiological interactions) and consistently found that similar to the dmPFC activations reflecting choice computations (see Fig. 5), the correlation between dmPFC and vmPFC activations decreased when (relative) chosen beliefs increased (Supplementary Fig. 5).”

Thank you for your comment.

Reviewer #3 (Remarks to the Author):

I think the revised paper has both strengths and weaknesses.

Strengths:

- *As before, the paper reports an interesting new test of computational decision models in the context of reversal learning, and reports interesting dissociations between dmPFC and vmPFC.*
- *The paper now provides a clearer description of the rationale for the utility normalization step (to put the belief and utility attributes on commensurable scales so they can be integrated in a weighted sum), and shows this is helpful for fitting behavioral data.*
- *Many of the other additions also strengthen the paper, including the clarification of statistical tests, additional comparisons with reduced models, the new fMRI figure controlling for reaction times, and the description of the leave-one-out procedure.*

Weaknesses:

My overarching reservation is that there are still weaknesses in the clarity of presentation, which make it difficult to assess certain aspects of the scientific contribution. To give examples:

- *An important part of the paper is the derivation of the critical parameter value ω_0 , which makes the weighted-additive model equivalent to a multiplicative model. Unfortunately I still was not able to follow the authors' explanation of how they arrived at this result (line 729). The revisions serve to assert the result more strongly, but not to clarify how it was obtained. (I'm happy to be overruled on this point if the editor and other reviewers felt the derivation was clear.)*

We are sorry that our mathematical derivation (line 729) was unclear. To clarify, we first labelled two key equations in the **Methods**:

1) the equation describing choice log-odds from model OPT (p. 23):

$$\log \frac{p_1}{p_2} \simeq \frac{\beta}{2} [(P_1(t) + P_2(t))(V_1(t) - V_2(t)) + (V_1(t) + V_2(t))(P_1(t) - P_2(t))] \quad (\text{E4})$$

2) the equation describing choice log-odds from model MIX (p. 26):

$$\log \frac{p_1}{p_2} \simeq \beta [(1 - \omega)(v_1 - v_2) + \omega(B_1(t|r) - B_2(t|r))]. \quad (\text{E5})$$

In the revised manuscript, we then rewrote the mathematical derivations correspond to special case $\omega = \omega_0$ in more details for better explaining how parameter ω_0 was obtained. The point is that for $\omega = \omega_0$, equations (E4) and (E5) are equivalent:

(Methods, Computational modelling: alternative model MIX, special case $\omega = \omega_0$, p. 26)

“Model MIX further features a special case when $\omega = \omega_0 = \frac{q_M - q_m}{q_M + q_m + q_M - q_m}$. In that case, model MIX is equivalent to model OPT with normalized utilities and affective state beliefs, and vice-versa. **Indeed, consider first choice log-odds from model OPT with normalized utilities, which write as follow (see Eq. (E4) above):**

$$\log \frac{p_1}{p_2} \simeq \frac{\beta}{2} [(P_1(t) + P_2(t))(v_1 - v_2) + (v_1 + v_2)(P_1(t) - P_2(t))], \quad (\text{E6})$$

where $P_1(t), P_2(t)$ are inferred reward probabilities deriving from marginalizing over affective state beliefs $B_1(t|r), B_2(t|r)$:

$$P_1(t) = q_M B_1(t|r) + q_m B_2(t|r)$$

$$P_2(t) = q_m B_1(t|r) + q_M B_2(t|r).$$

Note that $B_1(t|r) + B_2(t|r) = 1$ and consequently,

$$P_1(t) + P_2(t) = q_M + q_m$$

$$P_1(t) - P_2(t) = (q_M - q_m)(B_1(t|r) - B_2(t|r)).$$

As $v_1 + v_2 = 1$, **Eq. (E6)** is therefore equivalent to:

$$\begin{aligned} \log \frac{p_1}{p_2} &\simeq \frac{\beta}{2} [(q_M + q_m)(v_1 - v_2) + (q_M - q_m)(B_1(t|r) - B_2(t|r))] \\ &= \frac{\beta}{2} [(q_M + q_m) + (q_M - q_m)] \left[\frac{(q_M + q_m)}{(q_M + q_m) + (q_M - q_m)} (v_1 - v_2) + \frac{(q_M - q_m)}{(q_M + q_m) + (q_M - q_m)} (B_1(t|r) - B_2(t|r)) \right] \\ &= \beta q_M \left[\left(1 - \frac{(q_M - q_m)}{(q_M + q_m) + (q_M - q_m)}\right) (v_1 - v_2) + \frac{(q_M - q_m)}{(q_M + q_m) + (q_M - q_m)} (B_1(t|r) - B_2(t|r)) \right]. \end{aligned}$$

Consequently, **Eq. (E6)** writes as follow:

$$\log \frac{p_1}{p_2} \simeq \beta q_M [(1 - \omega_0)(v_1 - v_2) + \omega_0(B_1(t|r) - B_2(t|r))] \quad (\text{E7})$$

with $\omega_0 = \frac{(q_M - q_m)}{(q_M + q_m) + (q_M - q_m)}$. **Eq. (E7)** is identical to **Eq. (E5)** describing choice log-

odds from model MIX with $\omega = \omega_0$ (with inverse temperature βq_M). Thus, model OPT with normalized utilities and affective state beliefs is equivalent to model MIX with $\omega = \omega_0$, and vice-versa: maximizing a weighted sum of normalized utilities and affective state beliefs with weight parameter ω_0 is equivalent to maximizing the expected values of normalized utilities.”

We hope that this clarifies the derivation of parameter ω_0 .

- I'm still unclear on the basic question of how fMRI results were corrected for multiple comparisons. If results were thresholded at voxelwise FWE-corrected $p < 0.05$, then it shouldn't be necessary to add a cluster size threshold to obtain results corrected at $p < 0.05$. It's not clear what cluster-forming threshold was used, how the cluster size thresholds ($k > 10$ or $k > 40$, for different analyses) were obtained, or why a more restricted search volume would require a higher cluster-size threshold to reach the same corrected alpha level.

As mentioned in the manuscript, we exactly followed the Statistical Parametric Mapping (SPM) method based on the Random Field Theory (RFT) developed in the 90's (original paper: Worsley et al., 1996; see also the book: Frackowiak, Friston, Frith, Dolan, Price, Zeki, Ashburner, and Penny, (Eds), Human Brain Function. Academic Press, 2003) and which forms the statistical core in all SPM softwares. Below is a brief description of the method that should answer your questions:

1-Note first that the *cluster-wise* threshold $p_{cluster} < 0.05$ is not a correction for multiple comparison. This threshold determines the minimal significant size of activations as explained below.

2-A fundamental problem in fMRI is that adjacent voxels potentially belong to the same functional cluster: activations in adjacent voxels are potentially correlated in an unknown fashion, which considerably complicates the correction for multiple comparisons. To overcome this issue, SPM applies a known Gaussian spatial smoothing in order to insert a much stronger but systematic and known correlation and to properly use the Gaussian RFT for correcting for multiple comparisons.

3-As a result, one activated voxel will lead to a spatially Gaussian-distributed cluster of activations centered on this voxel. The cluster-wise threshold $p_{cluster} < 0.05$ thus determines the significant *number* (denoted k) of voxels exhibiting activations that through this smoothing procedure, should derive from *one* activated voxel. Number k depends of course on how we define “voxels exhibiting activations”: the more this definition is stringent, the lower number k .

4-So, one need to determine what is a “*voxel exhibiting activations*”: a voxel is defined as exhibiting activations when its activity is statistically significant, which yields to define the so-called *voxel-wise* activation threshold $p_{voxel} < 0.05$ corresponding to a given T -value. Number k therefore depends on this T -value.

5-Note that so far, there is no correction for multiple comparisons. The method therefore uses a *voxel-wise* activation threshold $p_{voxel} < 0.05$ corrected for multiple comparisons (here we used the FWE correction). Of course, the correction depends on the number of comparisons, i.e. the total number of tested voxels or the so-called *search volume* (the correction is also adjusted according to the applied Gaussian spatial smoothing, as the smoothing inserts known correlations across voxels).

6-In summary, the method first defines voxels exhibiting activations, using a *voxel-wise* activation threshold $p_{voxel} < 0.05$ corrected for multiple comparisons. Given this

definition, the *statistical inference* is then based on the size of activation clusters: according to the Gaussian spatial smoothing, significant *activation clusters* at alpha-level threshold $p_{cluster} < 0.05$ must comprise at least k voxels exhibiting activations.

6-In our first analysis (**Fig. 4**), the search volume was the whole brain: the FWE-corrected *voxel-wise* threshold $p_{voxel} < 0.05$ leads to a given T -value $T_{voxel}(\text{brain})$. Given $T_{voxel}(\text{brain})$, the cluster-wise threshold $p_{cluster} < 0.05$ leads to a voxel number $k=10$ voxels.

7-In our subsequent analyses, the search volume was the frontal lobes: the FWE-corrected *voxel-wise* activation threshold $p_{voxel} < 0.05$ leads to another T -value $T_{voxel}(\text{frontal})$. Given $T_{voxel}(\text{frontal})$, the cluster-wise threshold $p_{cluster} < 0.05$ leads to a voxel number $k=40$ voxels.

8-Note that number k is larger in the latter than former case, because $T_{voxel}(\text{frontal})$ is lower than $T_{voxel}(\text{brain})$, (resulting from a FWE-correction over a smaller search volume). One might view as counter-intuitive that a smaller search volume yields to a larger extend threshold. This derives, however, from the exact application of the rigorous RFT and standard SPM method: a less stringent activation threshold T_{voxel} yields to a larger extend threshold k . In both cases, extend threshold k corresponds to identical alpha-level thresholds $p_{cluster} = 0.05$ (as well as to identical alpha-level thresholds p_{voxel}).

We hope that this brief description responds to your questions. As a final note, the first analysis (**Fig. 4**) actually yields to the same significant dmPFC activations when we use the (too much) conservative extend threshold $k=40$ voxels instead of $k=10$ voxels. Indeed, $k=40$ voxels corresponds to a volume equal to $40 \times \text{voxel size} = 40 \times 0.3 \times 0.3 \times 0.3 \text{ cm}^3 = 1.08 \text{ cm}^3$. **Supplementary Table 3** shows that the volume of these dmPFC activations was equal to 1.1 cm^3 (41 voxels).

Following your comment, we provided more details in the revised manuscript about the SPM method and the derivation of cluster sizes k corresponding to the cluster-wise alpha-level $p=0.05$:

(**Methods, fMRI statistical analyses**, 2nd para.; p. 30):

“Following the standard SPM method, second-level parametric maps were then obtained for each contrast over the group of participants. Activated voxels were identified using a significance voxel-wise threshold set at $p < 0.05$ corrected for family-wise errors for multiple comparison over the search volumes (see below). Statistical inferences were based on cluster level: significant activations were identified as clusters of activated voxels with significance cluster-wise threshold set at $p < 0.05$. [...]”

(**Methods, fMRI statistical analyses**, 1-GLM; p. 30):

“[...]. This analysis was performed over the whole brain. Given this search volume and the resulting FWE-correction over voxel-wise T -values, the cluster-wise threshold set at $p < 0.05$ then corresponded to a cluster size $k > 10$ voxels (0.27 cm^3).”

(**Methods, fMRI statistical analyses**, 2-GLM; p. 31):

“[...]. This analysis was performed over the frontal lobes only (MNI coordinate $Y > 0$). Given this search volume and the resulting FWE-correction over voxel-wise T -values, the cluster-wise threshold set at $p < 0.05$ then

corresponded to a cluster size $k > 40$ voxels (1.08 cm³).”

(Methods, fMRI statistical analyses, 3-GLM; p. 32):

“[...] As in GLM#2, this analysis was performed over the frontal lobes only (MNI coordinate $Y > 0$). Consequently, the cluster-wise threshold set at $p < 0.05$ again corresponded to a cluster size $k > 40$ voxels (1.08 cm³).”

- I didn't understand the response to point #1 from my earlier review, where I noted that the parameterized OPT model seemed to capture qualitative features of the data. The authors contend that Fig. S2B shows the model failing to match the data, but the model and data in that figure seem to be in rather good qualitative agreement (all the trends that are present in behavior seem to be present in the model results), even if the fit is quantitatively not as good as the other models. Furthermore, the mention of this in the main text is misleading (line 186), maybe just because of ambiguous wording. It says, "Reduced models with no distortions failed to reproduce participants' choices across all conditions both along trial series following reversals and according to proposed rewards." But the authors say in their response letter that the model did not fail at both of these. They argue it succeeded in the former case by failing in the latter case.

We acknowledge that in the main text, our wording was ambiguous and potentially misleading. The fact is that in **Fig. 2SB** the performance of reduced models *significantly* differed from human data especially in the congruent and incongruent condition. By contrast, there were no significant differences for model DIST and MIX ($p > 0.05$) We acknowledge that though fairly evident from the figures, we did not mention it explicitly. We revised our manuscript accordingly:

(Results, behavioral results, first para.; p. 7)

“[...] Importantly, model DIST simulations *reproduced* participants' choices in every condition *both* in choosing true best bandits along trial series following reversals (**Fig. 2A**) *and* in bandits' choices according to proposed rewards (**Fig. 2B**)(no significant differences at $p < 0.05$). [...] . Simulations of reduced models with no distortions significantly differed from participants' performances especially in the congruent and incongruent condition and with respect to proposed rewards (**Supplementary Fig. 2**)”

(Results, behavioral results, fourth para.; p. 9)

“[...] Importantly, model MIX simulations *reproduced* participants' choices in every condition *both* in choosing true best bandits along trial series following reversals (**Fig. 2A**) *and* in bandits' choices according to proposed rewards (**Fig. 2B**)(no significant differences at $p < 0.05$). [...]”

We also revised **Supplementary Fig. 2B** by explicitly indicating the significant differences between model and human data:

Sup. Fig. 2 (related to Figs. 2 & 3). Simulations of fitted models RL (with proposed rewards) and OPT (parameterized).

Proportions of choosing true best bandits (maximizing reward frequency \times proposed reward monetary value) according to trial number after reversals (A) and choice proportion of proposed rewards (B) for participants (blue, same data as in Fig. 1), model OPT (parameterized, gray) and model RL (including proposed rewards, red). Data are shown separately for the conditions congruent, neutral and incongruent. Parameterized model OPT corresponds to model DIST with no distortions on reward probabilities and values, and was fitted to human data with the same free parameters except the four parameters specifying distortions (see Supplementary Table S2). Model RL shown here included four free parameters: inverse temperature β , lapse rate ϵ , learning rate α , and contribution of proposed rewards to RL values φ at decision time. Error bars are s.e.m. across participants. Note that none of the models succeed in reproducing human data especially according to proposed rewards. In B, significant differences between model and human data are indicated (** $p < 0.01$; *** $p < 0.001$).

Finally, note that our model falsification approach rules out model DIST and all its reduced models (see Results, Behavioral results, third para.).

- I remain somewhat confused about how to understand the different theoretical significance of the linear and quadratic versions of the "chosen minus unchosen" regressors (my original point #8). These two terms would be expected to share a lot of variance because people tend to choose the better option, meaning the chosen-minus-unchosen difference tends to be positive—the authors refer to this as a sampling artifact (line 346) but it's quite fundamental. On the other hand, if the quadratic transformation is applied to a z-scored version of the chosen-minus-unchosen parameter (as implied by the x-axis labels in Fig. 6C), then the zero level no longer has a clear meaning.

Regarding your second point first, you rightly mention that the quadratic transformation must be performed *before* z-scoring the variable to preserve the zero-level meaning in quadratic regressors. This is exactly what we did in all regression analyses. We revised our manuscript to make this point more explicit as follow:

(Methods, fMRI statistical analyses, first para.; p. 30):

“[...] All parametric modulations were z-scored to ensure between-subjects and between-regressors comparability of regression coefficients (quadratic regressors were z-scored after applying the quadratic transformation). [...]”

The graphs in **Fig. 6C** (as in **Figs. 5B**) are based on a different approach complementary to regression analyses. The graphs show the variation of activations based on binning the model variables across trials rather than on parametric regression analyses. This bin analysis requires building the same bins across subjects, in order to properly average across subjects and to reveal the variations of activations within-subjects (and not improperly, between-subjects). And building the same bins require normalizing the variables across subjects, *i.e.* z-scoring variables in order to avoid empty bins in some subjects (subjects exhibit distinct amplitudes, increments and sampling in model variables as captured in model free parameters). z-scoring evidently preserves the presence of linear and quadratic components in activations but as you mention, does not preserve the meaning of the zero level. This is the unavoidable limitation of the bin analysis compared to the regression analysis. By contrast, the bin analysis is unaffected by possible sampling biases you mention due to subjects' choices towards the better option (see below). Thus, the binning and regression analyses are complementary. This is the reason why we also provided in **Figs 5B, 6B** the quantitative results (beta coefficients) from regression analyses (preserving the zero-level as indicated above) along with the results from the binning analyses.

Second, as you rightly note, the linear and quadratic expansion regressor associated with a variable $X_{\text{chosen}}-X_{\text{unchosen}}$ driving choices share some variance as choices more frequently sample positive values of $X_{\text{chosen}}-X_{\text{unchosen}}$ (sampling bias). However, the linear and quadratic function mathematically form orthogonal regressors, so that the shared variance reflects only the effects due to the sampling bias, while the linear and quadratic regressor entirely captures the linear and quadratic component, respectively. In other words, the shared variance captures no genuine linear and quadratic effects but only the sampling bias. In a full variance regression analysis including multiple regressors, any shared variance is assigned to the residuals, while the variance attributed to each regressor capture the specific contribution of each regressor to activations. For linear and quadratic regressors consequently, the residuals capture the effects due to the sampling bias, while the variance attributed to the linear and quadratic regressor entirely captures the linear and quadratic component in activations, respectively. Accordingly, the full variance regression analysis is the proper way to investigate the linear and quadratic effects in activations, even in presence of sampling biases.

For that reason, all results reported in the paper are based on full variance regression analyses. As indicated in the manuscript, the presence of negative linear effects in activations reflects selection processes based on variable X , while the presence of positive quadratic effects (with a quadratic regressor properly centered on zero as indicated above) reflects the encoding of variable X , irrespective of selection processes (see also Duverne & Koechlin, *Cerebral Cortex*, 2017). For variable X =state beliefs, we found a negative linear effect in the dmPFC and a positive quadratic effect in the vmPFC (**Figs. 5 & 6**), indicating that the vmPFC encodes state beliefs irrespective of choice, while the dmPFC computes choices based on state beliefs. For variable X =normalized utilities, we found both effects in the dmPFC (**Figs. 5 & 6**). The presence of both effects in the same region has a first, direct interpretation: the region is involved both in encoding variable X irrespective of choices and in computing choices based on variable X . However, one might alternatively consider that the region exhibits activations varying as a “pure”

quadratic function of variable X centered on a non-zero value (indeed, $X^2 - aX = (X - a/2)^2 + \text{constant}$). We dismissed this alternative interpretation because as you mention, a non-zero value $a/2$ has no significance in the present protocol. (moreover, Duverne & Koechlin (2017) provides empirical evidence ruling out this alternative interpretation: they found that only the zero-centered quadratic effect remained present when choices were independent of variable X). We thus conclude that the dmPFC both encodes normalized utilities and computes choices based on these normalized utilities (along with state beliefs). This interpretation is further in agreement with the inverted quadratic effect observed in the dmPFC and associated with the decision variable linearly combining utilities and beliefs.

What might be confusing is that in the Results section, we also briefly reported the results from a control regression analysis whereby quadratic regressors were orthogonalized onto linear regressors in order to (arbitrary) assign the shared variance to these linear regressors rather than to residuals. In that case the variance attributed to linear regressors is identical to that obtained from a regression analysis including only linear regressors (while the variance attributed to quadratic regressors remains identical to that obtained in the full variance regression analysis). The rationale to report these results is that this analysis potentially allows replicating published data from previous studies investigating only linear effects in regression analyses. However, one should be aware that in that case, the variance attributed to linear regressors mixes both true linear effects *and the sampling bias*.

Note that for normalized utilities, the sampling bias was actually small because according to model MIX, normalized utilities had a small contribution to choices. For normalized utilities, our control analysis is therefore expected to provide the same results as the full variance analysis. As mentioned in the manuscript, we indeed found virtually the same results and especially no additional activations exhibited linear effects associated with relative chosen utilities.

For state beliefs, by contrast, the sampling bias was large because state beliefs predominantly contribute to choices. As expected, thus, the control analysis showed the same activations as the full variance analysis *plus* additional activations associated with the linear regressor $X = B_{\text{chosen}} - B_{\text{unchosen}}$. Replicating previous studies as mentioned in the manuscript, these “linear” activations were found in the vmPFC (in addition to the quadratic effect of relative state beliefs (i.e. $[B_{\text{chosen}} - B_{\text{unchosen}}]^2$) also found in the full variance analysis). However, because these “linear” activations were not found in the full variance analysis, we can conclude that they reflect the sampling bias (or a sampling artifact as termed in the manuscript) rather than a genuine linear effect. **Fig. 6C** (right panel) illustrates the point: while vmPFC activations appeared to vary as a pure quadratic function of $X = B_{\text{chosen}} - B_{\text{unchosen}}$, it is easy to see as shown below that *when removing the quadratic regressor* (or orthogonalising it onto the linear regressor), these vmPFC activations exhibit a positive linear effect associated with X *due to the sampling bias*:

moving the quadratic regressor
(or orthogonalising it onto the linear regressor)

We do agree that the point is quite fundamental: previous studies have most often overlooked this problem of sampling biases in interpreting the results of regression analyses comprising only linear regressors from variables driving choices. By contrast, our analyses provide a more rigorous approach that allowed us to identify and overcome this issue, as explained above.

We acknowledge that our original paper does not provide enough methodological details regarding this important point. Accordingly, we revised our manuscript as follows. In **Results** first, we clarified the corresponding paragraph by reporting only the results from the full variance regression analysis:

(**Results, fMRI activations**, 5th para.; p. 12):

“All the effects reported above remained significant when the analysis factored out reaction times. Consistent with previous findings (see **Discussion**), activations were also found to linearly increased with (relative) chosen utilities. These activations were located in the vmPFC (**Fig. 7A**). Importantly, all these results were properly obtained in full variance regression analyses (see **Supplementary Methods**). As expected also, additional fMRI analyses revealed no significant brain activations associated with distorted subjective probabilities posited in model DIST ($p > 0.01$, uncorrected).”

In **Discussion** secondly, we reported and discussed the results of the control analysis replicating previous findings as follow:

(**Discussion**, 6th para.; p. 16):

“[...]. We further note that vmPFC activations exhibited no linear effects associated with (relative) chosen beliefs. However, we found that consistent with previous studies using only linear regressors (Boorman et al., 2009; Donoso et al., 2014; Hampton et al., 2006), vmPFC activations correlated with this linear regressor, *only when* the corresponding quadratic regressor was removed or equivalently, projected onto this linear regressor. This operation arbitrary assigned the share variance between the two regressors to the linear one. Furthermore, this shared variance only stems from sampling biases due to participants’ choices (choices more frequently matched the larger belief, otherwise the linear and quadratic regressor would be orthogonal). We then conclude that this linear correlation is likely to reflect a sampling artifact (see **Methods**). Consistently, **Fig. 6C** shows that vmPFC activations varied as a “pure” quadratic function of (relative) chosen beliefs. Our results thus qualify previous findings and show that the vmPFC encodes the beliefs regarding

which option most surely leads to rewards rather than the beliefs associated with the chosen option [...].”

Finally, we added **Supplementary Methods** in the supplementary Information document to describe in details the methodology of our regression analyses comprising linear and quadratic regressors:

(**Supplementary Information**, p. 2):

“Supplementary Methods

Note that the linear and quadratic expansion regressor associated with a variable $X_{\text{chosen}} - X_{\text{unchosen}}$ driving choices share some variance as choices more frequently sample positive values of $X_{\text{chosen}} - X_{\text{unchosen}}$ (sampling bias). However, the linear and quadratic function mathematically form orthogonal regressors, so that the share variance captures only the effects due to the sampling bias, while the linear and quadratic regressor entirely capture the linear and quadratic component, respectively. In a full variance regression analysis including multiple regressors, the shared variance is assigned to the residuals, while the variance attributed to each regressor capture the specific contribution of each regressor to activations. Accordingly, the full variance regression analysis properly assigns the sampling bias to the residuals, while the variance attributed to the linear and quadratic regressor entirely captures the linear and quadratic component in activations, respectively.

For that reason, all results reported in the paper were based on full variance regression analyses. As indicated in the main text, the presence of a negative linear effect in activations reflects selection processes based on variable X , while the presence of a positive quadratic effect (with a quadratic regressor properly centered on zero corresponding to maximal coding entropy) reflects the encoding of variable X , irrespective of selection processes (see also Duverne and Koechlin, 2017). The presence of both effects in the same region has a direct interpretation: the region is involved both in encoding variable X irrespective of choices and in computing choices based on variable X . Note that one might alternatively consider that this region exhibits activations varying as a “pure” quadratic function of variable X centered on a non-zero value (indeed, $X^2 - aX = (X - a/2)^2 + \text{constant}$). We dismissed this alternative interpretation because a non-zero value $a/2$ has no significance in the present protocol (moreover, Duverne & Koechlin (2017) provided empirical evidence ruling out this alternative interpretation: they found that only the zero-centered quadratic effect remained present when choices were independent of variable X). As the dmPFC was found to exhibit both effects associated with normalized utilities (**Figs. 5 & 6**), we thus concluded that the dmPFC both encodes normalized utilities and computes choices based on these normalized utilities (along with state beliefs as the dmPFC also showed a negative linear effect associated with beliefs, **Figs. 5 & 6**). This interpretation is further in agreement with the inverted quadratic effect observed in the dmPFC and associated with the decision variable linearly combining utilities and beliefs.

In order to replicate previous studies investigating only linear effects, we also conducted regression analyses whereby quadratic regressors were projected/orthogonalized onto linear regressors. This operation arbitrarily assigns the shared variance to linear regressors rather than to residuals. The variance attributed to linear regressors is then identical to that obtained from a regression analysis including only linear regressors (while the variance attributed to quadratic regressors remains identical to that obtained in the full variance regression analysis). In that case, however, the variance attributed to linear regressors mixes both true linear effects *and the effects of sampling biases*.

For normalized utilities, the sampling bias was actually small because according to model MIX, normalized utilities had a small contribution to choices. Consistently, this regression analysis provided virtually the same results regarding normalized utilities as the full variance regression analysis reported in the paper (**GLM#2**): in particular, no additional activations exhibited linear effects associated with relative chosen utilities. For state beliefs, by contrast, the sampling bias was large because state beliefs predominantly contribute to choices. As expected, thus, the control analysis showed the same activations as the full variance analysis *plus* additional activations associated with the linear regressor $X=B_{\text{chosen}} - B_{\text{unchosen}}$ (see **Discussion**). These additional “linear” activations were found in the vmPFC (in addition to the quadratic effect of relative state beliefs found in the full variance analysis). As these “linear” activations were not found in the full variance analysis, we can conclude that they reflect the sampling bias rather than a genuine linear component in activations.”

Thank you for your comments.

REVIEWERS' COMMENTS:

Reviewer #1 (Remarks to the Author):

The authors have answered all my comments and I am happy to fully recommend this very interesting manuscript for publication.

Reviewer #2 (Remarks to the Author):

Thank you for performing the additional connectivity analysis. All of my concerns have now been adequately addressed.

Reviewer #3 (Remarks to the Author):

I appreciate the effort the authors have spent responding to my comments. The added information is somewhat helpful. I still cannot reconcile their description of their approach to multiple comparisons correction with my understanding of RFT and cluster thresholding, but perhaps it is I who am confused. My overall assessment of the paper's strengths and weaknesses remains similar to what I expressed in my previous review.

Point-to-point response to reviewers' comments

REVIEWERS' COMMENTS:

Reviewer #1 (Remarks to the Author):

The authors have answered all my comments and I am happy to fully recommend this very interesting manuscript for publication.

Reviewer #2 (Remarks to the Author):

Thank you for performing the additional connectivity analysis. All of my concerns have now been adequately addressed.

Reviewer #3 (Remarks to the Author):

I appreciate the effort the authors have spent responding to my comments. The added information is somewhat helpful. I still cannot reconcile their description of their approach to multiple comparisons correction with my understanding of RFT and cluster thresholding, but perhaps it is I who am confused. My overall assessment of the paper's strengths and weaknesses remains similar to what I expressed in my previous review.

Note that reviewers have no remaining concerns.